# Interaction and functional specialization across a distributed neural circuit for flexible task control in macaques

K. Marche [1,5] ✉, N. Trudel[1,5], U. Schüffelgen[1], S. Smart[2], C. M. Harbison[1], J. Scholl [3], J. Sallet [1,4], F-X Neubert[1], J. Algermissen [1], N. Khalighinejad [1], JX O'Reilly[1], MC Klein-Flügge [1,6] & MFS Rushworth [1,6]

Reversal tasks have been regarded as probes of behavioural inhibition and linked to prefrontal and specifically orbitofrontal cortex. The centrality of behavioural inhibition to reversal task performance and the task's dependence on particular prefrontal sub-regions have, however, been questioned in primates. Using a combination of whole brain recording, transient ultrasonic disruption, two types of reversal task, and a task model emphasizing identification of transitions between latent states, we show that male macaques track latent state transitions in addition to choice values in reversal tasks. Activity reflecting both these features is prominent in dorsomedial frontal cortex, and anterior and dorsomedial thalamus when, and just before, animals select choices. By contrast, hippocampal activity continually tracks the probability of a reversal between latent states. We identify patterns of activity interaction spanning the three nodes of this circuit and demonstrate that disruption of each leads to reversal task impairment albeit in different ways.

For many decades, behavioural flexibility has been assessed by examining how animals and people choose between two options that periodically change or "reverse" in value. Initially, one option is the better one to choose because it is more likely to lead to a reward. Subsequently, however, the first option becomes less likely to be rewarded and the alternative more likely to be rewarded. The ability to switch between one option and the other has been linked to the prefrontal cortex[1,2]. In recent years, however, there have been two important developments.

First, traditional views of what makes reversal tasks so challenging have evolved. Earlier accounts focused on the difficulty of inhibiting responses to objects or locations that were previously rewarding but which are no longer rewarding after reversal[3]. Instead, more recent accounts have emphasised the possibility that animals learn, despite the absence of any explicit cue, that they may enter one of two (or

more) states that are latent within the task. Even though there may be nothing directly observable in the environment to inform animals when they are in one state or the other, they nevertheless infer that in one state choosing one option is rewarding while in the other state, the other option is more rewarding[4,5]. This account contends that what makes reversal tasks so taxing for animals is the need to learn the existence of two or more latent states and to infer the one they are currently in. Animals may continually estimate the probability of a transition from one state to the other—a reversal estimate—instead of, or as well as, estimating the value of each choice they make by monitoring the frequency with which it is followed by reward[6,7].

Second, there has been a change in our understanding of which brain circuits mediate the performance of reversal tasks. For many years, the orbitofrontal cortex (OFC) has been linked to reversal tasks[1,2], but the centrality and uniqueness of its role, at least in

[1]Oxford Centre for Integrative Neuroimaging (OxCIN), Department of Experimental Psychology, University of Oxford, Oxford, UK. [2]Oxford Centre for Integrative Neuroimaging (OxCIN), Nuffield Department of Clinical Neurosciences, University of Oxford, Oxford, UK. [3]Université Claude Bernard Lyon 1, CNRS, INSERM, Centre de Recherche en Neurosciences de Lyon (CRNL), U1028 UMR5292, Le Vinatier Psychiatrie Universitaire Lyon Métropole, Bron, France. [4]INSERM U1208 Stem Cell and Brain Research Institute, University of Lyon, Bron, France. [5]These authors contributed equally: K. Marche, N. Trudel. [6]These authors jointly supervised this work: MC Klein-Flügge, MFS Rushworth. ✉e-mail: kjl.marche@gmail.com

primates, has been questioned[3]; when OFC lesions in macaques spare the white matter subjacent to the cortex, then they do not impair reversal task performance. The results suggest that other brain regions that are interconnected by white matter adjacent to OFC are important. One such region is the adjacent area 47/12o. Area 47/12o is situated just lateral to the lateral orbitofrontal sulcus in a region that has sometimes been treated as either the most lateral part of the OFC or as the most ventral part of ventrolateral prefrontal cortex; it is critical for accurate credit assignment of outcomes to choices but this process is most taxed by probabilistic reversal tasks with multiple options rather than simpler reversal tasks[8,9]. In summary, while it might be correct to reconceptualise reversal tasks as situations requiring animals to infer which latent state they are in and the probability of switching between them[5,7], it is not clear that this process is mediated by OFC or 47/12o. Other brain regions that are interconnected by white matter adjacent to OFC may be critical, but the identities of the regions most important for simple reversal tasks are unclear. Fibres interconnecting other prefrontal and cingulate regions to the temporal cortex, hippocampus, amygdala, thalamus, striatum, midbrain, and brainstem pass near OFC[10], raising the possibility that some combination of these areas, rather than OFC itself, may mediate reversal behaviour.

Here we attempt to re-examine how macaques perform reversal tasks, first, by recording across the whole brain and looking not just for neural activity tracking the values of choices but also testing whether activity reflects an estimate of the probability of a transition from one state to another: a reversal estimate. We employ a Bayesian model to estimate the choice values that monkeys might learn from feedback after taking choices, their uncertainty about those values, but also the probability of reversal from one task state to another. Importantly, we compare neural activity recorded during performance of a reversal task with neural activity recorded when animals perform a task with a similar pair of options but where the values of each option are independent of one another and change independently of one another. Because the options' values are negatively correlated with one another in our first task, which most closely resembles a classic probabilistic reversal learning task, we refer to it as the "correlated" task. Because the values of the two choices in the control task are unrelated to one another, we refer to this task as the "uncorrelated" task.

Functional magnetic resonance imaging (fMRI) was used to look across the whole brain for neural activity linked to choice value, choice uncertainty, and reversal estimates from the Bayesian model. Then, in the regions that had been identified, we next examined the impact of transient alteration of neural activity using offline transcranial ultrasound stimulation (TUS), a technique that can be used to manipulate neural activity even deep in the brain without affecting overlying tissue[11–19]. By comparing activity between the two tasks, we identify three regions, one in dmPFC (extending from the dorsal bank of the cingulate sulcus to the superior frontal gyrus), medial dorsal thalamus, and mid hippocampus, where activity reflects the probability of reversal between task states but in distinct and complementary ways, either in isolation or in addition to other task variables. We then showed that transiently manipulating each of the three regions with TUS changed the way animals performed the task; animals performing the correlated task were no longer guided by an estimate of the probability of reversal between states and instead explored options as a function of their uncertainty about them.

## Results

The trial structure experienced in both the correlated and uncorrelated tasks is summarised in Fig. 1a. Animals saw two green rectangular stimuli on the left and the right of a computer monitor. Touching one of the sensors led to the replacement of the green rectangle with a white rectangle and the disappearance of the unchosen green rectangle for a delay period varying between 0.103 s and 8.770 s (mean 3.394 s, variance 0.605 s). The green rectangle on the chosen side then reappeared at the same time as a reward was delivered when choices were rewarded, or appeared as a grey rectangle if no reward was obtained. In the context of the relatively fast haemodynamic response function of the macaque brain[20], the imposition of this delay was sufficient to separate activity related to the monkeys' decisions and their outcomes.

Reward associations in both the correlated and uncorrelated tasks (referred to as "correlated" and "uncorrelated" in the figures) were probabilistic rather than deterministic. In general, however, in the correlated task, when one option had a high probability of reward, then the other option's probability of reward was low and close to zero (Fig. 1b). Although the correlation between the reward probabilities associated with the two options was not exactly $r = -1$ in the correlated task but averaged approximately −0.798 across all schedules used, this correlation was consistently more negative than in the uncorrelated sessions (Fig. 1c; the bars indicate group mean performance but data for each individual session are plotted as circles and the circles in each column in this figure and elsewhere are the data for each of the four individual macaques). In the uncorrelated task, both options changed their reward probabilities independently of each other, with the correlation between them close to zero. By employing probabilistic rather than deterministic schedules, it was possible to limit correlation between the key explanatory variables, choice values, choice uncertainties, and reversal estimate from the Bayesian model which, as explained below, account for the behaviour and neural activity recorded (Supplementary Fig. S1). Ensuring that correlation between variables is eliminated in this way is a prerequisite for ensuring an absence of collinearity between regressors, which would be problematic in the regression analyses that we used to examine behaviour, neural activity, and the impact of TUS.

### Behaviour patterns indicate macaques track the probability of reversal between task states in the "correlated" task

We examined behaviour as animals switched from one option to another. First (Fig. 1d), we analysed the data in ways that were independent of the application of any model of behaviour and simply considered whether receiving no reward for a choice made animals likely to switch to the alternative on the next trial. This was indeed the case in both tasks. On average, animals switched choice from one trial to the next on ~13.84% and 16.09% trials in the correlated and uncorrelated task, respectively but, after one reward absence, animals switched to the alternative option at higher rates of, on average, 46.80 and 43.45% of trials in the correlated and uncorrelated task respectively (Fig. 1d). While these switching rates in the two tasks were approximately similar after one error, macaques were significantly more likely to switch after either two and three reward absences in the correlated than the uncorrelated sessions (on average, 73.01 and 52.85%, respectively, after two reward absences; 39.46 and 23.65%, respectively, after three reward absences). Indeed, an analysis of variance (ANOVA) with factors of session type (correlated versus uncorrelated) and number of consecutive errors (1, 2, or 3) showed an interaction between session type and error number ($F_{2,204} = 7.0002$, $P = 0.0012$). Such strategies are adaptive in the two task environments. If no reward has been received over two or three successive trials for taking the same option in the correlated task, it is likely that the animal has entered the other latent task state in which it is the other option will be rewarded. In other words, knowledge of the negative correlation structure and the presence of two states latent in the task allows for fast behavioural updates. This is not necessarily the case in the uncorrelated task. While switching away from an option that is repeatedly unrewarded is still adaptive, the outcomes received for one choice option do not provide any information about latent task states and the reward probability of the alternative, thus encouraging more persistence in animal's choices. In the uncorrelated task, it can sometimes be adaptive for an animal to continue making the same choice

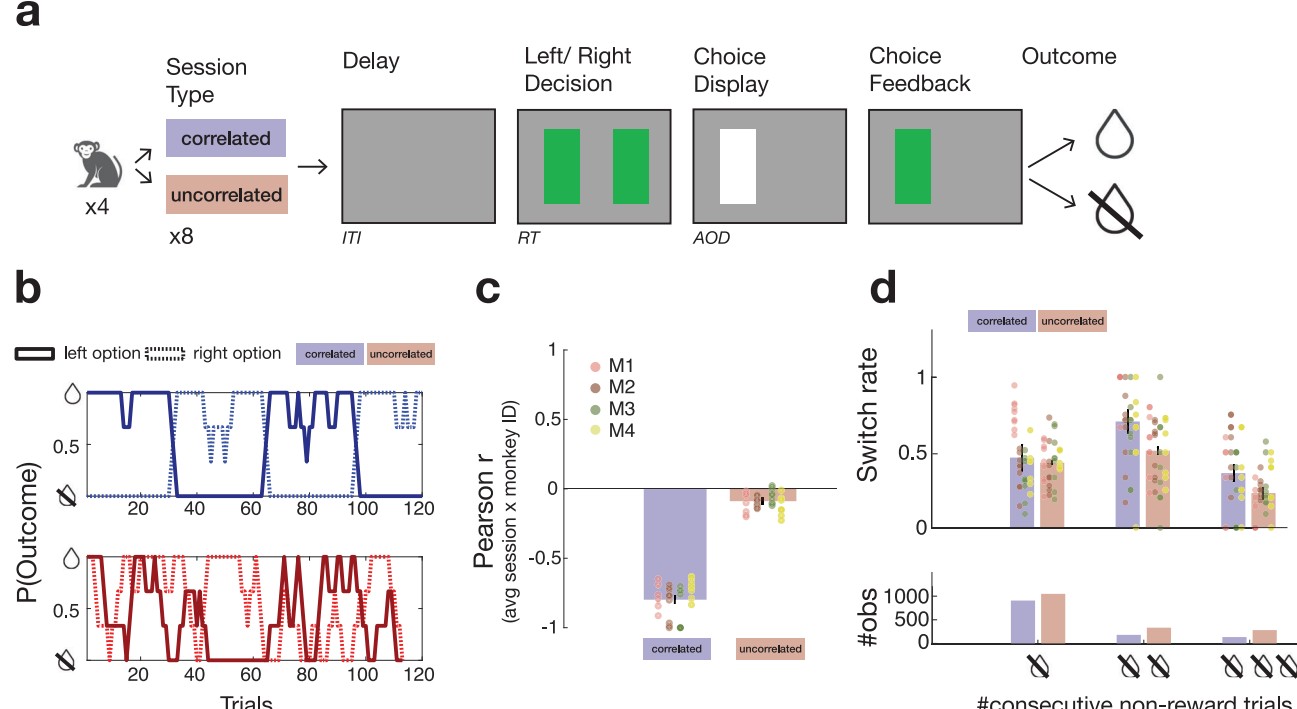

**Fig. 1 | Reversal task. a** Macaques performed a spatial reversal task in which two options were presented as green rectangles on the left and the right of a computer monitor. After the intertrial interval (ITI) macaques saw the two rectangular stimuli. They made their choice by touching a sensor immediately below the option they selected. If the choice was correct, then the rectangle returned to a green colour as juice was delivered; otherwise, it turned black. **b** Probability of reward associated with the left and right-hand options in the (negatively) correlated (blue, top) and uncorrelated (red, bottom) sessions (plot shows a representative example schedule). In general, in correlated sessions, when one option had a high probability of yielding a reward when chosen, the other option did not. In the uncorrelated sessions, however, this was not the case. **c, d** Data from four monkeys (M1-M4) in two schedules, nine (M1–M3) or eight (M4) sessions per schedule. The group mean +/−SEM results are indicated by bar plots with error bars. Data for individual sessions are also included and shown as circles; each vertical column of circles shows the sessions from one individual. **c** In the correlated sessions, the Pearson correlation coefficient between the two options' reward probabilities averaged at around −0.8. It was not exactly −1, but choice probabilities were still significantly more negatively correlated than in the uncorrelated sessions. **d** Across all trials, animals were unlikely to switch from one choice to the other. However, when animals did not receive a reward for a choice, they were likely to switch to choosing the alternative option on the next trial. While switching rates (top) in the correlated (blue) and uncorrelated sessions (red) were similar after one non-reward, switching was significantly more likely in correlated sessions after either two consecutive non-rewards or three consecutive non-rewards. The observations at the top of panel d could only be made on the trials that followed errors (or pairs or triplets of errors) and for clarity the lower part of the figure shows the number of observations (#obs; bottom) used when calculating the switching rate.

for longer despite repeated non-reward. This might happen because there are periods when both options have low probabilities of reward, and even the better option, which the animal should choose, might be followed by non-reward on three successive occasions. Finally, it is also worth noting that in a reasonably large data set such as this one, there are likely to some periods when performance lapses as animals become disengaged from the task for short periods before re-engaging[21].

Ultimately, the task that the animals are performing is a decision-making task in which they must select one choice rather than another. The important decision variable is how much better is one choice than the other. We therefore considered the difference between the probability of reward associated with the chosen option and the unchosen option−on neural activity and the difference between the are uncertainties associated with the estimates of reward probability that the animals might make. Finally, and a key focus for the study, is whether an additional third variable−reversal estimate−also explains variation in behaviour. We developed two Bayesian models to track the features of the correlated and uncorrelated tasks (Methods: Bayesian modelling for correlated environments; Fig. 2a). Both models tracked belief estimates of reward probabilities, and the associated uncertainties of those estimates, and thus belief estimates that macaques might optimally hold given the outcomes (reward/non-reward) observed following their choices (Fig. 2b). The beliefs about the values of each

option were expressed as a probability distribution and updated according to Bayes' theorem, by multiplication of the prior belief at the beginning of each trial and likelihood of the outcome observed at the end of the trial to produce a posterior belief. While these were the only task parameters tracked by model 1, model 2 also tracked the probability of a task state reversal (Fig. 2c, d). Beliefs were quantified by a joint probability density function (pdf) over a pre-defined hypothesis space (Fig. 2c). When it estimated that the animal was transitioning to a different latent task state, it quickly revised the probability and uncertainty estimates of the choices that animal took each time feedback (reward or non-reward) arrived after making a choice. Thus, the reward probability estimates for the choices and the uncertainties on those estimates in model 2 were informed by the reversal estimate. In the correlated environment, options' contingencies are partially negatively correlated with each other such that if one option has a high reward contingency, the other option is likely to have a low reward contingency (Figs. 1a and 2a, b) so that a value of an option−its probability of reward – tends to change in sudden jumps (from high to low or vice versa). In model 2, the subjective estimate of the probability of a reversal influences the estimates of choice value and helps to ensure the sudden changes in choice values that accord with the ground truth of the situation in correlated sessions. In other respects, however, the reward probability and uncertainty estimates furnished by the models were similar in nature (Fig. 2e). A separate but related

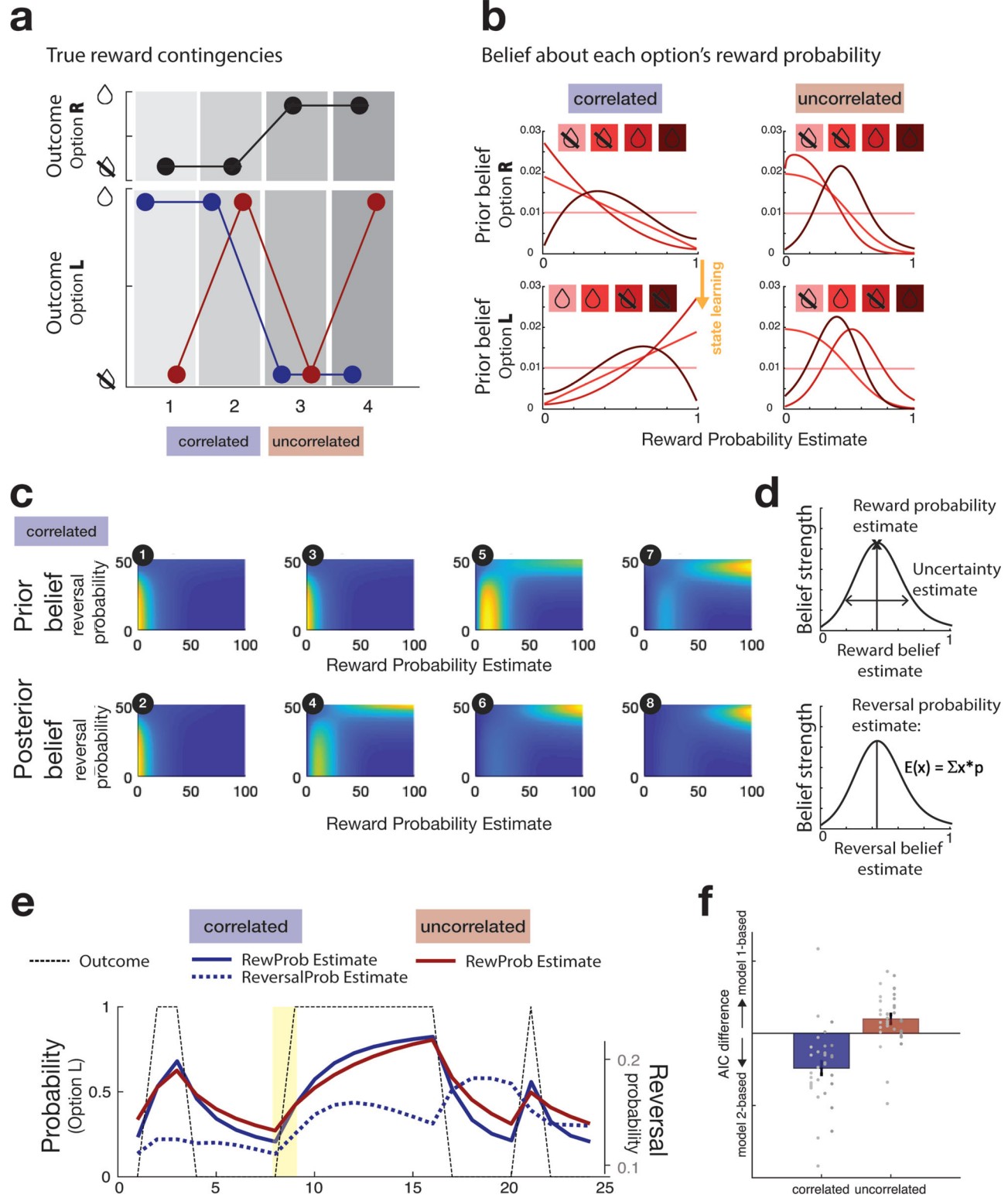

feature that might be incorporated into Model 2 is a capacity to infer an unchosen option's value from feedback obtained from the other, chosen option. We also consider such a capacity for inference below (Fig. S4). Even without inclusion of this extra feature, however, Model 1 provided a better account of the majority of uncorrelated sessions, while model 2 provided a better description of the majority of correlated sessions; the difference in Akaike's information criterion (AIC) scores for Model 2 minus Model 1 are shown for a comparison of the two models for each session (Fig. 2f).

One possibility is that macaques may attempt to infer latent task states even in the uncorrelated task, even if none are present. We can estimate what estimates animals themselves might have about such states and the probability of switching between them— the reversal estimate—in the uncorrelated task by applying model 2 to the uncorrelated task data. By doing this, we are estimating what beliefs the animals might hold at any point in time given the choices that they have made and the feedback— rewards and errors—they have observed even if those beliefs might be erroneous given the nature of the

**Fig. 2 | Bayesian models. a** An example of the true statistical contingencies between options in a correlated and an uncorrelated session. In this example the outcomes for one option, the right choice option (R, upper panel), are held constant while the other option, the left option (L, lower panel) is shown for example correlated (blue) and uncorrelated (red) sessions. **b** Examples of change in belief distribution, presented here as probability density functions, when observing the series of outcomes associated with options R (top) and L (bottom) shown in panel A. The sequence of outcomes comprises rewards (illustrated by juice drops) and non-rewards (illustrated by crossed-out drops). **c** In the correlated session, the monkey should hold beliefs about the options' associated probabilities of reward (here, option R's probability is shown on the x axis) and the probability of a switch between the two latent states in each of which a different option has the higher probability of reward (reversal probability), shown on the *y* axis. The top row shows prior beliefs, the bottom row posterior beliefs for the four example trials.

**d** Reward probability estimates and reward uncertainty estimates correspond to the probability density function's peak and 95% range (top). The reversal estimate corresponded to an uncertainty-weighted estimate of the probability of switching (bottom). **e** Example changes in parameters in models 1 and 2 as an animal experiences rewards or non-rewards for choosing option L (black dotted line), changing over example sessions of either the uncorrelated or correlated task. Reward probability estimates (RewProb Estimate) and Reversal estimates (ReversalProb Estimate). **f** Data from four monkeys (M1-M4) in two schedules, nine (M1-M3) or eight (M4) sessions per schedule. The group mean +/−SEM results are indicated by bar plots with error bars. Data for individual sessions are also included and shown as circles; each vertical column of circles shows the sessions from one individual. Individual grey dots indicate the AIC difference between Model 2 and Model 1 derived from fitting the choice data of each session, and sessions from the four animals are shown in four columns as in Fig. 1c.

reversal task. Because the reversal estimate is a continuous rather than binary variable, it may capture a weak or partial belief that the animal has about a latent state reversal, although if such beliefs are held in the uncorrelated task, the reversal estimate may often be high. We can then examine whether these estimates are predictive of actual switching behaviour or of neural activity even in the uncorrelated task. In this way, we can compare behaviour and neural activity in the two tasks more directly.

For each trial, we quantified five model estimates with a view to assessing whether they predicted macaques' switching behaviour. First, *chosen probability* is the belief about the probability with which a reward will follow the option that is being chosen on the current trial. It is the peak estimate of a probability density function (Fig. 2d, top panel). The *chosen uncertainty* is the model's uncertainty about this estimate of reward probability and is quantified as the 95% range of the probability density function[22]. *Unchosen probability* and *unchosen uncertainty* were similar to chosen probability and chosen uncertainty, but they indexed beliefs about the chances of reward if the option that is unchosen on the current trial were to be taken instead of the option that is chosen. Finally, we quantified an uncertainty-weighted estimate of the expected probability of reversal, which we refer to as the *reversal estimate* (Fig. 2d, lower panel). We show an example of how chosen probability estimates change with repeated choice of the same (left) option over the course of a series of trials when it is followed by reward and non-reward outcomes (Fig. 2e).

The Bayesian model-derived estimates can be used to predict whether animals will switch on the next trial in logistic regressions applied to all animals but separately for the correlated and uncorrelated sessions (Fig. 3, blue: correlated sessions; red: uncorrelated sessions). First, for the sake of simplicity, the chosen probability and unchosen probability were combined into a single index, reward probability difference (chosen probability−unchosen probability), which reflects model 1's estimate of how much more likely the option being chosen was to yield reward than the alternative. Similarly, the uncertainties of the estimates were combined into a single index, uncertainty difference (chosen uncertainty−unchosen uncertainty), which reflects model 1's estimate of how much more uncertain the estimate of reward was for the chosen option rather than the alternative, unchosen option. We used versions of these estimates derived from model 1, again for the sake of simplicity, when comparing the two tasks; it meant that we were able to compare correlated and uncorrelated tasks in terms of explanatory variables that had been quantified in an identical way. The reversal estimate is, however, only furnished by model 2 and so we employed the reversal estimate from model 2. As explained above, this means that we are able to compare the strength of influence the reversal estimate might have on both behaviour and brain activity. Below (Fig. S4), we consider a situation in which choice probabilities and uncertainties in model 2 are informed by the negative correlation between choice values and inferences about an unchosen

option's value are made after feedback for choosing the other option. In summary, we examined the influence of three factors on choice switching, the reward probability difference between the choices, the uncertainty difference between the choices, and the reversal estimate in the two tasks. A logistic mixed-effects model (LMEM1 in "Methods"; using the *lmer4* package in the R environment) with sessions and animals as random intercepts and choice probability difference, choice uncertainty difference, and reversal estimate (the factors illustrated in Fig. 3) as fixed effects with random slopes was used to quantify and compare these influences in correlated and uncorrelated sessions.

The negative intercept (Fig. 3, left) demonstrates that animals were predisposed to repeat choices from one trial to the next in both correlated and uncorrelated sessions. The difference in the probability of reward associated with the option that was chosen and the option that was unchosen (reward probability difference) is a key decision variable on each trial; the negative impact of the reward probability difference between choices (Fig. 3, second from left) indicates that the monkeys were less likely to switch on the next trial if the option that they have just chosen was associated with the higher probability of reward. As has been observed in previous studies[22–24], choice probability difference influenced switch rate ($\chi^2_1 = 94.9724$; $P < 0.001$) but there was no difference in this influence between correlated and uncorrelated sessions ($\chi^2_1 = 0.4329$; $P = 0.5106$). By contrast, the positive impact of the difference in the options' uncertainties (uncertainty difference; Fig. 3, second from right) indicates that monkeys were more likely to switch on the next trial when there was more uncertainty about the probability of reward that was associated with the option that they had chosen on the current trial than about the alternative; choice uncertainty difference exerted a significant influence on switching ($\chi^2_1 = 14.6892$; $P < 0.001$) and this influence was greater in uncorrelated sessions ($\chi_1 = 4.0804$; $P = 0.0434$). Importantly, the animals were more likely to switch when the reversal estimate−model 2's estimate of the probability of reversal between one latent state and another−was high (Fig. 3, right; $\chi^2_1 = 32.1149$; $P < 0.001$) and this effect was greater in correlated sessions ($\chi^2_1 = 24.7756$; $P < 0.001$). Even if animals might seek to identify latent states in the uncorrelated task and transitions between them, our estimates of when they might be doing so, derived from application of model 2 to the uncorrelated task suggests that they have significantly less influence in the uncorrelated task.

Analogous analyses can also be performed with the full set of five factors (chosen probability, unchosen probability, chosen uncertainty, unchosen uncertainty, reversal estimate) using LMEM2 (Supplementary Fig. S2B). These analyses confirmed that the animals' reversal estimates drove their switching behaviour more in correlated sessions even when each option's values and uncertainties were considered separately ($\chi^2_1 = 28.8443$; $P < 0.001$) again confirming that such beliefs had a bigger impact on behaviour in the correlated task. It also demonstrated that, by contrast, in the uncorrelated sessions,

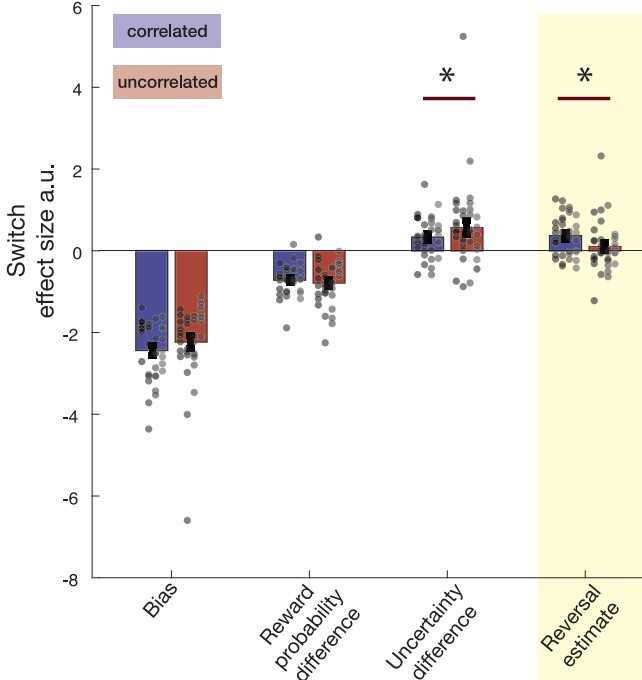

**Fig. 3 | Factors influencing switching in the correlated (blue) and uncorrelated (red) tasks.** Data from four monkeys (M1-M4) in two schedules, nine (M1–M3) or eight (M4) sessions per schedule. The group mean +/−SEM results are indicated by bar plots with error bars. Data for individual sessions are also included and shown as circles; each vertical column of circles shows the sessions from one individual as in Fig. 1c. Logistic regression analyses applied to each session from each animal were used to illustrate whether the reversal estimate (highlighted in the yellow column) influenced switching behaviour more in the correlated than the uncorrelated task. The logistic regression illustrates the influence of the reversal estimate in the context of the influence of other estimates or biases that the animals might hold in terms of regression coefficients for each animal and session. Statistical inference was performed in the hierarchical LMEM1 (*$P < 0.05$). The negative intercept (left: "Bias") demonstrates that animals tended to repeat choices in both correlated and uncorrelated sessions. The difference in the probability of reward associated with the option that was chosen and the option that was unchosen (reward probability difference; second from left) is a key decision variable on each trial; the negative impact of reward probability difference indicates that monkeys were less likely to switch on the next trial if the option that they had just chosen was associated with the higher probability of reward and there was no difference in this influence between correlated and uncorrelated sessions. By contrast, the positive impact of the difference in the options' uncertainties (uncertainty difference; second from right) indicates that monkeys were more likely to switch on the next trial when there was more uncertainty about the probability of reward that was associated with the option that they had chosen on the current trial than about the alternative and that this influence was greater in uncorrelated sessions. Finally, the animals were more likely to switch when the reversal estimate between one latent state and another was high, and this effect was significantly greater in correlated sessions.

macaques were more likely to switch both when their estimate of the unchosen option's probability increased ($\chi^2_1 = 13.6142$; $P < 0.001$) and when the uncertainty of their estimate for the chosen option's value increased ($\chi^2_1 = 7.2583$; $P < 0.007$). In summary, in the uncorrelated sessions, switching was most likely when macaques estimated that the alternative, unchosen option had a higher probability of reward and when they were uncertain about their estimate of the probability of reward for the option they had been choosing. By contrast, in the correlated sessions, macaques were especially likely to switch when their estimate of reversal between task states was high. We, therefore, next examined how neural activity was associated with these Bayesian model-derived variables that predicted switch behaviour, particularly

focusing on how these patterns differed between correlated and uncorrelated sessions.

## Activity in the dmPFC and hippocampus tracks the probability of reversal in the correlated task

We first focused on the neural activity related to the reversal estimate using GLM1 ("Methods"). GLM1 included the same variables that were predictive of switching in the behaviour analysis (Fig. 3), which were all time-locked to action selection. The reversal estimate is updated once the animal observes the outcome of each trial, and the degree to which the reversal estimate is updated between one trial and the next can be quantified by the Kullback-Leibler divergence (reversal $D_{KL}$). We therefore included an additional variable, reversal $D_{KL}$, time-locked to the outcome(reward/non-reward) to capture neural activity covarying with the change in the reversal estimate. We tested for changes in blood oxygen level-dependent (BOLD) activity in relation to reversal estimate and reversal $D_{KL}$ across the whole brain using cluster-based correction ($Z > 2.3$; $P < 0.05$), separately for correlated and uncorrelated tasks, and for the BOLD activity that differed between both tasks. Activity in dmPFC and hippocampus increased with reversal estimate (Fig. 4a, c). The cluster in dmPFC was significant for correlated sessions, while the hippocampus was identified for the difference between correlated and uncorrelated sessions. Additional activity was identified in more anterior, but overlapping, dmPFC (Fig. 4b) that was associated with reversal $D_{KL}$ in correlated sessions only. Both reversal-related and reversal $D_{KL}$-related activity in dmPFC was most prominent in the medial superior frontal gyrus but extended, ventrally, into the adjacent cingulate sulcus and laterally onto the dorsal convexity as far as the principal sulcus.

The whole-brain cluster-based approach that identified the reversal estimate-related activity in dmPFC (Fig. 4a) did so in data from correlated sessions. By contrast, the reversal estimate result in the hippocampus (Fig. 4c) reflected comparison of correlated and uncorrelated sessions. One potential interpretation of such a pattern is that there is some fundamental difference in terms of how selective the dmPFC and hippocampus are for correlated as opposed to uncorrelated tasks. However, an alternative interpretation is that activity in all the areas is broadly similar and, in general, it is most prominent in correlated sessions. We can arbitrate between these two accounts by testing whether activity related to the two tasks differed from one another across the three areas (Fig. 4a–c) using an ANOVA ("Methods") with factors of brain area and condition as well as session. There was, however, no interaction between task and brain area ($F_{2, 156} = 0.329$; $P > 0.05$). This suggests that overall, the areas—dmPFC and hippocampus—carried similar information about the potential for a reversal of task states in the correlated task but not in the uncorrelated task.

If the animals search for latent task states and reversals between them in the uncorrelated task, then by applying model 2 to the uncorrelated task, it is possible to estimate what estimates the animals themselves might have about reversal, even if these are weak and partial. We were, however, unable to find significant evidence for activity linked to reversal estimates in the uncorrelated task when we examined activity across the whole brain. This was not simply a consequence of the conservative nature of the whole-brain analysis approach, because ROI-based analyses similarly revealed no evidence for reversal effects in the uncorrelated task in dmPFC or hippocampus (all $r < 0.406$; all $P > 0.3$); even though the hippocampus and dmPFC carry reversal estimate-related activity in the correlated task, they do not do so in the uncorrelated task. Finally, we note that, as in the behavioural analyses, when examining the neural data, we took two complementary approaches. In the first, we employed versions of the reward probability difference and uncertainty difference that were derived from the application of model 1 to both correlated and uncorrelated sessions (GLM1). The second approach (GLM1A) was identical but here we used models 1 and 2 for analysing data from the

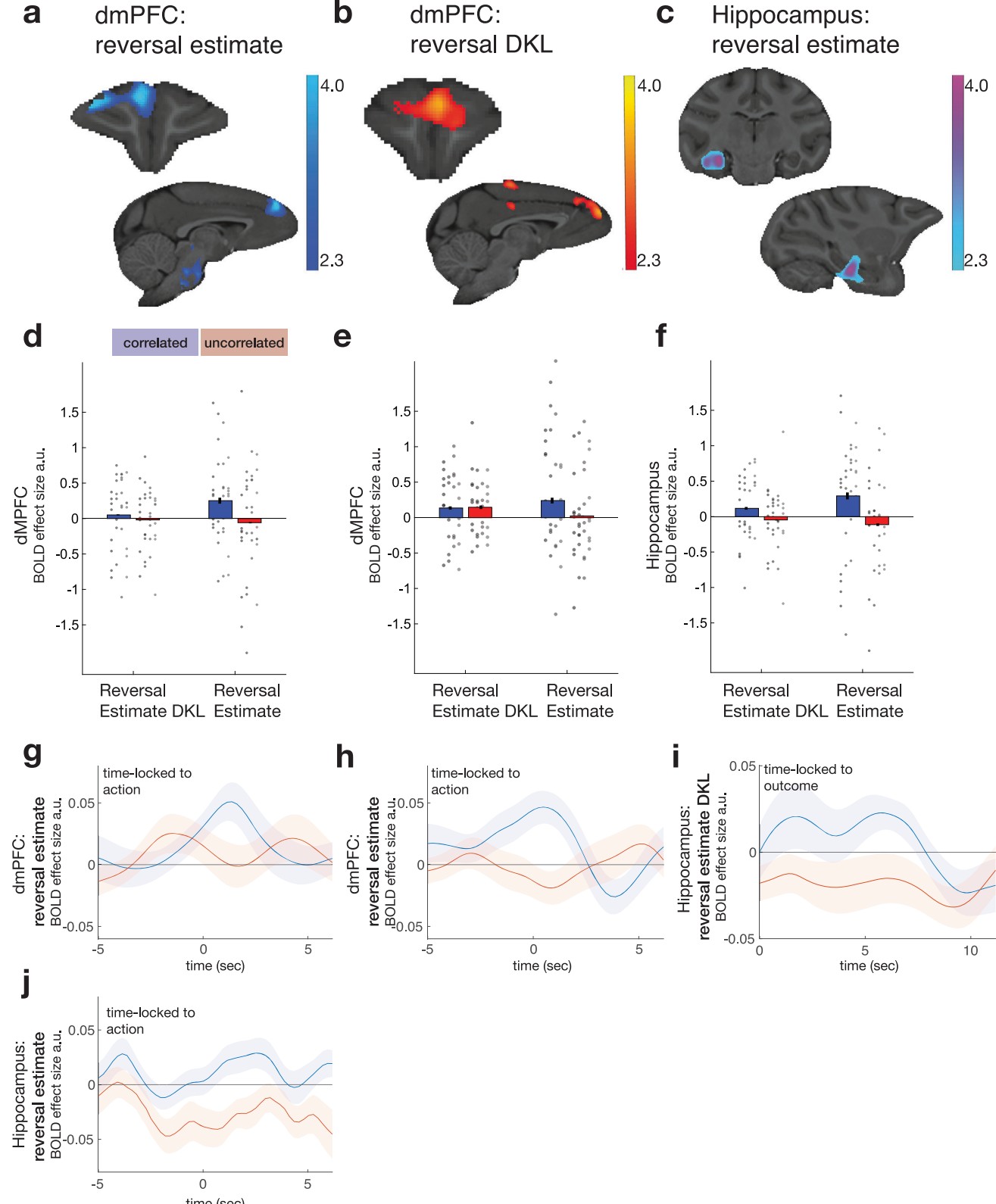

similar effects of reversal across all three areas, we did not find evidence for similar reversal $D_{KL}$ activity in all three areas. Reversal $D_{KL}$ only explained significant variance in activity in the anterior dmPFC (Fig. 4b, e), although there was a numerically similar but only marginally significant effect in the hippocampus ($t_{34} = 1.560$, $P = 0.064$). Moreover, further inspection of the data revealed that the difference between reversal $D_{KL}$ activity in correlated and uncorrelated sessions increased with time after the outcome (Fig. 4i). It peaked after the

uncorrelated and correlated sessions, respectively. The only difference in the results yielded by these two approaches was that the reversal estimate-related activity in dmPFC (Fig. 4a) survived whole-brain cluster-based significance when the second approach was used.

We also tested for reversal $D_{KL}$-linked activity at the time of outcome in all three areas using an analogous approach to the one that we had taken when examining reversal-related activity at the time of decision-making in the three regions. However, unlike the generally

**Fig. 4 | Neural activity related to the probability of a change in latent state.** Whole-brain cluster corrected ($z > 2.3$; $P < 0.05$) analyses (local analysis of mixed-effects) revealed activity significantly related to the estimate of the probability of a reversal between latent states—reversal estimate—at the time of choice selection in the correlated task in dmPFC (**a**), related to the reversal update (reversal $D_{KL}$) at the time of the outcome at the end of the trial in both tasks (**b**), and to the difference between reversal-related activity in correlated and uncorrelated sessions in hippocampus (**c**). Subsequent panels (**d**–**f**) beneath (**a**–**c**) further illustrate activity extracted from the locations in (**a**–**c**), respectively, in the correlated sessions (blue) and uncorrelated sessions (red). Data from four monkeys (M1-M4) in two schedules, nine (M1–M3) or eight (M4) sessions per schedule. The group mean +/−SEM results are indicated by bar plots with error bars. Data for individual sessions are also included and shown as circles; each vertical column of circles shows the sessions from one individual as in Fig. 1c. **a**, **b** Uncorrelated sessions do not show

activity related to reversal estimate (red bars) like that seen in correlated sessions (blue bars, shown purely for illustration and comparison). **c** Reversal $D_{KL}$ update-related activity at the time of choice outcome was significant in correlated sessions in anterior dmPFC, although there was a marginal effect in uncorrelated sessions (**e**). However, in the hippocampus, the reversal $D_{KL}$ effect was only significant in correlated sessions (**f**). **g**–**j** The group mean +/−SEM results are indicated by lines with shades. Reversal estimate-related effects in the correlated sessions (blue) occurred at or before the time of decision-making in dmPFC (**g**, **h**). In the hippocampus, updates in the reversal estimate—reversal $D_{KL}$—had a prominent effect on activity after each trial outcome was witnessed (**i**). **j** A PPI analysis confirmed dmPFC and hippocampal activity coupling as a function of the reversal estimate shortly after the choice was made in each trial, and that coupling was more prominent in correlated than uncorrelated sessions.

period captured by the initial analysis (Fig. 4f), consistent with a sustained change after the outcome up until the start of the next trial.

While any reversal $D_{KL}$-related activity in the hippocampus was restricted to the correlated task, intriguingly, $D_{KL}$ activity in anterior dmPFC was statistically greater than zero even in uncorrelated sessions (Fig. 4e; $t_{34} = 2.315$; $P < 0.013$). It is possible that this may reflect animals attempting to identify latent task states and transitions between them even in the uncorrelated task. We return to the question of $D_{KL}$-related activity in dmPFC below to show that, in the uncorrelated task, the animals may possibly do this at the time of feedback, but any attempt to do so does not result in sustained neural activity at the onset of the next trial. The activity may actually reflect tracking features of individual options as well as latent state reversal (discussed further below in Fig. 5).

Examining the temporal evolution of the reversal estimate effects ("Methods" and Fig. 4g, h) confirmed that in dmPFC, the BOLD signal peaked shortly after animals made their choice on each trial. The relatively fast haemodynamic BOLD response observed in macaques has a 2–4 s lag and so, once that is taken into account, the timing of dmPFC activity changes (Fig. 4g, h) suggests dmPFC reflected a reversal estimate around the time animals were deciding which option to pick. By contrast, as noted, the reversal $D_{KL}$ effect in the hippocampus was time-locked to the outcome (Fig. 4i). A similarly long sustained reversal $D_{KL}$ effect was observed in dmPFC (Fig. S3).

A direct anatomical connection between the subiculum of the hippocampal formation and anterior cingulate regions in or near dmPFC[25] might provide a route for dmPFC-hippocampus interactions during estimation and maintenance of reversal probability. We, therefore, performed a psychophysiological interaction (PPI)[26] analysis to examine the influence of dmPFC on the hippocampus as a function of the reversal estimate. The analysis showed that hippocampal and dmPFC activity coupling was, indeed, proportional to the reversal estimate (Fig. 4j). When we identified the peak positive effects in correlated and uncorrelated sessions after choice, using a leave-one-out (LOO) procedure, it was significantly greater in correlated than uncorrelated sessions (Wilcoxon test: $z = 2.03$, $P = 0.043$). Such a pattern suggests dmPFC and hippocampus interact with one another after each choice but that they continue to do so as the outcomes for the choices are received and as the reversal estimate that will be used on the next trial is established.

Activity in dmPFC and hippocampus reflected updates in reversal estimates—reversal $D_{KL}$. It is possible that this may reflect the animals trying to use feedback to infer a latent task or the possibility of a transition between latent task states. If this is the case, then it does not seem to result in sustained neural activity related to the reversal estimate by the beginning of the next trial (note the absence of positive red bars on the right-hand side of Fig. 5d–f). However, it is possible that this might simply be because dmPFC and hippocampus were actually tracking some other update-related variable that resembled reversal

$D_{KL}$. We therefore employed new versions of models 1 and 2 that yielded separate update estimates for each option. For models 1 and 2, this was under the assumption that options' reward probabilities were uncorrelated or negatively correlated, respectively ("Methods", "Model Estimates"). In other words, the models yielded estimates of how much an individual option's associated reward probability was to be updated when the outcome arrived after it was chosen as opposed to an estimate of whether there was likely to be a transition between latent states. We re-ran the fMRI analyses in the anterior dmPFC and hippocampus ROIs shown in Fig. 4, but now included both reversal $D_{KL}$ and individual option-specific update term (as appropriate for correlated and uncorrelated sessions) and allowed them to compete to explain activity. Two important results emerged. First, reversal $D_{KL}$ continued to explain significant variance in outcome-locked activity in correlated sessions in both dmPFC and hippocampus (Fig. 5, left, dmPFC: $t_{34} = 2.003$, $P = 0.027$; right, hippocampus: $t_{34} = 1.988$, $P = 0.027$). Second, the picture was very different in uncorrelated sessions in the hippocampus (Fig. 5b, right); the hippocampus showed no evidence of tracking option-specific updates in the same way as reversal $D_{KL}$. In anterior dmPFC in the uncorrelated task, reversal $D_{KL}$ and option-specific update both explained less variance than they had in correlated sessions and in neither case did the effects reach significance although they were marginal in one case (reversal $D_{KL}$: $t_{34} = 1.687$, $P = 0.051$). In summary, dmPFC tracked reversal $D_{KL}$ updates but did so most clearly when the task contained latent structure, as was the case in the correlated task. Anterior dmPFC may possibly have estimated, or attempted to estimate, reversal $D_{KL}$ even in uncorrelated sessions, perhaps as a consequence of the experience of its utility in the correlated sessions which the same individuals performed. Instead, in the uncorrelated task, dmPFC activity, if anything, reflected the updating of values of the individual options. However, by contrast, it was very clear that the hippocampus was only concerned with tracking of reversal estimates for latent task states, and not option-specific updates, and that it did so only in the correlated task where there was latent structure.

## Activity in dmPFC and medial and anterior thalamus tracks the decision variable at the time of choice

The next part of the analysis (implemented simultaneously in GLM1) was aimed at identifying activity related not to the reversal estimate but more directly to making the decision to select one option rather than the other. The difference in the reward probabilities for one option and the other determines their difference in value and is the key variable that should guide decision-making. Previous studies have shown that BOLD activity tracks the difference in value between the option selected and the option rejected in areas identified as essential for decision-making[17,27]. Such activity reflects the two options at the point in time when they are being compared but, immediately before they are compared, activity may reflect both options' values with the

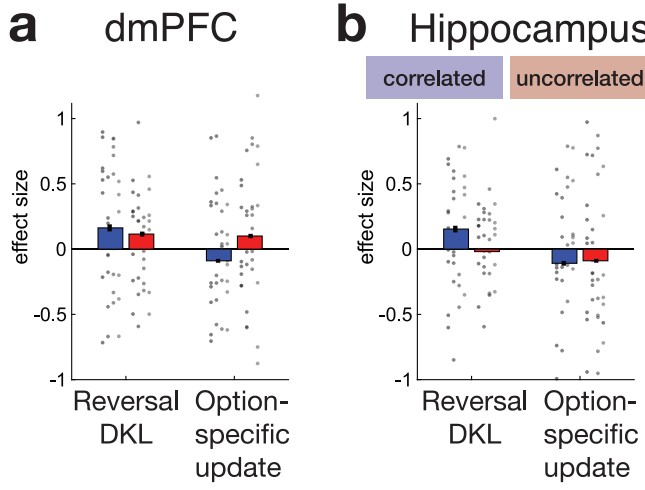

**Fig. 5 | Update signals in dmPFC and hippocampus signal changes in the latent state.** We assessed whether dmPFC and hippocampal activity still significantly reflected reversal $D_{KL}$ when the analysis included an additional option-specific update term. The option-specific update term was estimated either under the assumption that the options' reward probabilities were uncorrelated or negatively correlated, as was appropriate for uncorrelated and correlated sessions, respectively. However, both anterior dmPFC (**a**, blue bar on left; same ROI as illustrated in Fig. 4b) and hippocampus (**b**, blue bar on left; same ROI as illustrated in Fig. 4c) continued to show evidence that they tracked reversal $D_{KL}$ in correlated sessions even when the option-specific update term was also included in the analysis suggesting that the evidence that they tracked reversal $D_{KL}$ could not be explained away by them tracking option-specific updates instead or as well. In fact, there was no evidence that either dmPFC or hippocampus activity increased in proportion to the degree of option-specific update that occurred after an outcome. In uncorrelated sessions, the picture was different in the hippocampus (**b**, red); the hippocampus showed no evidence of tracking option-specific updates in the same way as reversal $D_{KL}$. In anterior dmPFC, in the uncorrelated task (**a**, red), reversal DKL and option-specific update both explained less variance than they had in correlated sessions and in neither case did the effects reach significance although they were marginal in the case of reversal $D_{KL}$. Anterior dmPFC may possibly have estimated, or attempted to estimate, reversal $D_{KL}$ even in uncorrelated sessions, perhaps as a consequence of the experience of its utility in the correlated sessions. However, the hippocampus was only concerned with tracking of reversal estimates for latent task states, and not option-specific updates, and that it did so only in the correlated task where there was latent structure. Data from four monkeys (M1-M4) in two schedules, nine (M1-M3) or eight (M4) sessions per schedule. The group mean +/-SEM results are indicated by bar plots with error bars. Data for individual sessions are also included and shown as circles; each vertical column of circles shows the sessions from one individual as in Fig. 1c.

same sign (and therefore reflect the sum of their values)[28]. After the option values are compared, the value of the option that is chosen, which is now the focus of behaviour, may be the only option represented in neural activity[29]. We therefore sought brain regions in which activity was related to the difference in value between the choices taken and rejected on each trial which, in the current experiment, corresponds to the difference in reward probability (Chosen Probability-Unchosen Probability). We tested for neural activity tracking Chosen Probability-Unchosen Probability to a greater degree in the correlated versus the uncorrelated task.

GLM1 identified (whole-brain analysis using cluster-based correction; $z > 2.3$; $P < 0.05$) activity in two regions. There was a significant effect of the Chosen-Probability-Unchosen Probability comparison in a dmPFC region (Fig. 6a) partly overlapping with, and ventral to, the dmPFC region linked to the reversal estimate. There was also a region in the anterior and medial thalamus, including the dorsomedial and anterior nuclei, in which activity was significantly better explained by the Chosen Probability-Unchosen Probability contrast in the correlated sessions than the uncorrelated sessions (Fig. 6b). Time-course

analyses (Fig. 6e, f) suggested that activity in both dmPFC and thalamus was positively related to Chosen Probability and negatively related to Unchosen Probability prior to choice selection (Fig. 6e, f). As the decision unfolds, activity in dmPFC and thalamus is increasingly dominated by the value of the choice taken and less influenced by the value of the choice rejected[29].

Our central interest in this study was to look at whether animals might infer a latent state and, if so, how they did this. The same piece of information—the feedback on each trial— informs the animal about both its estimate of the value of the choice it is taking and the reversal estimate. Given that they infer a reversal in latent state, there are two ways that they might then establish a value for the unchosen option. First, having inferred that the state has changed, they might move into an exploratory mode and try out the other response and learn by observation if the other choice, choice 2, is now rewarded. Alternatively, they might make a second type of inference; in addition to inferring the change in latent state, they might infer a precise value about the value of the unchosen option on the basis that the values of the two choices were negatively correlated with one another; if they observed that the chosen value decreased, they might infer that the value of the unchosen option increased. We therefore used a second version of Model 2 that incorporated the assumption that this second type of inference about the unchosen option might be made in the analyses shown in Supplementary Fig. S4. Anterior and medial thalamus and dmPFC activity reflect information about choice values in the correlated sessions even under this alternative assumption.

The dmPFC is reciprocally and monosynaptically interconnected with mediodorsal and anterior nuclei of the thalamus[30–32] and, in addition, hippocampal projections arrive in the anterior medial and anterior dorsal thalamus and, to a more limited degree, in mediodorsal thalamus[33–35]. The region of thalamic activation spanning these nuclei is, therefore, another potential site for interaction between activity encoding the reversal estimate in dmPFC and hippocampus and activity encoding the decision variable—the difference in reward probabilities for the two options. We therefore used a PPI analysis to examine the influence of dmPFC activity on anterior medial thalamus as a function of reversal estimate, the reward probability associated with the chosen option, and the interaction between these two variables (Fig. 6g); when we used the LOO cross-validation procedure to identify peak effects, there was a strong difference in dmPFC-thalamus interactions in correlated as opposed to uncorrelated sessions as a function of the interaction between the reversal estimate and the chosen option's probability of reward (Wilcoxon test: $z = 3.65$, $P < 0.001$). DmPFC-thalamic interactions may reflect integration of information about how likely the task is to be switching between latent states or to be stable and how good the option finally chosen will be as a consequence.

## Disruption of dmPFC, hippocampus, or thalamus impairs tracking of reversal estimates

So far, analysis of neural activity has suggested that dmPFC and hippocampus track the estimate of whether there will be a reversal between the two states of the correlated task, and anterior and mediodorsal thalamus interactions with dmPFC reflect the interplay between the reversal estimate and choice values. If these activity patterns are essential for tracking reversal estimates and the influence such estimates exert on choice selection, then their disruption should diminish the influence that the reversal estimate has on switching behaviour (right-hand bar in Fig. 3 should be reduced).

To test this possibility, in experiment 2, we trained three new macaques to perform the correlated task and examined the impact of disrupting activity in dmPFC, hippocampus, and anterior and dorsomedial thalamus with transcranial ultrasound stimulation (TUS). We used a similar procedure to that used in the past where a short 40-s train of TUS disrupts subsequent neural activity and behaviour for a

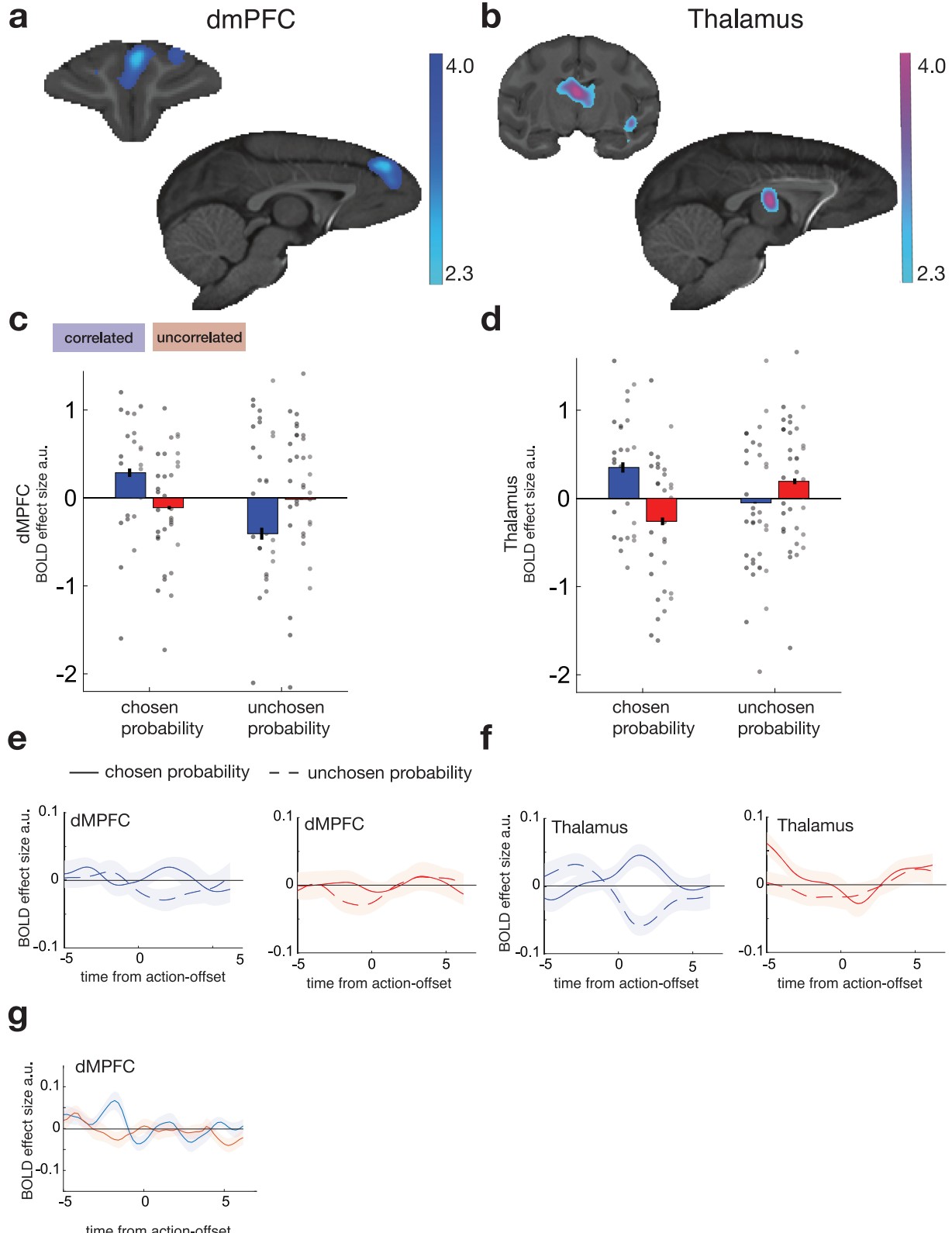

sustained period; effects gradually decline over a period of hours and are not discernible on the following day[11-19]. The approach of delivering TUS prior to task performance and then examining its subsequent offline effects precludes the possibility of the TUS simply distracting animals via a peripheral effect or auditory confound that occurs only at the time of stimulation. Previous studies using this approach have found no evidence of any change in auditory brain areas that outlast the stimulation period. TUS to each area was compared with a sham

procedure in which animals were prepared for TUS but once the TUS transducer was placed on the head, no stimulation was delivered. DmPFC, hippocampus, thalamus, and sham TUS were tested on interleaved days (TUS was applied to dmPFC, hippocampus, thalamus, and as a sham on 6, 6, 6, and 8 days, respectively; an additional sham day was scheduled at the start and end of the testing procedure).

The same pattern of behaviour as in the correlated task in experiment 1 was also observed in the sham condition in experiment 2;

**Fig. 6 | Decision-related activity.** Whole-brain cluster-corrected ($z > 2.3$; $P < 0.05$) analyses (Local Analysis of Mixed Effects) revealed activity significantly related to chosen-unchosen probability—the difference in probability of reward associated with the option that the macaques took (chosen probability) and rejected (unchosen probability) at the time of choice selection in dmPFC in correlated sessions (**a**). There was also activity related to the difference between chosen-unchosen probability in correlated versus uncorrelated sessions in a medial and anterior thalamic region spanning mediodorsal and anterior thalamic nuclei (**b**). Subsequent panels (**c**, **d**) beneath (**a**, **b**) further illustrate activity extracted from the locations in (**a**, **b**), respectively, in the correlated sessions (blue) and uncorrelated sessions (red). Data from four monkeys (M1-M4) in two schedules, nine (M1–M3) or eight (M4) sessions per schedule. The group mean +/−SEM results are indicated by bar plots with error bars. Data for individual sessions are also included and shown as circles; each vertical column of circles shows the sessions from one individual as in Fig. 1c. **e** Time course analysis revealed that activity related to the probability of reward associated with the chosen action (chosen probability) began to diverge from activity related to the unchosen action (unchosen probability) around the time that animals selected actions in both dmPFC (**e**, blue) and in medial and anterior thalamus (**f**, blue) in correlated sessions but not uncorrelated sessions (red). Note that, given that there is a 2–4 s haemodynamic delay in macaques, neural activity begins to diverge at around the time of choice deliberation. **g** PPI analyses demonstrated that dmPFC-thalamus activity coupling was modulated by the interaction of the reversal estimate and the chosen option's probability of reward, suggesting that dmPFC-thalamus interactions encode how good the action will be given the likelihood of a change in latent task state.

higher reversal estimates were associated with increased choice switching (Fig. 7a). However, LMEM analyses (LMEM3 and LMEM4 in Methods which employed similar factors to LMEM1 and LMEM2, respectively, with sessions and animals as random intercepts and the interaction between TUS (sham, dmPFC, hippocampus, thalamus) and chosen-unchosen option probability difference, chosen-unchosen uncertainty difference; and reversal estimate as fixed effects, together with their respective random slopes) revealed that the influence of the reversal estimate on switching was significantly different in the four conditions ($\chi^2_3 = 57.4757$; $P < 0.001$). It was reduced following dmPFC ($\beta_{sham-dmPFC} = -0.3436$; SE = 0.0798; $z = -4.305$; $P < 0.001$; Fig. 7b), hippocampus ($\beta_{sham-hippocampus} = -0.4155$; SE = 0.0814; $z = -5.104$; $P < 0.001$; Fig. 7c), and anterior/dorsomedial thalamic ($\beta_{sham-thalamus} = -0.6010$; SE = 0.0838; $z = -7.174$; $P < 0.001$; Fig. 7d) TUS. As a consequence of the changes to the way that the task was performed, TUS led to a decrease in the rate at which rewards were received (main effect of TUS: $F_{3, 6} = 9.222$; $P = 0.012$; sham versus dmPFC: $F_{1,2} = 36.241$; $P = 0.027$; sham versus hippocampus: $F_{1,2} = 42.912$; $P = 0.023$; sham versus thalamus: $F_{1,2} = 24.904$; $P = 0.038$).

In order to understand how if the impact of TUS changed over the course of the session, we re-ran the analyses examining the differences in the reversal effect between sham and each of the three TUS targets but now with an additional factor time in the session (indexed by trial number). We found no differences between sham and each of the three TUS conditions when we looked at the interaction between the reversal estimate and time (all $z > -1.821$; all $P > 0.069$—the area with a possible trend effect was the anterior dorsomedial thalamus) and, in the same analyses, the reversal estimate continued to differ between each of the three TUS conditions and sham (all $z < -3.919$; all $P < 0.001$).

In experiment 1, comparison of the correlated and uncorrelated sessions had suggested that while switching was driven by the animals' reversal estimates in correlated sessions, it was more likely to be driven by uncertainty about the value of the choice that had recently been chosen in uncorrelated sessions (Supplementary Fig. S2A). We therefore examined whether this pattern—relatively greater influence of uncertainty about recently chosen option value compared to reversal estimates—emerged when TUS was applied in the correlated task (LMEM4; Supplementary Fig. S5). This was indeed the case; after anterior/dorsomedial thalamic TUS ($\beta_{sham-thalamus} = 0.2309$; SE = 0.1011; $z = 2.283$; $P < 0.022$; fourth bars in supplementary Fig. S5A, D) but not when TUS was applied elsewhere. By contrast, dmPFC TUS led to a different pattern of impairment where animals' switching was determined only by their estimate of the reward probability of the choice that they had been taking recently ($\beta_{sham-dmPFC} = -0.3693$; SE = 0.0847; $z = -4.358$; $P < 0.001$); as this declined they were more likely to switch but other factors such as the reversal estimate and the uncertainty of value estimates for both options now exerted no significant influence on their choices (unchosen value: $\chi^2_1 = 0.5168$; $P > 0.4722$; chosen uncertainty: $\chi^2_1 = 0.9796$; $P > 0.3223$; unchosen uncertainty: $\chi^2_1 = 0.6362$; $P > 0.4251$; reversal estimate: $\chi^2_1 = 0.0066$; $P > 0.9351$).

Finally, we examined whether the impact of TUS in each area did not just differ from sham but that the pattern of TUS effects for each area differed from those seen after TUS to each other area. We used an ANOVA to examine the pattern of influences on switching of the predictors relating to the value of the chosen option, the value of the alternative option, the uncertainty of the chosen option, and the uncertainty of the unchosen option, and the reversal (Supplementary Fig. S5) in pairs of areas. We found a significant interaction between TUS target and predictors influencing switching when we compared dmPFC and thalamus ($F_{4, 8} = 9.325$; $P = 0.004$) and when we compared dmPFC and hippocampus ($F_{4, 8} = 4.341$; $P = 0.037$). While we did not find evidence for the same interaction when we compared hippocampus and thalamus TUS ($F_{4, 8} = 1.71$; $P = 0.240$) we did find evidence that the influence of the reversal estimate on switching differed significantly between thalamus TUS and hippocampus TUS ($z = 2.322$; $P = 0.020$) as indeed it also did between thalamus TUS and dmPFC TUS ($z = 2.684$; $P = 0.007$).

In summary, after dmPFC TUS animals had little sense of task structure or choice uncertainty and relied simply on how rewarding the option they were currently taking had recently been in order to determine whether to stick with it or whether it was time to switch to the alternative option. After hippocampus TUS, animals' switching behaviour in the correlated task was similar to behaviour in the uncorrelated task; it was no longer influenced by the reversal estimate. In the anterior medial thalamus, activity and interactions with dmPFC reflected integration of reversal estimates with choice evaluation. TUS here led to the reversal estimate now having an opposite influence over switching to the one that it had normally so that the higher reversal estimates hindered rather than promoted switching.

## Discussion

Previous studies have recorded activity in OFC and in adjacent regions, such as area 47/12o on the border between the OFC proper and the ventrolateral prefrontal cortex, when animals are learning how likely choices are to lead to rewards[20,36,37]. OFC activity also reflects choice-reward probability and other aspects of choice value and choice comparison during decision-making[29,38–40]. While lesions or manipulations of OFC lead to alterations in value-guided decision-making[3,8,40], disruption of 47/12o causes failures of credit assignment—a diminished ability to learn which choices caused which outcomes[8,15]. The deficit makes it especially difficult to establish which one of several choices leads to a reward when multiple choices are interleaved with one another because the credit for any reward is assigned not just to the choice that actually led to it but to other choices made at adjacent points in time[41,42]. Area 47/12o disruption, therefore, makes it difficult for animals to perform reversal tasks because at the point of the reversal animals should switch from one choice to another as the choice-reward associations change. It is at precisely such points, when choices of different types are interleaved with one another, that 47/12o disruptions lead animals to struggle to assign rewards to the choice that actually caused them, as opposed to other choices that occurred

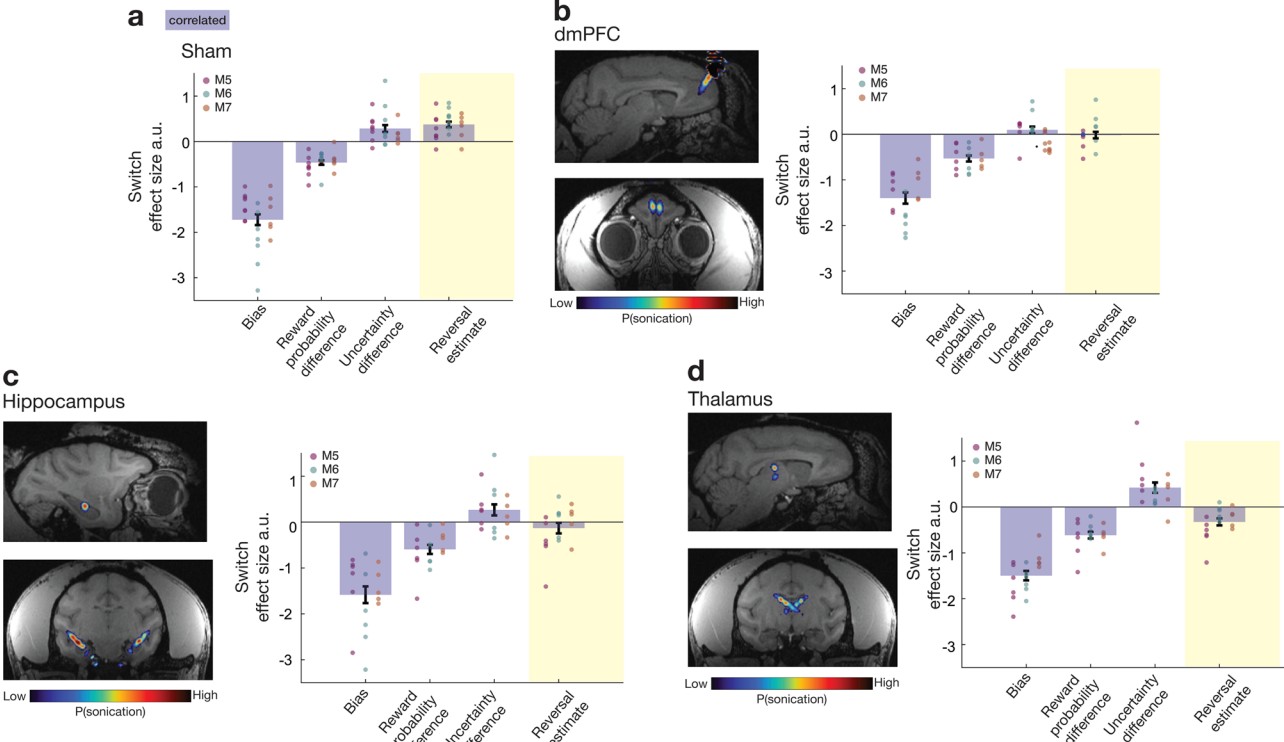

**Fig. 7 | TUS diminished the influence of the reversal estimate on switching.** Data from three monkeys (M5-M7) in four conditions, eight (sham) or 6 (TUS × 3 ROIs) sessions per condition. The group mean +/−SEM results are indicated by bar plots with error bars. Data for individual sessions are also included and shown as circles; each vertical column of circles shows the sessions from one individual. Logistic regression analyses applied to each session from each animal were used to illustrate whether the reversal estimate (highlighted in yellow column) changed between sham (**a**) and after disruption of dmPFC (**b**), hippocampus (**c**), or thalamus (**d**). Insets in each panel illustrate estimates of the ultrasound relative field intensities obtained using the acoustic simulation toolbox (k-Plan, Brainbox, UK) and in every case these are maximal in the grey matter region targeted rather than the white matter. The impact of the same factors as in Fig. 3, difference in reward probability associated with the two options (reward probability difference), the difference in the uncertainties of the estimates of reward probability for each choice (uncertainty difference), alongside the influence of the reversal estimate, are illustrated in terms of logistic regression coefficients from each animal and session. Statistical inference was performed in the hierarchical LMEM3. As in experiment 1, reversal estimate influenced switching in the correlated task in the sham condition in experiment 2 (Fig. 3). The influence of the reversal estimate was, however, significantly decreased after dmPFC (**b**), hippocampus (**c**), and thalamus (**d**) TUS. Further analysis revealed differences in the ways in which TUS to each area affected performance (Supplementary Fig. S5).

close in time but before or after. This, in turn, hampers the animal's ability to switch effectively as the reward contingencies change. However, such impairments are especially apparent in stochastic and constantly changing environments in which there are many possible choices.

Other mechanisms, however, may be in play during reversal tasks that have sometimes been thought to be simpler because the animals reverse between a limited number of discrete states[5–7], usually two, in each of which the better choice is different. If this view is correct, then the capacity to track the different states that are latent within the task and the possibility of transitioning between one state and another may be what is compromised by prefrontal lesions in reversal tasks. In line with this view, here we show that macaques do indeed track the likelihood of switching from one latent state to another (Fig. 3). Moreover, activity in dmPFC and hippocampus, and the interactions between them, reflected the estimate of the probability of transitioning between states (Fig. 4). From dmPFC, activity tracking reversal estimates extended laterally and medially, respectively, to adjacent dorsolateral prefrontal cortex and cingulate sulcus. The activity was only apparent or stronger in the correlated task. Both this task and the uncorrelated task required macaques to track choice values as they changed over time, but only the correlated task contained two latent states in which one or the other choice was the better one. That it is a distributed neural circuit comprising both hippocampus and frontal cortex that tracks transitions between latent states, or contexts, and

translates context information into choice values is consistent with an emerging view of the importance of hippocampal-frontal interactions in primates in making context-appropriate choices[4,43–47]. Both the current results and previous work by Bernardi and colleagues[4] suggest activity in medial and dorsal frontal cortex encodes context or state information at the time that it is used to guide action selection, but that context-related information in the hippocampus is more prominent before and afterwards.

Aspiration, but not excitotoxic, lesions of OFC disrupt reversal task performance[3,48], suggesting that the well-known association between OFC lesions and reversal deficits in primates might actually be the consequence of damage to white matter subjacent to OFC. This suggestion is supported by the observation that even subtotal OFC lesions comprising only a thin strip of aspirated tissue at the posterior boundary of OFC are sufficient to cause a reversal deficit. The current results suggest that such white matter lesions may compromise reversal task performance because they disrupt the coordination of activity between dmPFC and hippocampus. In addition to the cingulum bundle, projections from the hippocampus and adjacent and interconnected medial temporal lobe regions take a second route coursing through white matter subjacent to OFC, and a portion of these projections extends to dmPFC[49–51]. The importance of dmPFC activity in the correlated task may partly reflect the presence of spatially defined action choices in the correlated task. OFC may be less critical when animals make choices between actions as opposed to

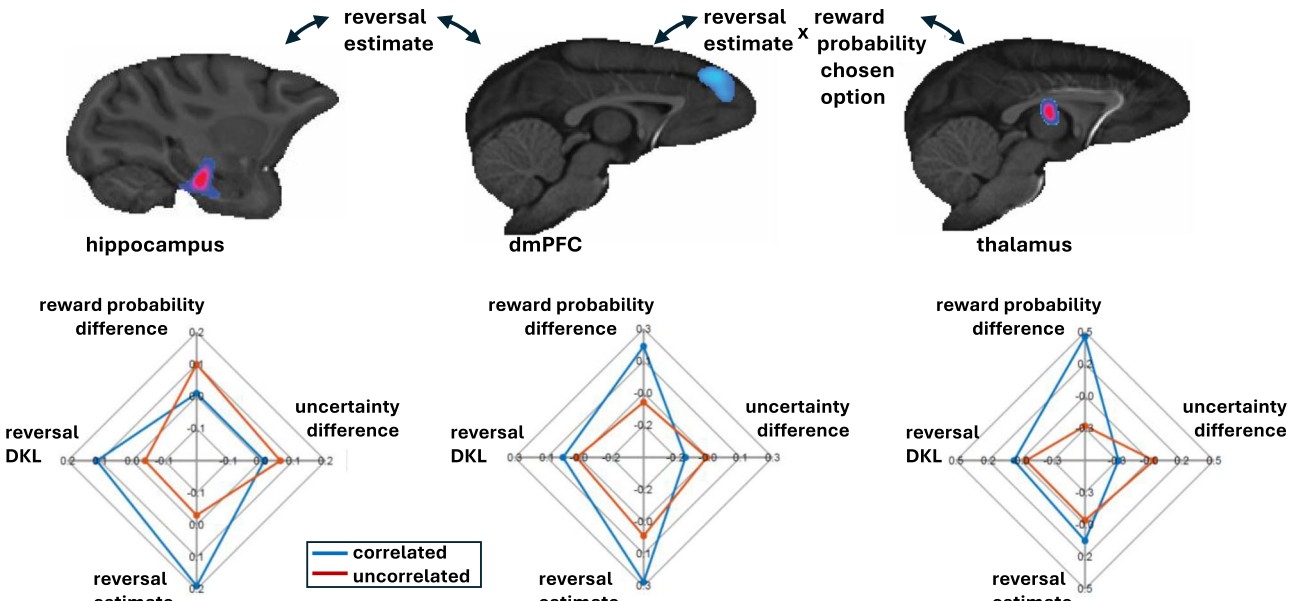

**Fig. 8 | Summary of results.** Each of the principal three regions, hippocampus, dmPFC, and anterior medial thalamus, is shown at the top. Beneath each brain section, the radar plots indicate the behavioural variables with which activity was most strongly related in the correlated (blue) and uncorrelated (red) tasks (reversal estimate and reversal $D_{KL}$ in hippocampus, reversal estimate, reversal $D_{KL}$, and the difference between the reward probabilities of the two choices in dmPFC, and the difference between the reward probabilities of the two choices in anterior medial thalamus). Interactions between hippocampal and dmPFC activity occurred as a function of the reversal estimate (top left) and between dmPFC and anterior medial thalamus as a function of the reversal estimate in interaction with the chosen option's reward probability.

stimuli[52] although this has been disputed[53]. What is certain, however, is that simply the presence of spatially defined action choices is not sufficient to implicate dmPFC; the uncorrelated task also entailed the making of spatially defined action choices, but neither dmPFC nor hippocampus tracked reversal estimates in the uncorrelated task. It is also worth noting that the dmPFC region linked to tracking reversal updates is dorsal to anterior cingulate regions linked to other aspects of choice flexibility and action selection[15,17,54].

The presence and strength of the reversal estimate-related activity in the correlated as opposed to the uncorrelated task is notable. Typically, more difficult tasks are associated with greater activity in dmPFC and other prefrontal regions[55,56], and the uncorrelated task may initially appear the more difficult of the two tasks. The dmPFC, appears to have a very central role not just in holding an estimate of the probability of transitioning from one latent task state to another but also in updating of these estimates (reversal $D_{KL}$) as each choice outcome is observed (Fig. 4). In addition, dmPFC activity also reflects the evaluation of choices when they are linked to the presence of latent task states in the correlated but not the uncorrelated task (Figs. 5A and 8). By contrast, action value-related activity was not found in the hippocampus. Hippocampal activity was more related to the tracking and updating of reversal estimates than in linking the estimates to changes in decision-making; correlated-uncorrelated task differences in reversal estimate-related activity in the hippocampus were more prominent prior to decision-making as opposed to at the time of decision-making (Figs. 4 and 8). While dmPFC-hippocampus activity coupling reflected the tracking of reversal estimates, dmPFC-anterior medial thalamus activity coupling reflected interactions between these reversal estimates and choice evaluation (Figs. 6J and 8). Together, these three areas, dmPFC, hippocampus, and anterior medial thalamus, which are interconnected, constitute a circuit tracking and updating latent state reversal estimates and translating these estimates into choice evaluation and selection. The identification of the primate hippocampus as important for establishing the latent state that the animal is in is consistent with very recent findings in rodents[57].

As noted, in addition to tracking reversal estimates, activity in dmPFC and hippocampus also tracked updates in the reversal estimate, reversal $D_{KL}$, at the time that macaques obtained feedback, reward or no reward, for their choices. While it was clear that reversal estimate-related activity was only found in the hippocampus in the correlated task, there was some evidence that dmPFC was modulated, albeit weakly, by reversal $D_{KL}$, in the uncorrelated task. This pattern of activity was, at first, difficult to interpret; it is difficult to know what such an activity pattern might mean when reversal estimates did not influence behaviour in the same way in the uncorrelated task (Fig. 3). To interrogate the meaning of the activity patterns further an alternative Bayesian model was constructed that tracked the values of individual choices in ways that were appropriate to either the correlated or uncorrelated task (Fig. 4). These models yielded individual choice option value updates—individual option $D_{KL}$ estimates—onto which brain activity could be regressed. When this was done, it was clear that both dmPFC and hippocampus tracked reversal $D_{KL}$ and only reversal $D_{KL}$ in the correlated task rather than individual option $D_{KL}$. Moreover, it was clear that the hippocampus contained no update-related activity in the uncorrelated task when no latent task states were present. However, the dmPFC activity pattern was less clear in the uncorrelated task. The marginally significant activity patterns suggest dmPFC may track option-specific $D_{KL}$ or even attempt to track reversal $D_{KL}$ in the uncorrelated task, perhaps because this had been adaptive in the other tasks that the animals had performed. However, these estimates did not propagate to the hippocampus. While dmPFC may have a more general role in updating, the hippocampus only tracks latent states.

Activity in all three areas—hippocampus, dmPFC, and anterior medial thalamus—survived more stringent thresholds for establishing activity in the whole-brain analyses ($z > 3.1$ as well as $z > 2.3$). This was apparent in whole-brain contrasts for reversal estimate in correlated versus uncorrelated sessions in hippocampus and in contrasts for correlated versus uncorrelated sessions and contrasts for correlated sessions for the reward probability difference associated with the two choices during decision-making in anterior medial thalamus and

dmPFC, respectively. We note, however, that the smaller size of the macaque brain may mean that techniques using whole-brain cluster correction need to be used with caution in macaque studies.

This picture of a distributed, multi-component circuit for translating latent state reversal estimates into choice evaluation was also supported by examination of the impact of disrupting each of the three areas with TUS (Fig. 7). TUS produces regionally specific changes in neural activity and behaviour that may be mediated by neuronal mechanosensitive ion channels[11–19,58,59]. Application of TUS to any of the three areas led macaques to perform the correlated task in a manner that was more similar to the uncorrelated task, in that switching was no longer guided by reversal estimates. Moreover, just as switching in the uncorrelated task was significantly more influenced by uncertainty about the choice values than was the case in the correlated task (Fig. 3 and Supplementary Fig. S2), switching also became more influenced by uncertainty about choice values in the correlated task after TUS of the hippocampus or anterior medial thalamus (Fig. 7 and Supplementary Fig. S5). Consistent with the notion that it is within dmPFC that latent state tracking and choice evaluation converge, we found that dmPFC TUS led to the loss of influence of almost all information concerning latent states, choice values, and uncertainties about choice values on switching. Now, only the value of the choice that animals had just been taking influenced whether they would take it again.

Some features of our task design are worth noting. First, in experiment 1, we tested animals on the correlated task prior to the uncorrelated task. One important reason for doing this, however, is to avoid introducing a potential confound; if macaques alternate over days, between the correlated and the uncorrelated tasks then they would effectively learn and track a higher order latent state variable: is the current state the one in which the rules of the correlated task prevail or the one in which the rules of the uncorrelated state prevail. Neural mechanisms for tracking latent tasks states would, thus, be employed even in the uncorrelated task and it would no longer be possible to identify their operation by comparing the correlated and uncorrelated tasks.

Second, we note that the TUS procedure we employed was an offline one; the TUS was applied for 40 s, and animals only began performing the task several minutes later and continued to do so for several tens of minutes. In this way, we hoped to reduce the impact of any auditory sensation associated with TUS application; the auditory impact of the TUS produces auditory-associated changes in neural activity that last several hundreds of milliseconds or possibly even 2 s, but they have not reported to produce neural activity changes beyond that time when we collected behavioural data in experiment 2. The 40 s train induces a change in activity in the stimulated area that exerts a measurable impact on neural activity for ~3 h and on behaviour for ~1 h. Because the impact of the stimulation outlasts the auditory impact of the stimulation, the behavioural effects cannot be explained as the consequence of auditory stimulation. In line with this interpretation, while this TUS protocol consistently produces neural changes that outlast the TUS application period in the targeted area, it does not induce any measurable change in auditory brain structures. We note that other neurostimulation techniques, such as theta burst transcranial magnetic stimulation, similarly induce neural changes that outlast the stimulation period but no auditory changes that outlast the TUS application. Moreover, it is important to emphasise that the sham TUS procedure that we employed entailed bringing the animal to the laboratory and making all the normal preparations for TUS administration such as registering the animal's head and brain to an MRI of the head and brain using neuronavigation tools and preparing the skin for TUS. The sham stopped short of actual TUS delivery and ended simply with simulation of the TUS sound. Through very careful and gradual training of the animals, we ensured that the TUS

procedure caused as little as anxiety as possible but any residual anxiety that might have been associated with the TUS procedure was controlled for in sham and TUS conditions.

The offline TUS method we used in the current study entails application of TUS several minutes before task performance even begins and therefore depends on the induction of a short-term plastic changes at synapses synapses synapses[12,14–19]. The ultrasound waves produced by TUS interact with neuron and astrocyte cell membranes and this, in turn, affects adjacent synapses in grey matter through an N-methyl-d-aspartate (NMDA)-dependent plasticity mechanism. This leads to the stimulated grey matter regions, but not adjacent or other regions, becoming insensitive to active changes in interconnected areas.

It is unlikely that there are precise one-to-one correspondences between primate and rodent prefrontal areas, but some broad similarities can be identified[60]. Individual anatomical regions within the prefrontal cortex may have become more specialised in primates while rodent OFC holds a wide range of activity patterns[60,61]. Therefore, while it may not be possible to identify an area that is the precise homologue of primate dmPFC in every respect, some aspects of the interactions between hippocampus, anterior medial thalamus, and a prefrontal region may be similar in rodents[62,63].

## Methods

### Statistics and reproducibility

No statistical method was used to predetermine sample size. No data were excluded from the analyses. The experiments were not randomised. The investigators were not blinded to allocation during experiments and outcome assessment.

### Animals

Seven adult male rhesus monkeys (*Macaca mulatta*) were tested; four performed the correlated and uncorrelated reversal tasks in experiment 1 while functional magnetic resonance imaging (fMRI) data were collected. Three performed the correlated task in experiment 2, in which the impact of transcranial ultrasound stimulation (TUS) on behaviour was measured while animals were in the same MRI scanner environment. They were aged between 9 and 12 years and weighed between 11 and 17 kg at the time of the experiments. They lived on a 12-h light–dark cycle, were fed once per day after testing, and had ad lib water access for an average of 15 h a day (and a minimum of 3 h per day).

Monkeys were previously trained to work in a sphinx position in a magnetic resonance imaging (MRI) scanner-compatible chair and to make arm-reaching movements to custom-made infrared touch sensors to obtain a liquid reward from a tube positioned directly in front of their mouth. Initially, they were trained while the MRI-compatible chair was placed in a wooden custom-made mock scanner. After the animals became accustomed to working in this way, they were trained to work in an actual MRI scanner. Task stimulus images were displayed on a screen (24 inches diagonal, 1000 mm away, centred at the monkey's eye level). Animals underwent aseptic surgery to implant an MRI-compatible head post (Rogue Research, Montreal, CA).

### Task

We used Presentation software (Neurobehavioral Systems Inc., Albany, CA) to control the experimental task displayed on the screen and to receive triggers from the MRI scanner. The task entailed presentation of two visually identical stimuli (plain rectangles) on either side of the monitor. The two stimuli were associated with different and specific reward schedules. A typical trial structure in experiment 1 is illustrated in Fig. 1a. At the beginning of each trial, two stimuli were presented. This time point was designated the stimulus presentation time. The monkey made a choice by touching a sensor immediately below one or other stimulus. This time point was designated the response time (RT)

point. The stimuli remained on, until the animal made contact with the sensor or 10 s elapsed. If the animal chose not to engage in the trial, the same trial was repeated until the monkey responded. When the animal responded, only the stimulus corresponding to the chosen option remained displayed on the screen. This marked the beginning of the action-outcome delay (AOD—random distribution between 0.103 and 8.770 s; mean 3.394 s and variance 0.605 s). After the AOD, a feedback picture was shown (green rectangle if the trial was rewarded, grey rectangle if it was not) for 2000 ms. If the choice was rewarded then, at the same time, the animal received a juice reward that was delivered by a spout next to the animal's mouth. There was then an intertrial interval (ITI) that was randomly distributed between 6500 and 7500 ms before the stimuli were presented again for the start of the next trial.

In experiment 2, the task was formally the same; animals chose between choice options defined by their location on the left and right of a computer monitor and received reward according to a similar schedule to that used in the correlated sessions in experiment 1. In experiment 2, however, animals did not reach directly towards the choice option itself but instead a cursor moved from the left to the right of the screen. Monkeys began each trial by pressing one response sensor which initiated cursor movement. They could then implement a choice of either the left or right option by waiting until the cursor had moved to be over the option. Our aim behind the change in response procedure in experiment 2 had initially been to test additional hypotheses by making further changes to the task but we changed this goal after disruption to testing during the COVID lockdown. Despite differences between superficial features of the tasks, the key behaviours and influences on behaviours were similar across the two tasks. This can be seen by comparing the blue bars in Fig. 3 and Fig. 6a and by comparing the blue bars in Supplementary Figs. S2 and S5A.

## Experimental design

In experiment 1, macaques performed two versions of a two-armed reversal bandit task characterised by specific action-reward contingencies (Fig. 1b illustrates example schedules). In the correlated environment, reward schedules for each option were negatively correlated with one another so the probability of reward for one option was high while the probability for the other option was low. Note, however, that because both options yielded reward stochastically, rather than deterministically, the negative correlation between the options' reward schedules was not $r = -1$ (Fig. 1c). In the second environment, the uncorrelated environment, reward schedules for each option were unrelated so the probability of reward for one option changed independently of the probability for the other option. The correlated task reversals marked the moments when abrupt changes in the reward schedules occurred. In the correlated environment, a change in the reward schedule for one option was linked with an opposite change for the other option such that one option was now rewarded, while the other option was no longer rewarded. In the uncorrelated environment, changes in the reward schedules for each option were independent. In the correlated task, reversals occurred randomly regardless of animal performance, and each session contained several such reversals (typically more than 3 reversals but varying between 1 and 7). Typically, an error occurred; the animal received no reward when reversals occurred. Animals became more likely to switch to the alternative option after an error occurred and, as multiple errors accrued, this tendency increased (Fig. 1d). The number of reversals in the correlated environments varied because of the stochastic nature of the task design. The reversal was not always programmed to occur after a fixed number of trials or after a fixed number of correct responses. This stochasticity ensured that animals monitored the outcomes of their choices in order to decide when to switch rather than switching after a fixed number of trials regardless of choice feedback. It also had the secondary advantage of helping to minimise collinearity between the reversal estimate and choice values.

In experiment 1, four monkeys performed the task first in the correlated environment then in the uncorrelated environment. Each monkey contributed 8-9 sessions of data in the correlated environment and 8-9 sessions in the uncorrelated environment. There were ~6000 and 6800 trials included in the correlated and uncorrelated task analyses, respectively. In experiment 2, a second group of three monkeys performed the task in the correlated environment only. Each animal contributed 5-6 sessions in which TUS was applied to the dorsomedial (dmPFC) site, 5-6 sessions in which TUS was applied to the hippocampus target, 5-6 sessions in which TUS was applied to the thalamus target, and 6-8 sham TUS sessions. By observing whether or not each choice is followed by reward over the course of trials, animals in both correlated and uncorrelated sessions can estimate the probability of reward associated with each option as well as their uncertainty about these estimates. In addition, however, in correlated sessions, animals may also estimate the probability of a reversal from one state to another[6,7]. In other words, the task moves between two relatively discrete phases where one option has a high probability of being rewarded while the other does not, and reward contingencies reverse repeatedly after successive intervals. We, therefore, use similar Bayesian models to derive trial-wise belief estimates about each option's reward probability and the uncertainties of these estimates in both correlated and uncorrelated sessions. We also examined whether, in addition, animals estimated a probability of reversal—the reversal estimate—from one state to the other and whether such estimates influenced the behaviour to a greater extent in the correlated as opposed to uncorrelated sessions.

## Bayesian modelling for correlated and uncorrelated environment

Two Bayesian models were used to estimate the beliefs that monkeys might optimally hold about task features given the observations of whether or not reward was delivered after each choice that was taken. We first describe the "correlated model". Because of the order in which the models are described in the main text, this model is referred to as "model 2" in the main text; as model 1 is in fact a simplification of model 2, we here describe model 2 first.

**Model 2 (correlated model).** First, we modelled the monkeys' beliefs (Fig. 2a) concerning the probability of reward associated with each choice, which determined each choice's value (left choice probability and right choice probability). In all cases, beliefs were captured by probability density functions (PDFs) comprising a mode that corresponded to the most likely level of the task feature (Fig. 2d, upper panel), for example, reward probability associated with the left choice, and a distribution reflecting uncertainty about the estimate given the observations that the monkey had made up until that point in the task.

We denote the reward probability under model 2 ($V_2$) for the chosen option (Ch) on any given trial, ($t$), as $V_2Ch_t$; the reward probability for the unchosen option as $V_2UnCh_t$ and the probability of a reversal (the reversal rate) as $H_{Ch}$; a representation of the reversal rate for the unchosen option is also maintained but is not used in any behavioural or neural analyses; hence for simplicity we will refer to $H_{Ch}$ simply as H. The ground truth value of $H$ is fixed throughout the experiment, but it is estimated on a trial-by-trial basis by the model; hence, the *belief* over H varies over time.

Beliefs were quantified by a joint probability density function (pdf) over a pre-defined hypothesis space (Fig. 2C):

$$p(V_2Ch_t = i, H = j) \quad \text{for} \begin{cases} i & \text{in } (0,1) \\ j & \text{in } (0,0.3) \end{cases} \quad (1)$$

for brevity, we refer to the probability $p(V_2Ch_t = i)$ and $p(H = j)$ as $p(V_2Ch_t)$ and $p(H)$ throughout.

The choice of the value range used for the reversal probability H (between 0 and 0.3) was based on behavioural simulations with the types of schedules actually used during the experiments. These simulations showed that even at true reversal time points, H would not take values large than 0.3 and that this was a reasonable parameter range to assume (blue dotted line in Fig. 2e illustrates the range of values of H in a typical session).

On each trial, monkeys selected one of the two available options and observed the outcome of that choice, defined as $x_t$:

$$x_t = \begin{cases} 1, & reward \\ 0, & no\ reward \end{cases} \tag{2}$$

The belief about the value $V_2Ch_t$ associated with each option on trial $t$ is expressed as a probability distribution $p(V_2Ch_t)$. On each trial, the belief is updated according to Bayes' theorem, by multiplication of the prior belief and the likelihood of the observed outcome $x$. This results in a posterior belief.

Before the very first observation, belief estimates about $V_2Ch_t$ and $H$ were assumed to be uniformly distributed across parameter space such that all possible reward probability values and reversal values, respectively, occurred with equal probability:

$$p(V_2Ch_t) = U(0, 1) \tag{3}$$

$$p(H) = U(0, 0.3) \tag{4}$$

A likelihood function is then calculated that describes the probability of a reward outcome $x_t$ on the current trial $t$ given all possible contingency values $V_t = [0 : 0.01 : 1]$

$$p(V_2Ch_t|x_t) = p(x_t|V_2Ch_t) = \begin{cases} V_2\,Ch_t & if\ x_t = 1 \\ 1 - V_2\,Ch_t & if\ x_t = 0 \end{cases} \tag{5}$$

With Bayes' rule, we derived a trial-by-trial posterior distribution proportional to the multiplication of a likelihood and a prior distribution $p(V_2Ch_t, H|x_{1:t-1})$, integrating both beliefs about an option's reward probability, $V_2Ch_t$, and the reversal probability, H:

$$p(V_2Ch_t, H|x_{1:t}) \propto p(x_t|V_2Ch_t) \cdot p(V_2Ch_t, H|x_{1:t-1}) \tag{6}$$

After each update, the posterior distribution was normalised such that all probabilities across the joint belief space $p(V_2Ch_t, H|x_{1:t})$ summed up to one.

In the correlated environment, options' contingencies are partially negatively correlated with each other such that if one option has a high reward contingency, the other option is likely to have a low reward contingency (Figs. 1a and 2a, b). This means that the ground truth value of $V_2Ch$ tends to change (from high to low or vice versa) in sudden jumps. Hence, belief updates over $V_2Ch$ are impacted by the subjective estimate of the probability of a reversal of the same option for the next trial, denoted as H.

The prior belief on the subsequent trial was therefore obtained via a transition function implementing a uniform leak:

$$p(V_2Ch_t, H|x_{1:t-1}) \propto ((1 - H) \cdot p(V_2Ch_{t-1}|x_{1:t-1}, H)) \\ + ((H) \cdot U(0,1)) \cdot p(H) \tag{7}$$

Again, the joint distribution $p(V_2Ch_t, H|x_{1:t-1})$ was normalized to sum to 1 over the joint hypothesis space in $V_2Ch_t$ and H.

Consequently, the monkey's prior belief about $V_2Ch_t$ on each trial is a weighted combination of the posterior belief about $V_2Ch_{t-1}$ on the previous trial, and a uniform or completely uncertain belief distribution over $V_2Ch_t$. The relative weighting depends on the estimate of the reversal frequency H. The resulting combined distribution has an

uncertainty (variance) that is high when the sequence of recent outcomes suggests a reversal has occurred[22,64].

Note, in contrast to some reversal learning tasks, the options' contingencies are not precisely the inverse of each other. Instead, the options' values negatively correlate with one another with different coefficients across sessions (Fig. 1c). Hence, the value of $V_2UnCh_t$ for the alternative option cannot be directly estimated as (for example) the complement of the value of $V_2Ch_t$ for the chosen option. Instead, the structure of the environment (with sudden "reversals") is captured in the structure of the transition function, and quantified by a single parameter (H). The model's estimate of H increases when it encounters trials on which the outcome (for the *chosen* option) is inconsistent with the current estimate of $V_2Ch_t$, which tends to happen after a reversal, as illustrated in Fig. 2e.

In summary, model 2 is a Bayesian model appropriate for correlated sessions. It is appropriate for the correlated sessions because it tracks the latent state structure of the environment to inform selection between options. The model makes it possible to define the expected value of reversal (referred to as the reversal estimate throughout the Results) as the expected value of the unchosen option:

$$ReversalEV = \sum p(V_2UnCh_t) * V_2UnCh_t \tag{8}$$

Note that while Eq. (8) summarises the estimated reward value to the animal of switching and taking the alternative action, our focus in both the behavioural and neural analyses is on how animals tracking the probability of reversal, which we refer to as the reversal estimate, and then separate variables track the reward values of the chosen option and the unchosen option (which are the estimates of the probabilities of reward for the chosen and unchosen option respectively) and the uncertainties of those estimates. The reversal estimate and the choice value variables used in our analyses were uncorrelated (Supplementary Fig. S1).

Each time an outcome is observed after a choice is made, the estimate of the reversal probability H is updated. The degree of update can be captured by the Kullback-Leibler divergence ($D_{KL}$) between the old (prior) and the new (posterior) belief distributions over H:

$$D_{KL}(H) = \sum_j p(H_t = j|x_{1:t}) \cdot log\frac{p(H_t = j|x_{1:t})}{p(H_t = j|x_{1:t-1})} \tag{9}$$

Where $j \in (0, 0.3)$ are candidate values of $H$ as per Eq. (1) above, and $p(H_t = j)$ is marginalised over $V_2Ch_t$

We show that neural activity in some brain areas reflects the reversal estimate $D_{KL}$ at the time of outcome (Fig. 4).

The model parameters used in the analysis of neural and behavioural data are as follows:

- Chosen probability estimate is the maximum likelihood value of $V_2Ch_t$ for the chosen option, marginalising over H.
- Chosen uncertainty is the width of the posterior distribution for $V_2Ch_t$, measuring from the 2.5 to 97.5 centile, for the chosen option, marginalising over H.
- Unchosen probability estimate is the maximum likelihood of $V_2UnCh_t$ for the unchosen option, marginalising over H.
- Unchosen uncertainty is the width of the posterior distribution for $V_2UnCh_t$, measuring from the 2.5 to 97.5 centile, for the unchosen option, marginalising over H.
- Reversal probability estimate is the expected value of H, marginalising over $V_2Ch_t$.
- The expected value of switching options: Reversal EV (as per Eq. (8)).
- The update in the estimate of H, Reversal $D_{KL}$ (as per Eq. (9)).

**Model 1 (uncorrelated model).** Because of the order in which models are presented in the Results, we refer to the model described above as

model 2. However, as noted in the Results, we developed an additional model, model 1, which quantified beliefs about each option without imposing the structure of abrupt jumps or 'reversals' in value. Here, we denote the reward probability or value under model 1 ($V_1$) of the chosen option (Ch) on trial t as $V_1Ch_t$ and for the unchosen option, $V_1UnCh_t$.

The transition function used in model 1 differed from that in model 2. Instead of inferring an abrupt jump in value (as in model 2), in model 1 the transition function applied a Gaussian "leak" between the posterior distribution on one trial, and the prior on the next.

$$p(V_1Ch_t|x_{1:t-1}) \propto p(V_1Ch_{t-1}|x_{1:t-1}) \otimes \mathcal{N}(0, \sigma^2) \quad (10)$$

Thus, the model allowed for gradual changes in value rather than abrupt jumps or reversals. The rate of change or 'drift rate' of each option, operationalized as the standard deviation of the Gaussian leak, was inferred by the model, by introducing a free parameter $\sigma^2$, which was inferred from the data analogous process to the inference of H in model 2 (Eqs. (2) and (6) above).

In summary, model 1 tracks the reward probability of a given option from observations of rewards and non-rewards received for choosing that option itself, independently of the observations made about the alternative option. While model 2 should be appropriate for correlated sessions, the simpler model 1 should be appropriate for the uncorrelated sessions, where both options' reward probabilities were independent of one other. It was, therefore, hypothesised that model 1 and model 2 were likely to provide the better accounts of data gathered in the uncorrelated and correlated sessions respectively and this was indeed the case (Fig. 2f). Model 1 did not track the latent task structure and therefore did not estimate the probability of a reversal (H in model 2), there was no equivalent of the reversal EV or of the reversal $D_{KL}$.

However, we calculated a complementary $D_{KL}$ measure that was appropriate for the uncorrelated environment (Fig. 5). This was the KL divergence over the option value itself, from the prior to the posterior:

$$D_{KL}(V_1Ch_t) = \sum_i p(V_1Ch_t = i|x_{1:t}) \cdot log \frac{p(V_1Ch_t = i|x_{1:t})}{p(V_1Ch_{t-1} = i|x_{1:t-1})} \quad (11)$$

Where $i \in (0, 1)$ are candidate values of $V1_t$ as per Eq. (1) above.

The model parameters used in the analysis of neural and behavioural data are as follows:

- The chosen probability estimate is the expected value of $V_1Ch_t$ for the chosen option.
- Chosen uncertainty is the width of the posterior distribution for $V_1Ch_t$, measuring from the 2.5 to 97.5 centile.
- Unchosen probability estimate is the expected value of $V_1UnCh_t$, for the unchosen option.
- Unchosen uncertainty is the width of the posterior distribution for $V_1UnCh_t$, measuring from the 2.5 to 97.5 centile.
- The option-specific KL divergence as per Eq. (10).

## Behavioural analysis

We used a logistic mixed-effects regression model (LMEM1) analysis to test whether model-derived belief estimates about each option's reward probability and reward uncertainty, and the reversal estimate predicted whether animals switch their choice selection on the next trial, and we examined whether there were differences in how this was done in the two conditions (correlated and uncorrelated). Choice values were coded in terms of the probability of reward estimated for the choice that had been taken (*chosen_probability*; the Bayesian model's modal estimate of the probability of reward for the choice taken), the probability of reward estimated for the choice that had not been taken (*unchosen probability*; the Bayesian model's modal estimate of the probability of reward for the choice that was not taken).

The regression model also examined the influences of uncertainties of these two estimates, chosen_uncertainty and unchosen_uncertainty, respectively (corresponding to 95% widths of the probability density functions for the estimates of each probability in the Bayesian model). Finally, the regression model also contained *reversal estimate*, an estimate of whether the task was changing from one latent state to another). An initial bias term captured a baseline tendency to switch; the negative values of this term indicate animals are predisposed to repeat the same choice as on the last trial. We used the *lmer4* package in the R environment. All LMEMs included by-subject and by-session random intercepts and by-subject random slopes for the main effects. The first analysis (LMEM1) employed a simplified approach that examined the impact of the reward probabilities and uncertainties for the two options in terms of their differences (where probability difference is the difference between the reward probability associated with the chosen option and the unchosen option; uncertainty difference is the difference between the reward uncertainty associated with the chosen option and the unchosen option) in the two conditions (correlated and uncorrelated). LMEM1 thus included each of the following variables (probability difference, uncertainty difference, reversal estimate, and task) as fixed effects together with random slopes for *choice-probability-difference, choice-uncertainty-difference*, and *reversal-estimate*. A second model, LMEM2, was similar but considered the influence of the chosen option probability, the unchosen option probability, the uncertainty of the chosen option probability estimate, and the uncertainty of the unchosen probability option separately. All LMEMs were performed in *R* and models were fit via maximum likelihood estimation as implemented in the *lme4* toolbox.

**LMEM**1: $logit(switching_t) = \beta_o + \beta_1 choice-probability-difference_t + \beta_2 choice-uncertainty-difference_t + \beta_3 reversal-estimate_t + \beta_4 condition_t + \beta_5(choice-probability-difference_t*condition_t) + \beta_6(choice-uncertainty-difference_t*condition_t) + \beta_7(reversal-estimate_t*condition_t)$

where, on trial t, *choice-probability-difference_t* is the difference between chosen probability_t and unchosen probability_t, *choice-uncertainty-difference_t* is the difference between the chosen uncertainty_t and unchosen uncertainty_t, *reversal-estimate_t* is reversal estimate and *condition_t* indicates whether the macaque was performing the correlated or uncorrelated task.

**LMEM**2: $logit(switching_t) = \beta_o + \beta_1 chosen-probability_t + \beta_2 unchosen-probability_t + \beta_3 chosen-uncertainty_t + \beta_4 unchosen-uncertainty_t + \beta_5 reversal-estimate_t + \beta_5 condition_t + \beta_6(chosen probability_t*condition_t) + \beta_7(unchosen probability_t*condition_t) + \beta_8(chosen-uncertainty_t*condition_t) + \beta_9(unchosen-uncertainty_t*condition_t) + \beta_{10}(reversal-estimate_t*condition_t)$

## MRI data collection

During collection of fMRI data, monkeys were head-fixed in a sphinx posture within an MRI-compatible chair (Rogue Research). In experiment 1, the data collection was performed using a 3 T horizontal bore MRI scanner coupled with a four-channel phased array receiving coil and a radial transmission coil (Windmiller Kolster Scientific Fresno, CA)[12,20,27]. To optimise the animal's head coverage, the coil's loops were each 8 cm large and similar to previous studies on awake non-human primates[12,15,17,18,20]. The animals were positioned in a chair which was placed on the sliding bed of the scanner. The receiver coils were placed on the side of the animal's head with the transmitter placed on top. Functional data were acquired using a gradient-echo T2* echo planar imaging (EPI) sequence with a 1.5 × 1.5 × 1.5 mm resolution, repetition time (TR) 2.28 s, echo time (TE) 30 ms and flip angle 90°. At the end of each session, proton-density-weighted images were acquired using a

gradient-refocused echo (GRE) sequence with a $1.5 \times 1.5 \times 1.5$ mm resolution, TR 10 ms, TE 2.52 ms, and flip angle 25°. These images were later used for offline MRI reconstruction.

In experiment 2, data were collected in the same MRI scanner but with an upgraded 15-channel non-human primate (NHP)-specific receive coil (RAPID Biomedical). Functional images were acquired via the CMMR multiband gradient-echo T2* echo planar imaging (EPI) sequence designed specifically to achieve high signal-to-noise (SNR) in subcortical structures[65,66]. This was characterised by 1.25 mm isotropic voxels with a repetition time (TR) of 1.282 s, echo time (TE) of 25.40 ms, multiband acceleration factor MB = 2, in-plane acceleration factor R = 2, and flip angle of 63°.

We acquired T1-weighted MP-RAGE images on separate days when animals were under general anaesthesia. Anaesthesia was only used for collecting T1-weighted structural images. The parameters of the T1 images were as follows: resolution $f = 0.5 \times 0.5 \times 0.5$ mm, TR = 2.5 s, TE = 4.04 ms, inversion pulse time (TI) = 1.1 s, and flip angle = 8°. Anaesthesia was induced by intramuscular injection of 10 mg/kg ketamine, 0.125–0.25 mg/kg xylazine, and 0.1 mg/kg midazolam and maintained with isoflurane.

## fMRI data processing

Preprocessing was performed using a series of custom-made scripts using functions from FMRIB Software Library (FSL)[67], Advanced Normalization Tools (http://stnava.github.io/ANTs)[68], the Human Connectome Project Workbench (https://www.humanconnectome.org/software/connectome-workbench)[69], as well as the Magnetic Resonance Comparative Anatomy Toolbox (MrCat; https://github.com/neuroecology/MrCat). These steps have been validated in previous fMRI awake macaque studies[17,24,70]. In a first step, T2*-weighted EPI images were reconstructed by an offline-SENSE method that achieved higher signal-to-noise and lower ghost levels than conventional online reconstruction[71] (Offline_SENSE GUI, Windmiller Kolster Scientific, Fresno, CA). For each session, low-noise EPI reference images were calculated in order to non-linearly register all volumes on a slice-by-slice fashion along the phase-encoding direction in order to correct for time-varying distortions in the main magnetic field that can be caused by body and limb motion. Both aligned and distortion-corrected EPI images were subsequently non-linearly registered to each animal's high-resolution T1-weighted images. A group-specific template was then created by coregistering each monkey's structural image to the CARET macaque F99 space[71]. Finally, we applied high-pass temporal filtering (3-dB cutoff of 100 s) and spatially smoothed the functional images using a Gaussian spatial smoothing (full-width half maximum of 3 mm).

The T1-weighted images were processed in an iterative fashion, cycling through a macaque-optimised implementation of FSL's brain-extraction tool (BET)[72], RF bias-field correction, and linear and non-linear registration (FLIRT and FNIRT)[73,74] to the *Macaca mulatta* McLaren template in F99[75] as implemented in MrCat. The GRE image was used to aid the offline T2* EPI image reconstruction based on SENSE (Windmiller Kolster Scientific)[71].

A T1w group template specific to the set of subjects was constructed using two iterations of (1) registration to an initial template in F99 space[75], (2) group averaging, (3) registration to the new group template. This was accomplished using tools from Advanced Normalisation Tools (ANTs) as implemented in MrCat whereby at each step the group template was registered to the source template, thus avoiding drift and retaining registration to F99 space. All coordinates reported (in millimetres) refer to F99 space, and results are shown on the group template.

While the monkeys were head-fixed, their limb and body movements during task performance distorted the main (B0) magnetic field in a time-varying manner, causing non-linear motion-related artefacts in the phase-encoding direction varying on a slice-by-slice basis. To correct for these artefacts, using a processing pipeline implemented in MrCat, each slice was registered, first linearly, then non-linearly, to a robust reference based on EPI volumes from the same timeseries with the least distortion. To avoid overfitting, the degrees of freedom were constrained in several ways: only distortions along the phase-encoding direction were considered; registration was initialised using priors from temporally neighbouring slices; low-order solutions were preferred over high-order registration (rigid > affine > non-linear); non-linear degrees of freedom were regularised using b-splines.

Finally, the slice-registered average functional image was non-linearly registered to the high-resolution structural reference of each subject, and this was registered to the group-specific template using ANTs. Brain-extraction of EPI timeseries was based on masks obtained in the high-resolution structural space. Next, EPI images were spatially smoothed (3 mm FWHM) and temporally high-pass filtered (cutoff 100 s). First-level whole-brain analyses (see below) were performed on the low-resolution images in the original acquisition space.

## fMRI analysis

A univariate general linear model (GLM) approach was taken for the statistical analysis of the whole-brain functional data using FEAT (FMRI Expert Analysis Tool) Version 6.00, part of FSL[76]. GLM1 was based on LMEM1, described above, which had been used to examine behaviour. It compared the influence of the probabilities of reward associated with the chosen and unchosen option and it compared the influences of uncertainties in these estimates.

**GLM1:**   $\mathrm{BOLD} = \beta_1 \text{decision event} + \beta_2 \text{chosen probability} + \beta_3 \text{unchosen probability} + \beta_4 \text{chosen uncertainty} + \beta_5 \text{unchosen uncertainty} + \beta_6 \text{reversal estimate} + \beta_7 \text{outcome event} + \beta_8 \text{reward} + \beta_9 D_{KL} + \beta_{10} \text{Side} + \beta_{11} \text{Left\_instantaneous} + \beta_{12} \text{Right\_instantaneous} + \beta_{13} \text{Juice\_instantaneous} + \beta_{14\ldots26} \text{Motion} + \beta_{27\ldots n} \text{LowQ}$

The first four regressors in GLM1 and the first six regressors in GLM1a (see below) relate to the decision-making period when the macaque made a choice:

decision event: main effect of decision (boxcar with onset 1000 ms before response and duration 500 ms).

chosen probability: probability of reward associated with the chosen option.

unchosen probability: probability of reward associated with the unchosen option.

chosen uncertainty: uncertainty of the estimate of reward probability for the chosen option.

unchosen uncertainty: uncertainty of the estimate of reward probability for the unchosen option.

reversal estimate: uncertainty weighted probability of reversal from the current latent task state to the other.

The next three regressors relate to the decision outcome period when the macaque either does or does not receive reward:

outcome event: main effect of outcome (boxcar with onset at outcome onset and duration 500 ms).

Reward: whether reward was delivered or not.

$D_{KL}$: updating of the reversal estimate (Kubler-Liebach divergence).

The remaining regressors relate to potential confounding variables.

Side response side (left or right).

Amongst the confound variables, the following were not convolved by the haemodynamic response function in order to ensure that they would capture changes in the BOLD signal that were not neural in origin but which were instead instantaneously caused by field distortion.

Left_instantaneous: Regressor aligned to the onset of the volume in which a left-hand movement was recorded, with a duration of 1 TR = 2.28 s.

Right_instantaneous Regressor aligned to the onset of the volume in which a left-hand movement was recorded, with a duration of 1 TR = 2.28 s.

Juice_instantaneous Regressor with onset and duration corresponding to juice delivery but fixed amplitude.

$\beta_{12...24}$Motion + 13 noise regressors indexing the time-varying signal distortions, including the mean signal intensity time course and 12 remaining principal components describing the volume-by-volume magnetic field distortions induced by limb and body movements as estimated during preprocessing.

$\beta_{26...n}$ LowQ Regressors flagging low-quality EPI volumes, suffering from strong artefacts, to be excluded from the analysis.

In GLM1, the chosen probability, unchosen probability, chosen uncertainty, and unchosen uncertainty regressors were derived from the application of model 1 to uncorrelated and correlated conditions. This meant that the regressors were calculated in an identical manner when both correlated and uncorrelated conditions were analysed. GLM1a was identical in every respect but chosen probability, unchosen probability, chosen uncertainty, and unchosen uncertainty regressors were derived from the application of model 1 to the uncorrelated condition data and model 2 to the correlated condition data. This meant that the model that was more appropriate for the correlated sessions was now used to make the regressors. However, because the regressors remain similar regardless of the method by which they are calculated, the results obtained with either approach remained similar too. In all cases, the first-level analysis was performed on each scanning session (70 sessions in total). The contrast of parameter estimates (COPEs) and variance estimates (VARCOPEs) for each scanning session were then combined in a second-level mixed-effects analysis (FLAME 1 + 2), treating sessions as random effects with a cluster forming threshold of $z > 2.3$ and $P < 0.05$.

Additional analysis of reversal estimate-related activity in ROIs (Fig. 4) was performed with an ANOVA examining the effects of brain area (posterior dmPFC, anterior dmPFC, hippocampus), condition (correlated versus uncorrelated), and session (8-9 sessions). Additional analysis of reversal $D_{KL}$-related and option-specific $D_{KL}$ activity in correlated sessions (Fig. 5) was performed with an ANOVA examining the effects of reversal update type (reversal $D_{KL}$ versus option-specific $D_{KL}$) in correlated sessions.

Volume quality was assessed based on (1) slice-registration cost (the normalized correlation between the current volume and the robust average after optimal registration), (2) linear scaling along the phase-encoding direction (directly related to signal intensity loss due to motion distortion), (3) non-linear deformation (penalising volumes that require highly non-linear deformations). Each LowQ regressor was equal to one for one volume and zero otherwise.

For BOLD time-course analyses, we extracted the filtered BOLD time series from each ROI (dmPFC, anterior medial thalamus, hippocampus). The extracted signals were then averaged, normalised and upsampled by a factor of 15[12,18–20]. The upsampled data was then epoched to 10 s time windows spanning 5 s before to 5 s after the appearance of the reward-opportunity stimulus on each trial, or the outcome on each trial. We then examined the relationship between behaviour and brain activity with ordinary least squares (OLS) GLMs performed at each timepoint in each epoch.

Whenever it was necessary to perform inferential statistics on time-course data while avoiding circularity ("double dipping") given the statistical test first used to identify the area of activity, we used a leave-one-out (LOO) cross-validation procedure designed to estimate the peak regression-coefficient in each session without selection bias[12,18,24]. For each session, s ($N = 70$), we determined the timepoint t at which the largest absolute-value regression coefficient occurred in the remaining N-1 (i.e., 69) sessions. We restricted our search for t to a 4 s window from 1 s after to 5 s time-locked to action (i.e., a 4-s window centred on the mean macaque haemodynamic response function). We then calculated the regression coefficient in session s at time $t$. We repeated this iteratively for each session, which yielded a series of 70 regression coefficients. We performed significance testing on regression coefficients between conditions with a Wilcoxon signed-rank test. All-time course analysis was conducted in MATLAB (MathWorks) using custom analysis scripts.

**Transcranial ultrasound stimulation (TUS) data collection**
TUS was performed using a four-element annular array transducer (NeuroFUS CTX-250, 64 mm active diameter, Brainbox Ltd, Cardiff, UK) combined with a programmable amplifier (Sonic Concept Inc's Transducer Power Output System, TPO−105, Brainbox Ltd, Cardiff, UK). The transducer was paired with a transparent coupling cone filled with degassed water and sealed with a latex membrane. The water was degassed for 4–5 h before each stimulation session and was replaced after each session. The resonance frequency of the ultrasonic wave was set to 250 kHz. The stimulation protocol was based on previously established protocols in macaques[12–19]. We used the following protocol: duty cycle 30%; pulse length 30 ms; pulse repetition interval 100 ms; total stimulation duration 40 s. The pressure field from the transducer was measured in a water tank with a 75-μm diameter PVDF needle hydrophone (Precision Acoustics, Dorset, UK) which had been calibrated at 250 kHz by the National Physical Laboratory (Teddington, UK). The free-field spatial-peak pulse-average intensity (Isppa) corresponded to 60 W/cm$^2$ in free field at the focus (based on our own calibration of the device and in line with the intensity recorded in previous studies[19]). Acoustic simulations were performed using the pseudo-CT from a representative macaque brain. The pseudo-CT scan was estimated from pseudo-CT images obtained from a typical monkey using a Black Bone MRI sequence[77] (https://github.com/ucl-bug/petra-to-ct). We debiased the Black Bone MR image using Slicer's N4ITK MRI bias correction[78,79], performed a segmentation to obtain head and skull masks, and then created the pseudo-CT by setting the background to values of −1000, setting voxels classified as soft tissue to +42, and linearly mapping the intensities of voxels classified as skull into a range of 42–3274. We checked that the distribution of pseudo-Hounsfield units inside the skull of the pseudo-CT was similar to the distribution of Hounsfield units inside the skull of a real CT acquired for another animal (mode around 1500, roll-off to 2500). We then used a mapping of Hounsfield units onto units of mass density (kg/m$^3$) based on the pseudo-CT scan of a phantom with known mass densities on a comparable 3 T scanner. The simulations indicated values of 18.5 W/cm$^2$, 17 W/cm$^2$ and 16 W/cm$^2$, for the Isppa achieved in situ around the focus in hippocampus, anterior medial thalamus, and dmPFC, respectively. The MI was 1.49, 1.42 and 1.39 and the thermal dose 0.009 and 0.0325 and >100, respectively, in the vicinity of the three targets. Although the highest thermal dose was in the skull rather than in the brain itself, this shows that for human studies, the chosen pulse protocol is not recommended and can exceed ITRUSST-recommended safety limits.

At the beginning of each stimulation session, the animal's skull was shaved, and a conductive gel (SignaGel Electrode; Parker Laboratories Inc.) was applied to the skin. The water-filled coupling cone and the gel were used to ensure ultrasonic coupling between the transducer and the animal's head. Next, the ultrasound transducer/coupling cone was placed on the skull, and a Brainsight Neuronavigation System (Rogue Research, Montreal, CA) was used to position the transducer so that the focal spot would be cantered on the targeted brain region. There were four

stimulation conditions: (1) dmPFC; (2) hippocampus; (3) thalamus; (4) sham (passive control condition). In the first three cases, TUS was administered bilaterally (40 s to the left hemisphere; 40 s to the right hemisphere in each session) to the same region in each hemisphere (dmPFC: $x = 1.43$, $y = 23.31$, $z = 16.72$ and $x = -2.17$, $y = 23.31$, $z = 16.6$, hippocampus: $x = 15.1$, $y = -13.34$, $z = -10.11$, and $x = -15.21$, $y = -13.34$, $z = -10.11$, anterior medial thalamus $x = -0.18$, $y = -8$, $z = 6.04$ which were, respectively, ~40, 52, and 52 mm from the transducer surface (there were small variations depending on subject identity).

After stimulation, monkeys were immediately moved to the testing room for behavioural data collection. We estimate that the delay between the stimulation and the start of the behavioural task took 5–10 min on average. The sham condition completely matched a typical stimulation session (setting, stimulation procedure, neuronavigation, targeting, transducer preparation and timing of its bilateral application to the shaved skin on the head of the animal) except that sonication was not triggered, although the TUS sound was simulated. During the sham session, the montage was pseudo-randomly positioned to target dmPFC, hippocampus, and thalamus. Each stimulation condition was repeated 5–8 times for each animal, and each test took place on a different day. The order of TUS targets was pseudo-randomised for each animal in order to cycle through the different conditions, but with the addition of a sham day at the beginning and end of the series. This resulted in 8 sham testing days and 6 days of testing at each other site with the exception of one animal that only underwent 5 sessions of testing at each target site. The stimulation and testing session were always performed at the same time of the day, and there was always a 24-h gap between each session, regardless of it being a real or sham stimulation session.

We used the acoustic simulation toolbox (k-Plan, Brainbox, UK) to simulate the depth and acoustic intensity of the TUS focus in the dmPFC, hippocampus, and thalamus. The simulations used both the T1-weighted scan and a pseudo-computed tomography scan, which was generated based on the T1-weighted scan.

## TUS analysis

We used logistic mixed-effects regression models (LMEM3; LMEM4) to test whether model-derived belief estimates about each option's value and the reversal estimate predicted whether animals would switch their choice selection on the next trial. LMEM3 and LMEM4 were similar to models LMEM1 and LMEM2 reported above. However, rather than examining the effect of the task type (correlated versus uncorrelated), LMEM3 and LMEM4 examined the effect of TUS (sham, dmPFC, hippocampus, thalamus). As before, we used the *lmer4* package in the R environment, with sessions and animals as random intercepts and the interaction between TUS (sham, dmPFC, hippocampus, thalamus) and choice probability difference, choice uncertainty difference, and reversal estimate as fixed effects, together with their respective random slopes. Model LMEM4 was similar but considered the influence of the chosen option probability, the unchosen option probability, the uncertainty of the chosen option probability estimate, and the uncertainty of the unchosen probability option separately.

**LMEM3**: $logit(switching_t) = \beta_o + \beta_1 choice - probability - difference_t + \beta_2 choice - uncertainty - difference_t + \beta_3 reversal - estimate_t + \beta_4 TUS_t + \beta_5 (choice - probability - difference_t * TUS_t) + \beta_6 (choice - uncertainty - difference_t * TUS_t) + \beta_7 (reversal - estimate_t * TUS_t)$

where, on trial t, *choice-probability-difference_t* is the difference between chosen probability_t and unchosen probability_t, *choice-*

*uncertainty-difference_t* is the difference between the chosen uncertainty_t and unchosen uncertainty_t and *reversal-estimate_t* is reversal estimate, and *TUS* is ultrasound condition at trial $t$

**LMEM4**: $logit(switching_t) = \beta_o + \beta_1 chosen - probability_t + \beta_2 unchosen - probability_t + \beta_3 chosen - uncertainty_t + \beta_4 unchosen - uncertainty_t + \beta_5 reversal - estimate_t + \beta_5 TUS_t + \beta_6 (chosen\ probability_t * TUS_t) + \beta_7 (unchosen\ probability_t * TUS_t) + \beta_8 (chosen - uncertainty_t * TUS_t) + \beta_9 (unchosen - uncertainty_t * TUS_t) + \beta_{10} (reversal - estimate_t * TUS_t)$

## Ethics

The experiment was conducted in accordance with European Union Directive 2010/63/EU of the European Parliament on the protection of animals used for scientific purposes and with the United Kingdom (UK) Animals (Scientific Procedures) Act 1986 under licences approved by the UK Home Office.

## Reporting summary

Further information on research design is available in the Nature Portfolio Reporting Summary linked to this article.

## Data availability

The processed data and results to reproduce the figures of the paper, including Supplementary Figs., have been deposited in the OSF repository https://osf.io/54k9g/.

## Code availability

The MATLAB custom code supporting the behavioural results of this study is available in the OSF repository https://osf.io/54k9g/. Any remaining code that supports the findings of this study is available from the corresponding author upon request.

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

## Acknowledgements

Funded by the MRC, BBSRC, and Wellcome Trust. This research was funded in whole or in part by Biotechnology and Biological Sciences Research Council (BBSRC) (grant number BB/W003392/1) and Wellcome Trust (grant number 221794/Z/20/Z). For the purpose of Open Access, the author has applied a CC BY public copyright licence to any Author Accepted Manuscript (AAM) version arising from this submission.

## Author contributions

Initial experimental design: F.-X.N., M.C.K.-F., and M.F.S.R.; functional magnetic resonance image data collection for experiment 1: K.M., U.S., F.-X.N., and J.Sa.; functional magnetic resonance image data collection forr experiment 2: K.M. and U.S.; transcranial ultrasound stimulation: K.M. and U.S.; data analysis: K.M., C.M.H., N.T., J.X.O'.R., M.C.K.-F., and M.F.S.R.; Methodological support: S.S., J.Sc., J.A., and N.K.; manuscript preparation: K.M., N.T., M.C.K.-F., and M.F.S.R.

## Competing interests

The authors declare no competing interests.
