## [Transparent Peer Review file · Nature Communications]

Interaction and functional specialization across a distributed neural circuit for flexible task control in macaques

Corresponding Author: Dr Kevin Marche

Version 0:

Reviewer comments:

Reviewer #1

(Remarks to the Author)

This study demonstrates that flexible task control in primates relies on interactions among multiple brain regions, each playing a distinct role. By conducting experiments with macaques, the authors illuminate how three critical areas in the prefrontal cortex, hippocampus, and thalamus cooperate to anticipate rule changes (reversals) and evaluate choice value. They also reveal functional specialization within these regions—most notably, the dorsomedial prefrontal cortex (dmPFC) encodes the probability of reversals to guide behavior, while the hippocampus tracks updates related to these reversals. Furthermore, by utilizing transcranial-focused ultrasound stimulation (tFUS), the authors provide causal evidence that each of these areas contributes uniquely to flexible switching based on reversal estimates. Altogether, this work illustrates how a distributed neural network in the macaque brain organizes and allocates processes essential for flexible decision-making and learning. Although the task used is relatively simple and the computational framework for analyzing decision-making is mathematically complex, the authors convincingly show how these three regions specifically engage in reversal learning and inferential processes that underpin decision-making.

Below are four major issues that, if addressed, could make the research even more systematic and robust.

Major Comments

1. Comparison between “correlated” and “uncorrelated” tasks

While the authors show that both “correlated” and “uncorrelated” tasks share a basic target-switch response after no-reward outcomes (Fig. 1d), Fig. 1b indicates that monkeys persist with the same choice longer in the “correlated” task than in the “uncorrelated” task. Consequently, they employ distinct computational models for each condition (Fig. 2E) to accommodate these differing strategies. However, the markedly different reversal frequencies suggest that the “uncorrelated” task may not serve as a straightforward control for the “correlated” task.

Two key questions could make this comparison more compelling:

(1-1) What if the authors increased the reversal frequency in the “correlated” task?

If the resulting behavior begins to resemble that of the “uncorrelated” task, it would highlight continuity between the two tasks, suggesting that each model represents a different position on a broader spectrum of learning strategies.

Demonstrating such a shift would strengthen the rationale for using the “uncorrelated” task as a control and clarify why two distinct models are necessary.

(1-2) Would parameters from Model 1 converge toward those of Model 2 if reversal frequency were increased?

Assuming that both tasks lie along a continuum, it is equally important to examine whether their underlying computational models might converge. If increasing the reversal frequency makes the monkeys’ behavior in the “correlated” task more akin to that in the “uncorrelated” task, do the parameters in Model 1 gradually align with those from Model 2? One approach could be to examine such convergence via model selection procedures (e.g., comparing AIC or BIC) to see whether the “correlated” task’s parameter estimates transition toward those of the “uncorrelated” task’s model under higher reversal frequencies. If direct experimentation is challenging, at least a discussion of how these two tasks and models relate to one another would help clarify the broader framework in which they both operate.

2. Bayesian Model and Neural Decoding Consistency

The authors employ a computational framework (e.g., Bayesian or reversal-learning models) to derive parameters that they then correlate with neural activity. The strength of this work depends on how these model-derived variables are defined, validated, and interpreted. Two points warrant careful attention:

(2-1) Multivariable Regression and Multicollinearity

A common approach is to use multivariable regression to find neural correlates of computational parameters. However, such methods can face interpretive challenges—especially regarding multicollinearity. Because parameters like chosen probability, unchosen probability, chosen uncertainty, unchosen uncertainty, and reversal estimate may interact or correlate, the authors should demonstrate that each parameter is a legitimate reflection of an underlying neural process. They should also clarify how they have addressed or minimized multicollinearity. Providing explicit criteria for parameter selection—and a sound rationale for assigning each parameter to decode specific brain regions—would further validate the manuscript.

(2-2) Arbitrariness of Model-Derived Parameters

While the authors propose seemingly appropriate variables for capturing relevant computations, such parameter choices can be somewhat arbitrary. Justifications or evidence are needed to show that these parameters are both necessary for their computational goals and plausible from a neural coding perspective. For example, the authors could explain why these particular parameters—rather than alternative formulations—best capture the behavioral and neural data and how they align with or extend established decision-making models. Demonstrating that each parameter plausibly mirrors a distinct neural function would substantially strengthen the link between these computational variables and their neural correlates.

3. Consistency With Prior OFC Research

Previous literature has shown that orbitofrontal cortex (OFC) lesions can compromise reversal tasks, emphasizing OFC's role. Nevertheless, this study—and related work—indicates that other areas connected to OFC also contribute. Here, the authors identify specific regions involved in a “correlated” task with a reversal component, applying whole-brain analysis and tFUS to confirm causal involvement. However, given OFC's established causal role in simpler reversal paradigms, the authors should clarify how their new task design diverges from previous ones. For example, do lower reversal frequencies or probabilistic structures (as opposed to deterministic ones) modulate OFC engagement? The authors should compare their methodology with earlier protocols and discuss whether the identified circuit is uniquely tied to probabilistic or latent state-based reversals, whereas OFC might be more pivotal in simpler or more discrete reversal tasks.

Minor Comments

By addressing these points, the authors could provide a more cohesive and comprehensive account.

(minor-1) Additional Behavioral Indices (RT, Gaze, etc.)

Beyond basic choice data, measures like reaction time (RT) or eye tracking can provide additional insights into how animals track and update latent states or reversal probabilities. For instance, RT may reflect decision difficulty or uncertainty, while gaze patterns might indicate exploratory behavior regarding alternative options. Such richer behavioral metrics (e.g., exploration-exploitation balance, confidence levels, hesitation times) could reveal how tFUS affects learning strategies and further link model parameters to observed behavior. Including these metrics would greatly enrich interpretations of the tFUS results and model-based analyses.

(minor-2) Reproducibility and Individual Differences

How consistent are the findings across individual macaques? To what extent do variations in behavior or anatomy among subjects influence the main conclusions?

(minor-3) Local Impact of tFUS

When tFUS is applied to deeper regions such as dmPFC, hippocampus, or thalamus, how confined is the stimulation to the intended target? Could nearby white matter tracts or adjacent regions be inadvertently affected? Further, how did the author confirm the selective and focal modulation of the tFUS? Clarifying this would help ensure an accurate interpretation of any observed behavioral and neural effects.

(Remarks on code availability)

Reviewer #2

(Remarks to the Author)

The manuscript reports a study investigating the networks and computations underlying performance in reversal learning tasks in non-human primates. Specifically, a model emphasizing identification of transitions between latent states is tested. An fMRI experiment shows that macaques track latent state transitions in addition to choice values in dorsomedial frontal cortex and anterior and dorsomedial thalamus. The hippocampus tracks the probability of a reversal between latent states. In a follow-up experiment, focused transcranial ultrasound (TUS) is applied to these regions impaired reversal learning, mainly the use of the reversal estimate parameter.

This is an interesting manuscript addressing a long-debated topic. It has a potentially high impact on the way reversal learning (or, more generally, learning latent task structures) is implemented in the primate brain. The approach is innovative

in several ways. Recording fMRI in the awake and behaving macaque is technically challenging and TUS is an innovative method of non-invasive brain stimulation.

However, at this point, a number of questions and unclarities remain, such that it is difficult to evaluate whether the inferences drawn are firm enough to warrant publication. The following list of comments may be helpful for revising and clarifying the manuscript:

1. How were the animals trained to learn the different task structures (correlated vs. uncorrelated)? Were they cued before each session which task variant will be performed? If not, how did the animals find out on which version they currently work? Could there be a spill-over from one variant to the other? E.g., how can we be sure that the animals did not try to search for latent state transitions in the uncorrelated version or actually tracked two sets of states (one for each stimulus) that could change between high or low reward probability?
2. It seems counterintuitive that the switch rate is reduced after three consecutive non-rewarding trials compared to the switch rate after two non-rewarding trials. It is astonishing that the macaques would persevere further in 2/3 to 3/4 of the trials when they already received three negative feedbacks. What was the frequency of these trials overall? Wouldn't the models and intuition on the task structure predict a monotonic increase of switch probability as a function of accumulating, consecutive non-reward trials?
3. What was the reasoning behind the change of response procedure in Experiment 2? Did the fact that the cursor always moved from the left to the right induce any response bias? Reaction times for left responses must have been substantially shorter. Could this have had any effects on how the task was performed and perceived by the animals?
4. For the two experiments, two different groups of macaques were trained. Was there any specific reason to do so? Could the fact that the animals in Exp. 2 knew only the correlated version of the task have influenced their learning of the task structure and their performance? At the very least, assuming that there was no task cue in Exp. 1, uncertainty about the structure was conceivably much higher in the beginning of each session in Exp. 1 compared to Exp. 2.
5. Regarding the modeling, how was the hypothesis space for the reversal probability (H) determined? Was the 0.3 based on the maximal hazard rate that an animal could encounter during testing?
Related to this, why did the number of reversals vary between 1-7 in the correlated environment? Was reversal occurrence depending on the performance in the uncorrelated task and how many reversals did occur in the uncorrelated environment? Did the mean reversal rates differ between the correlated and uncorrelated variants, and, if so, could that have confounded the results?
6. Lines 229-232: It remained unclear how the reversal estimate could be applied to uncorrelated sessions. Further above it is stated that model 1 showed a better fit to the uncorrelated task data.
7. Did the model compute inferences separately for the two response options? From the description of the model, it is not clear how the expected value of reversal (or expected value of the unchosen option) is derived.
8. How was the model fit reported in Figure 2F and on page 5 line 175 determined. Was the model set fit to the choice behaviour of the animals or is this the fit of the logistic regression based on parameters extracted either from model 1 or 2?
9. In the description of Exp. 2 the reader may get the impression that fMRI was recorded as well and be surprised to see no results from that (see line 591). Careful rereading then made clear that the animals were only tested behaviourally after TUS. It might be helpful to clarify this more explicitly to avoid sending readers on a wrong garden path.
10. Some details on the TUS methodology seem to be missing.
 - How long was the interval between stimulation and task (approx. in min)?
 - Can you specify how the pseudo-CTs were created? Depending on the parameter value, the skull estimation can be more or less conservative, leading to different values on the thermal and acoustic simulations.
 - The Isppa (80W/cm²) is quite high compared to previous studies (60W/cm²) such as Mahmoodi et al. (Neuron, 2023) . Please specify the results of the acoustic and thermal simulations.
 - Line 1099: "Each stimulation condition was repeated 5-8 times for each animal and each test took place on a different day." Why not a fixed number?
11. In the introduction (line 107), it would be more informative to justify the use of TUS some lines above, instead of jumping directly to the findings. It might also be mentioned that TUS is not in all cases disruptive but that certain protocols could be facilitating.
12. There was a significant difference between sham and stimulated regions. But was there really a significant difference between the regions in any of the factors?
More generally, did TUS actually impair overall performance in the task (reflected, e.g., in lower accumulated reward)?
13. Line 1050. "The upsampled data was then epoched to 6s time windows spanning 1s before to 5s after..." – this contrasts with the figures showing 10s epochs, spanning 5s before to 5s after the event of interest.
14. Line 553: after discussing the TUS results, the results of the models were included in the discussion. I would suggest to move it right after the fMRI discussion. This way you would have a discussion relating the correlational results and ending

with the causal relation of the regions and behavioural outcomes.

15. Moreover, a schematic figure or a radar plot for visualising the regions which are connected to each other and what they represent would be helpful for summarizing the findings.

16. fMRI figures should contain color bars.

17. Fig. S1 is not correctly reproduced in the MS.

(Remarks on code availability)

The code was not available. The link to OSF did not reveal the code, because it is protected.

Reviewer #3

(Remarks to the Author)

This study investigates two reversal tasks in macaque monkeys. In both tasks, the animals make a choice between two identical-looking stimuli at each trial. In the correlated task, reward probability of the two stimuli is negatively related (when the reward probability of stimulus A is high, that of stimulus B is low and vice-versa). By contrast, in the uncorrelated task, the reward probabilities of the two stimuli are not correlated and fluctuate independently of each other. The authors then used fMRI to measure brain activity and built a Bayesian model to estimate the underlying latent belief states (note that the correlated but not the uncorrelated task contains distinct states). The Bayesian estimate of reversal probability correlated with activity in multiple regions, including dmPFC and hippocampus. Moreover, dmPFC and thalamus differentially encoded chosen and unchosen reward probability. The authors then used transcranial ultrasound stimulation (TUS) to disrupt dmPFC, hippocampus, or thalamus activity and report that TUS at all three regions reduces the impact of the reversal estimate on behavior.

The work presented in this paper is relevant for the reversal task literature. By using Bayesian estimation methods to track the latent state structure as estimated by the animals, the authors could show which areas are important to track latent states, as opposed to the simpler interpretation in terms of inhibition by frontal areas. However, multiple findings are brittle and currently not borne out by appropriate statistics. Moreover, several methodological issues limit the impact of the work.

Major Comments

#1 It is unfortunate that, and unclear why, all the monkeys performed the correlated task first. As the authors allude to, this could have affected beliefs in the uncorrelated task, e.g., that the reversal of one option meant the other option was now more rewarding. Why not randomize task order? More detailed explanation of this design choice is needed. Moreover, it is not clear whether the 8 sessions were consecutive for each type of task or intermingled. It would be important to state this clearly.

#2 (Methods). For the fMRI analyses, the authors used cluster-level correction with a cluster-inducing threshold of $p < 0.01$ ($z = 2.3$). However, as has been shown before (Eklund et al., 2016, PNAS), $p < 0.001$ ($z = 3.1$) should be used as cluster-inducing threshold. Should this not work and a completely ROI-based be necessary, then the next point becomes even more important.

#3 ROIs (lines 85-87): The authors mention that there was a recent shift away from the OFC in the reversal literature. This alone however does not sufficiently specify the choice of ROIs. There should be a more explicit link between which areas will be of interest for the current study, by showing literature that clearly shows the role of the ROIs (dmPFC, medial dorsal thalamus, and mid hippocampus) specifically for reversal (estimates or other relevant model parameters).

#4 (Results). The authors (paragraph starting at line 440) suggest that TUS had specific effects for different brain regions. To bear this out statistically, a region-by-effect interaction analysis is required. If this is not significant, the interpretation should be toned down.

#5 (Methods). Given the task design with two stimuli, the variables characterizing reversal apply also to the unchosen option (in the correlated task). Accordingly, the question arises whether the findings portrayed as related to reversal indeed concern reversal or apply to unchosen options more generally. This point should be addressed more explicitly than in currently the case.

#6 (Methods). Given the auditory (e.g., Braun et al., 2020, Brain Stimul.) and potentially other confounds of active transcranial ultrasound stimulation, it seems insufficient for the control (sham) condition to use no stimulation at all. This is relevant also in the present offline protocol, because like the stimulation effects, the auditory effects may gradually subside with time.

Minor Comments

#1 (Methods/Intro). Some details of the methods, which are necessary to understand the results, are found only in the Methods section. The authors might consider providing some more information, e.g. an intuition on models 1 and 2, so that the results section can be read without having to continuously jump to the methods section.

#2 (Results). In the correlated task, a reversal means that the option which was previously most rewarding becomes the least rewarding and vice-versa. In the uncorrelated task, however, it means that either one of the options which was previously rewarding, is now not rewarding (and the opposite is also true)? How is the reversal specifically defined? Presumably “when the probability of reward goes from below (above) 0.5 to above (below) 0.5”? The authors should better specify this definition, which can only be found in the Methods section. Also, what are the possible latent states in the uncorrelated task?

#3 (Results). In the text, it is mentioned that figure 2F shows the AIC for both models for both sessions. It is not clear which is which, as model1 and model 2 are not mentioned in the figure (line 176).

#4 (Results, Methods). Based on Fig. 1d, bottom, it appears that overall, the animals switched more often in the uncorrelated than in the negatively correlated task even though session numbers were similar. Was the total number of trials the same in the two tasks? If not, why not? If yes, was the difference in overall switching frequency adaptive and how can it be reconciled with the lower switch rates in the uncorrelated experiment as shown in Fig. 1d, top? Also, how many trials entered the imaging analyses? Finally, Lines 648-649 state that switching became more likely as multiple errors accrued and refers to Fig. 1D to back the statement up. However, Fig. 1D suggests that if anything, the statement only holds for two consecutive errors, not for three.

#5 (Results). In Fig. 4, the authors analyze multiple timepoints, without providing a clear rationale and without correcting for multiple time points. This needs to be addressed properly.

#6 (Methods, Results). Seven animals were used; however, it remains unclear whether the behavior was similar across animals within experiment.

#7 (Discussion). Line 518-519 highlights the “striking” nature of differential reversal estimate encoding between correlated and uncorrelated tasks. However, only the hippocampus seems to have shown such a difference at the statistical thresholds used, and when testing the correlated task alone, the hippocampus did not seem to significantly encode reversal estimate at the same timepoint. If so, the statement should be qualified and toned down.

#8 (Discussion). dmPFC and hippocampus apparently tracked latent states in the uncorrelated task. However, one should not see such a correlation if reversal estimates are not supposed to exist in the task. The authors could discuss these points in more detail, as well as add suggestions on how future research could go about to interrogate this unexpected activity.

#9 (Introduction). When using Brodmann area notation (Area 47/12o, line 76), it might be useful to mention which area this refers to as a clarification but also for readers who are less familiar with Brodmann area notations.

#10 (Methods). Function 5) in the bottom line, V2 instead of V2. Same in lines 739-740. Line 777: Liebler instead of Leibler.

(Remarks on code availability)

Reviewer #4

(Remarks to the Author)

(Remarks on code availability)

The code was not available

Version 1:

Reviewer comments:

Reviewer #1

(Remarks to the Author)

All of the concerns I previously raised have been adequately addressed and satisfactorily resolved by the authors.

(Remarks on code availability)

N/A

Reviewer #2

(Remarks to the Author)

The authors revised the manuscript in a remarkably thorough and comprehensive fashion. All points were addressed well and there are no issues left.

I have a very minor question and I would like to leave it to the authors whether they want to include a response into the manuscript: The delay between sonication and the task was quite short. Recent TUS work suggests that first TUS effects are rather local, whereas later effects can be found in the entire network. Are there any hints in the data that the TUS effects changed over the course of the experiment?

(Remarks on code availability)

While OSF says that files are accessible, I can't find any. Perhaps this has to do with the recent service disruption event at OSF.

Reviewer #3

(Remarks to the Author)

Thank you for a responsive and thorough revision. My previous major point #3 needs a bit more care: There are in principle any number of regions other than hippocampus, thalamus and dmPFC that could play a role in reversal outside OFC. There should be a clear, literature-based (including references) argument for why exactly these three and not other regions are the most likely candidates. This is not provided at the moment. Alternatively, if all regions do survive whole-brain correction (as the authors argue), such a motivation would not be necessary, the identification of regions outside OFC would be data driven rather than based on ROIs and the procedure thus could be clarified more explicitly rather than in the current half-way manner.

(Remarks on code availability)

Reviewer #4

(Remarks to the Author)

(Remarks on code availability)

Reviewer #1 (Remarks to the Author):

This study demonstrates that flexible task control in primates relies on interactions among multiple brain regions, each playing a distinct role. By conducting experiments with macaques, the authors illuminate how three critical areas in the prefrontal cortex, hippocampus, and thalamus cooperate to anticipate rule changes (reversals) and evaluate choice value. They also reveal functional specialization within these regions—most notably, the dorsomedial prefrontal cortex (dmPFC) encodes the probability of reversals to guide behavior, while the hippocampus tracks updates related to these reversals. Furthermore, by utilizing transcranial-focused ultrasound stimulation (tFUS), the authors provide causal evidence that each of these areas contributes uniquely to flexible switching based on reversal estimates. Altogether, this work illustrates how a distributed neural network in the macaque brain organizes and allocates processes essential for flexible decision-making and learning. Although the task used is relatively simple and the computational framework for analyzing decision-making is mathematically complex, the authors convincingly show how these three regions specifically engage in reversal learning and inferential processes that underpin decision-making.

Below are four major issues that, if addressed, could make the research even more systematic and robust.

We thank the reviewer for their interest in the manuscript and the positive comments that they have made. We are also grateful for the comments that the reviewer has made that have us to see where the manuscript might be improved. We have replied in detail to the reviewer's comments below and made a number of changes to the manuscript.

Major Comments

1. Comparison between “correlated” and “uncorrelated” tasks

While the authors show that both “correlated” and “uncorrelated” tasks share a basic target-switch response after no-reward outcomes (Fig. 1d), Fig. 1b indicates that monkeys persist with the same choice longer in the “correlated” task than in the “uncorrelated” task. Consequently, they employ distinct computational models for each condition (Fig. 2E) to accommodate these differing strategies. However, the markedly different reversal frequencies suggest that the “uncorrelated” task may not serve as a straightforward control for the “correlated” task.

Two key questions could make this comparison more compelling:

(1-1) What if the authors increased the reversal frequency in the “correlated” task?

If the resulting behavior begins to resemble that of the “uncorrelated” task, it would highlight continuity between the two tasks, suggesting that each model represents a different position on a broader spectrum of learning strategies. Demonstrating such a shift would strengthen the rationale for using the “uncorrelated” task as a control and clarify why two distinct models are necessary.

First, one quick point to note is that where the reviewer says that “monkeys persist with the same choice longer in the “correlated” task than in the “uncorrelated” task”, in fact the opposite is the case: the monkeys persist with the same choice for longer in the uncorrelated task than in the correlated task (figure 1d). We mention this point just to avoid any ambiguity and realise that this may be a typographical error.

Turning, next, to the core points that the reviewer is making, it seems that there are two key aspects that might be addressed. Perhaps, the key question is whether it is possible to compare the two tasks – correlated and uncorrelated – on a continuous measure or spectrum. Potentially, a subsidiary question is can this aim be achieved by thinking about reversal in a more continuous way. We are very sympathetic to the first of these points – can we compare the tasks in a quantitative and not just a qualitative manner and we have revised the manuscript in an attempt to do this and hope that we address the reviewer’s concern with our revisions. However, while we understand why the reviewer suggests that this might be done by increasing the reversal frequency, we argue that this might not be the best way to achieve the important goal that the reviewer has suggested of comparing the tasks in a quantitative way.

We understand why the reviewer focusses on the reversal frequency as a key difference between correlated and uncorrelated tasks and we return to the issue of reversal frequency in the next response to next point made by the reviewer. Here, however, we focus on the fact that reversal frequency alone does not capture the difference between the tasks. A key difference is the negative correlation between the reward probabilities for the two choices in the correlated task which means that there are limited number of latent states defined by the relative reward probabilities of the two choices but this is not the case in the uncorrelated session. It is for this reason that the monkeys do not “persist with the same choice longer in the “correlated” task than in the “uncorrelated” task” but do the opposite; if they find an option is unrewarded in the correlated task then the chances are that the latent state that prevails is the opposite one than had been hypothesized.

However, if the key question is whether it is possible to compare the two tasks on a continuous measure or spectrum then we think that there may be a way to do this. This is something that we had actually sought to do in the first draft of our manuscript but we may not have done so as clearly as we had hoped. We argue that rather than thinking about whether the correlated tasks could be “sped up” so that it resembles the uncorrelated task perhaps it is better to consider the complement – whether the uncorrelated task could be considered in the same light as the correlated task and analysed in the same way. We think that this is an important idea and the way that it can be addressed is to consider whether macaques might seek latent states in the uncorrelated task. We can estimate what reversal estimates the macaques might have by fitting model2 to the uncorrelated task data to estimate the beliefs the macaque might hold, including about reversal, on each trial given the experiences it has had. The model can, therefore, be applied to the uncorrelated task just as it can be applied to the correlated task (even though model 1 is a better fit to the uncorrelated task). Even in the uncorrelated task, an animal might hold beliefs about a reversal that we can estimate. This means that if neural activity were to be found that was correlated with the reversal estimate in the uncorrelated task, then that activity would be related to an estimate of the reversal probability that the animal had formed even though that estimate might not be appropriate for that task. We can, similarly, also look at activity related to the change in the reversal estimate that occurs at the

time of reward/error feedback at the end of each trial which we refer to as reversal estimate D_{KL} or simply as reversal D_{KL} . We found activity related to reversal D_{KL} in one part of dmPFC even in the uncorrelated task (red bars on the left hand sides of figures 4D, E). When animals receive feedback, they may transiently attempt to update a reversal estimate even in the uncorrelated task but, unlike in the correlated task, this does not result in a sustained pattern of activity that is related to a reversal estimate and that is still present at the beginning of the next trial (right hand sides of figures 4D, E). Moreover figure 5 suggests that this might be because, in the uncorrelated task, but not in the correlated task, macaques update an estimate of something akin to an option specific update (ie not an estimate about the probability of a reversal in latent state but an estimate of the change in one specific choice). Such option-specific updates versus latent state updates, however, maybe be yet more part of the way in which we can identify clearer points of comparison between the uncorrelated and correlated tasks. In the revised manuscript we have attempted to address these issues by making a series of changes as follows:

One possibility is that macaques may attempt to infer latent task states even in the uncorrelated task even if none are present. We can estimate what estimates animals themselves might have about such states and the probability of switching between them – the reversal estimate – in the uncorrelated task by applying model 2 to the uncorrelated task data. By doing this we are estimating what beliefs the animals might hold at any point in time given the choices that they have made and the feedback – rewards and errors – they have observed even if those beliefs might be erroneous given the nature of the reversal task. Because the reversal estimate is a continuous rather than binary variable it may capture a weak or partial belief that the animal has about a latent state reversal, although if such beliefs are held in the uncorrelated task the reversal estimate may often be high. We can then examine whether these estimates are predictive of actual switching behaviour or of neural activity even in the uncorrelated task. In this way we can compare behaviour and neural activity in the two tasks more directly.

...The reversal estimate is, however, only furnished by model 2 and so we employed the reversal estimate from model 2. As explained above, this means that we are able to compare the strength of influence the reversal estimate might have on both behaviour and brain activity...

... Importantly, the animals were more likely to switch when the reversal estimate – model 2's estimate of the probability of reversal between one latent state and another – was high (Fig.3, right; $\chi^2_1=32.1149$; $p<0.001$) and this effect was greater in correlated sessions ($\chi^2_1=24.7756$; $p<0.001$). Even if animals might seek to identify latent states in the uncorrelated task and transitions between them, our estimates of when they might be doing so, derived from application of model 2 to the uncorrelated task suggests that they have significantly less influence in the uncorrelated task.

Analogous analyses can also be performed with the full set of five factors (chosen probability, unchosen probability, chosen uncertainty, unchosen uncertainty, reversal estimate) using LMEM2; (Supplementary Fig.S2B). These analyses confirmed that the animals' reversal estimates drove their switching behaviour more in correlated sessions even when each option's values and uncertainties were considered separately ($\chi^2_1=28.8443$; $p<0.001$) again confirming that such beliefs had a bigger impact on behaviour in the correlated task.

If the animals search for latent task states and reversals between them in the uncorrelated task then by applying model 2 to the uncorrelated task it is possible to estimate what estimates the animals

themselves might have about reversal even if these are weak and partial. We were, however, unable to find significant evidence for activity linked to reversal estimates in the uncorrelated task when we examined activity across the whole brain. This was not simply a consequence of the conservative nature of the whole brain analysis approach, because ROI-based analyses similarly revealed no evidence for reversal effects in the uncorrelated task in dmPFC or hippocampus (all $r < 0.406$; all $p > 0.3$); even though the hippocampus and dmPFC carry reversal estimate-related activity in the correlated task, they do not do so in the uncorrelated task.

While any reversal D_{KL} -related activity in hippocampus was restricted to the correlated task, intriguingly, D_{KL} activity in anterior dmPFC was statistically greater than zero even in uncorrelated sessions (Fig.4E; $t_{34}=2.315$; $p < 0.013$). It is possible that this may reflect animals attempting to identify latent task states and transitions between them even in the uncorrelated task. We return to the question of D_{KL} -related activity in dmPFC below to show that, in the uncorrelated task, the animals may possibly do this at the time of feedback, but any attempt to do so does not result in sustained neural activity at the onset of the next trial. The activity may actually reflect tracking features of individual options as well as latent state reversal (discussed further below in Fig.5).

Activity in dmPFC and hippocampus reflected updates in reversal estimates – reversal D_{KL} . It is possible that that this may reflect the animals trying to use feedback to infer a latent task or the possibility of a transition between latent task states. If this is the case, then it does not seem to result in sustained neural activity related to the reversal estimate by the beginning of the next trial (note the absence of positive red bars on the right hand side of figure 5D, E, and F). However, it is possible that this might simply be because dmPFC and hippocampus were actually tracking some other update-related variable that resembled reversal D_{KL} . We therefore employed new versions of models 1 and 2 that yielded separate update estimates for each option. For models 1 and 2 this was under the assumption that options' reward probabilities were uncorrelated or negatively correlated, respectively (Methods, *Model Estimates*). In other words, the models yielded estimates of how much an individual option's associated reward probability was to be updated when the outcome arrived after it was chosen as opposed to an estimate of whether there was likely to be a transition between latent states. We reran the fMRI analyses in the anterior dmPFC and hippocampus ROIs shown in figure 4 but now included both reversal D_{KL} and individual option-specific update term (as appropriate for correlated and uncorrelated sessions) and allowed them to compete to explain activity. Two important results emerged. First, reversal D_{KL} continued to explain significant variance in outcome-locked activity in correlated sessions in both dmPFC and hippocampus (Fig.5, left, dmPFC: $t_{34}=2.003$, $p=0.027$; right, hippocampus: $t_{34}=1.988$, $p=0.027$). Second, the picture was very different in uncorrelated sessions in hippocampus (Fig5B, right); hippocampus showed no evidence of tracking option-specific updates in the same way as reversal D_{KL} . In anterior dmPFC in the uncorrelated task, reversal D_{KL} and option-specific update both explained less variance than they had in correlated sessions and in neither case did the effects reach significance although they were marginal in one case (reversal D_{KL} : $t_{34}=1.687$, $p=0.051$). In summary, dmPFC tracked reversal D_{KL} updates but did so most clearly when the task contained latent structure as was the case in the correlated task. Anterior dmPFC may possibly have estimated, or attempted to estimate, reversal D_{KL} even in uncorrelated sessions, perhaps as a consequence of experience of its utility in the correlated sessions which the same individuals performed. Instead in the uncorrelated task, dmPFC activity, if anything, reflected the updating of values of the individual options. However, by contrast, it was very clear that hippocampus was only concerned with tracking of reversal estimates for latent task states, and not option-specific updates, and that it did so only in the correlated task where there was latent structure.

(1-2) Would parameters from Model 1 converge toward those of Model 2 if reversal frequency

were increased?

Assuming that both tasks lie along a continuum, it is equally important to examine whether their underlying computational models might converge. If increasing the reversal frequency makes the monkeys' behavior in the "correlated" task more akin to that in the "uncorrelated" task, do the parameters in Model 1 gradually align with those from Model 2? One approach could be to examine such convergence via model selection procedures (e.g., comparing AIC or BIC) to see whether the "correlated" task's parameter estimates transition toward those of the "uncorrelated" task's model under higher reversal frequencies. If direct experimentation is challenging, at least a discussion of how these two tasks and models relate to one another would help clarify the broader framework in which they both operate.

We understand why the reviewer notes that the frequency of behavioural reversals is higher in the correlated condition, and that this could suggest that monkeys estimated a higher volatility level in the correlated condition, which could offer an alternative explanation for the observation that macaques reversed their choice after fewer missed rewards in the correlated condition.

Nevertheless, while it is true that there are more frequent reversals (switches) in behaviour in the correlated session, this does not imply that the programmed schedules also reverse more frequently in the correlated task than the uncorrelated task. In fact, the frequency of reversal in the uncorrelated sessions cannot even be estimated because the option values do not reverse per se. Instead, the options' value, by definition, vary independently. Therefore, there is no direct way to match the 'reversal frequency' between task schedules. What we can do, however, is ask what would the macaques believe is happening in the uncorrelated task if they were to treat in the same way as the correlated sessions: we can ask what would they estimate is the reversal frequency in the uncorrelated sessions. Would they estimate that the reversal frequency was higher in the uncorrelated sessions. The answer turns out to be negative (see figure below); contrary to what seemed to be a reasonable intuition from the reviewer, the animals should actually, on average, estimate that the reversal frequency is higher in the uncorrelated session. The counter-intuition, to understand why this happens is because, not only is it important to consider, as the reviewer intuited, when the reversal estimate is high, but also it is important to estimate when it is low and often the reversal estimate in the uncorrelated sessions is not low; in other words, often in the uncorrelated sessions the animals are not certain that things are not changing.

This is summarized in the figures below which show the average reversal estimate across animals and sessions in uncorrelated (blue) and correlated (red) cases (apologies that the colour code here is opposite to our other figures).

In summary, as described in the response above, we address the reviewer’s point by asking whether the macaques’ behaviour in the uncorrelated task could be fit by the correlated model. Although we note that the correlated model is a worse fit to behaviour in uncorrelated sessions than the uncorrelated model (fig 2f), we note that if we do fit the correlated model, model 2, to the uncorrelated sessions we can, nevertheless, observe that the average fitted estimate of beliefs about reversal frequency (reversal estimate) across the session is higher in the uncorrelated sessions. That is, a macaque that used the hierarchical/correlated model even in uncorrelated sessions should estimate that the environmental volatility in these sessions was, on average, higher than in the correlated sessions. This is clearly at odds with the argument that macaques may have reversed choices after fewer trials in the correlated condition due to a higher reversal frequency/volatility in that condition. It is for this reason that in the text added in response to the previous question that we stated “Because the reversal estimate is a continuous rather than binary variable, it may capture a weak or partial belief that the animal has about a latent state reversal, although if such beliefs are held in the uncorrelated task the reversal estimate may often be high” as shown below.

One possibility is that macaques may attempt to infer latent task states even in the uncorrelated task even if none are present. We can estimate what estimates animals themselves might have about such states and the probability of switching between them – the reversal estimate – in the uncorrelated task by applying model 2 to the uncorrelated task data. By doing this we are estimating what beliefs the animals might hold at any point in time given the choices that they have made and the feedback – rewards and errors – they have observed even if those beliefs might be erroneous given the nature of the reversal task. Because the reversal estimate is a continuous rather than binary variable it may capture a weak or partial belief that the animal has about a latent state reversal, although if such beliefs are held in the uncorrelated task the reversal estimate may often be high. We can then examine whether these estimates are predictive of actual switching behaviour or of neural activity even in the uncorrelated task. In this way we can compare behaviour and neural activity in the two tasks more directly.

2. Bayesian Model and Neural Decoding Consistency

The authors employ a computational framework (e.g., Bayesian or reversal-learning models) to derive parameters that they then correlate with neural activity. The strength of this work depends

on how these model-derived variables are defined, validated, and interpreted. Two points warrant careful attention:

(2-1) Multivariable Regression and Multicollinearity

A common approach is to use multivariable regression to find neural correlates of computational parameters. However, such methods can face interpretive challenges—especially regarding multicollinearity. Because parameters like chosen probability, unchosen probability, chosen uncertainty, unchosen uncertainty, and reversal estimate may interact or correlate, the authors should demonstrate that each parameter is a legitimate reflection of an underlying neural process. They should also clarify how they have addressed or minimized multicollinearity. Providing explicit criteria for parameter selection—and a sound rationale for assigning each parameter to decode specific brain regions—would further validate the manuscript.

The reviewer is indeed correct that a regression-based analysis requires there to be no collinearity between the regressors that are used. Moreover, we would emphasize that this must be considered from the very initial design of the project and not just after the data are collected. This is precisely what we have done. An important way in which this was achieved was by making the reward associations of the two options probabilistic rather than deterministic and by ensuring that, even in the correlated task, the reward probabilities of the two options were not perfectly negative correlated. A key section of the manuscript is the following one that has been updated following the reviewer's comments.

Reward associations in both the correlated and uncorrelated tasks (referred to as “correlated” and “uncorrelated” in the figures) were probabilistic rather than deterministic. In general, however, in the correlated task, when one option had a high probability of reward, then the other option's probability of reward was low and close to zero (Fig.1B). Although the correlation between the reward probabilities associated with the two options was not exactly $r = -1$ in the correlated task but averaged approximately -0.798 across all schedules used, this correlation was consistently more negative than in the uncorrelated sessions (Fig.1C; **the bars indicate group mean performance but data for each individual session are plotted as circles and the circles in each column in this figure and elsewhere are the data for each of the four individual macaques**). In the uncorrelated task, both options changed their reward probabilities independently of each other, with the correlation between them close to zero. By employing probabilistic rather than deterministic schedules, it was possible to limit correlation between the key explanatory variables, choice values, choice uncertainties, and reversal estimate from the Bayesian model which, as explained below, account for the behaviour and neural activity recorded (Supplementary Fig.S1). **Ensuring that correlation between variables is eliminated in this way is a prerequisite for ensuring an absence of collinearity between regressors, which would be problematic, in the regression analyses that we used to examine behaviour, neural activity, and the impact of TUS.**

There is one point at which the issue of collinearity between regressors become especially important. This concerns the analysis performed in Supplementary Figure S4. In this supplementary figure we consider how, given that they infer a reversal in latent state might they also infer a precise value for the unchosen option. They might do this if they exploited the knowledge that the values of the two choices were negatively correlated with one another; if they observed that the chosen value decreased, might they infer that the value of the unchosen option increased. However, under this assumption – that the values of the two choices are negatively correlated with one another – their value estimates become collinear. Under this assumption, we cannot look for activity separately

related to the value of the chosen option and the value of the unchosen option but we can only look for activity related to the decision variable – the difference between the value of the chosen and unchosen options. This is what we do in Supplementary Figure S4. The blue bars in Supplementary Figure S4 should therefore approximately correspond to the difference between the values of each pair blue bars in Figures 6C-D where we can look separately at the value of the chosen option and the value of the unchosen option. We have revised our discussion of this section of the manuscript to ensure that there is no confusion about whether we have avoided collinear regressors even at this point. The following section has been added to the Results:

Our central interest in this study was to look at whether animals might infer a latent state and, if so, how they did this. The same piece of information – the feedback on each trial – informs the animal about both its estimate of the value of the choice it is taking and the reversal estimate. Given that they infer a reversal in latent state, there are two ways that they might then establish a value for the unchosen option. First, having inferred that the state has changed, they might move into an exploratory mode and try out the other response and learn by observation if the other choice, choice 2, is now rewarded. Alternatively, they might make a second type of inference; in addition to inferring the change in latent state, they might infer a precise value about the value of the unchosen option on the basis that the values of the two choices were negatively correlated with one another; if they observed that the chosen value decreased, they might infer that the value of the unchosen option increased. We therefore used a second version of Model 2 that incorporated the assumption that this second type of inference about the unchosen option might be made in the analyses shown in Supplementary Figure S4. Anterior and medial thalamus and dmPFC activity reflect information about choice values in the correlated sessions even under this alternative assumption.

To further address this point, we have also made the following changes to Supplementary Figure S4.

S4. Decision-related activity estimated under the assumption that monkeys make inferences about the value of unchosen options from their experience with chosen options. Given that animals infer a reversal in latent state, there are two ways that they might then establish a value for the unchosen option. They might try out the unchosen response and learn by observation what its value now is. Activity related to the chosen and unchosen values under this assumption is illustrated in figure 6. Alternatively, animals might make a second type of inference; in addition to inferring the change in latent state, they might infer a precise value about the value of the unchosen option on the basis that the values of the two choices were negatively correlated with one another; if they observed that the chosen value decreased, they might infer that the value of the unchosen option increased. Here, we therefore used a second version of Model 2 that incorporated the assumption that this second type of inference about the unchosen option might be made. In other words, If the animals learn from experience that one option that they have chosen recently has a probability, p , of being rewarded then, because the options' reward probabilities are negatively correlated, they should infer that the unchosen option's probability of reward is $1-p$. Decision-related activity in dmPFC (A) and anterior and medial thalamus (B) is estimated under the assumption that monkeys make such inferences in correlated sessions. Because the regressors for the two options are now perfectly negatively correlated with one another it is not possible to look at neural activity that is specifically related to one option or the other (because activity that appears positively related to the value of one option is equally negatively related to the value of the other option) but only to look at activity related to the difference in the options' reward probabilities (chosen option's reward probability – unchosen option's reward probability) and so this is what is illustrated here. The blue bars in this figure should, therefore, approximately correspond to the difference between the values of each pair of blue bars in Figures 6C-D where we can look separately at the value of the chosen option and the value of the unchosen option. This is, indeed, the case.

There is also a point in the Methods where perhaps greater emphasis needs to be given to a lack of correlation between some important variables as shown below:

Note that while equation 8 summarizes the estimated reward value to the animal of switching and taking the alternative action, our focus in both the behavioural and neural analyses is on how animals track the probability of reversal, which we refer to as the reversal estimate, and then separate variables track the reward values of the chosen option and the unchosen option (which are the estimates of the probabilities of reward for the chosen and unchosen option respectively) and the uncertainties of those estimates. The reversal estimate and the choice value variables used in our analyses were uncorrelated (Supplementary Fig. S1).

(2-2) Arbitrariness of Model-Derived Parameters

While the authors propose seemingly appropriate variables for capturing relevant computations, such parameter choices can be somewhat arbitrary. Justifications or evidence are needed to show that these parameters are both necessary for their computational goals and plausible from a neural coding perspective. For example, the authors could explain why these particular parameters—rather than alternative formulations—best capture the behavioral and neural data and how they align with or extend established decision-making models. Demonstrating that each parameter plausibly mirrors a distinct neural function would substantially strengthen the link between these computational variables and their neural correlates.

We contend that the variables that we have used to capture the relevant computations are indeed the minimal set that might be employed. Ultimately the task that the animals are performing is a decision-making task in which they must select one choice rather than another. Decision-making should be guided by a variable and the important decision variable is how much better is one choice than the other. We therefore examine behaviour, neural activity, and the impact of TUS by examining the influence of the estimate of this decision variable – the difference between the probability of reward associated with the chosen option and the unchosen option – on neural activity. We usually illustrate its impact in terms of the separate components relating to the probability of reward for each of the two choices. Because the probabilities of reward associated with the two options are estimates, there are uncertainties associated with these estimates and so our other variable is the difference in the uncertainties associated with the estimates of reward probability associated with the options. Finally, the focus of our manuscript is whether an additional third variable – reversal estimate – also explains variation in behaviour. We hope that the following revised section of the manuscript makes the rationale clearer.

Ultimately the task that the animals are performing is a decision-making task in which they must select one choice rather than another. The important decision variable is how much better is one choice than the other. We therefore considered the difference between the probability of reward associated with the chosen option and the unchosen option – on neural activity and the difference between the uncertainties associated with the estimates of reward probability that the animals might make. Finally, and a key focus for the study, is whether an additional third variable – reversal estimate – also explains variation in behaviour. We developed two Bayesian models to track the features of the correlated and uncorrelated tasks (Methods: Bayesian modelling for correlated environments; Fig.2A). Both models tracked belief estimates of reward probabilities, and the associated uncertainties of those estimates, and thus belief estimates that macaques might optimally hold given the outcomes (reward/non-

reward) observed following their choices (Fig.2B). The beliefs about the values of each option were expressed as a probability distribution and updated according to Bayes' theorem, by multiplication of the prior belief at the beginning of each trial and likelihood of the outcome observed at the end of the trial to produce a posterior belief. While these were the only task parameters tracked by model 1, model 2 also tracked the probability of a task state reversal (Fig.2C, D). Beliefs were quantified by a joint probability density function (pdf) over a pre-defined hypothesis space (Figure 2C). When it estimated that the animal was transitioning to a different latent task state, it quickly revised the probability and uncertainty estimates of the choices that animal took each time feedback (reward or non-reward) arrived after making a choice. Thus, the reward probability estimates for the choices and the uncertainties on those estimates in model 2 were informed by the reversal estimate. In the correlated environment, options' contingencies are partially negatively correlated with each other such that if one option has a high reward contingency, the other option is likely to have a low reward contingency (Figure 1A, Figure 2A, B) so that a value of an option – its probability of reward – tends to change in sudden jumps (from high to low or vice versa). In model 2 the subjective estimate of the probability of a reversal influences the estimates of choice value and helps to ensure the sudden changes in choice values that accord with the ground truth of the situation in correlated sessions. In other respects, however, the reward probability and uncertainty estimates furnished by the models were similar in nature (Fig.2E). A separate but related feature that might be incorporated into Model 2 is a capacity to infer an unchosen option's value from feedback obtained from the other, chosen option. We also consider such a capacity for inference below (Fig.S4). Even without inclusion of this extra feature, however, Model 1 provided a better account of the majority of uncorrelated sessions, while model 2 provided a better description of the majority of correlated sessions; the difference in Akaike's information criterion (AIC) scores for Model 2 minus Model 1 are shown for a comparison of the two models for each session (Fig.2F).

An aspect of our report that is perhaps easy to overlook and which was perhaps not presented with the greatest clarity concerns the model comparisons presented in Figure 2F. As noted in the previous excerpt from the manuscript we have clarified that this result relates to a comparison of different models of behaviour. We have also made changes to the legend for Figure 2F too:

(F) In general, the animals' behaviour in uncorrelated and correlated tasks was explained better by models 1 and 2 respectively. Individual grey dots indicate the AIC difference between Model 2 and Model 1 derived from fitting the choice data of each session and sessions from the four animals are shown in four columns as in figure 1c.

3. Consistency With Prior OFC Research

Previous literature has shown that orbitofrontal cortex (OFC) lesions can compromise reversal tasks, emphasizing OFC's role. Nevertheless, this study—and related work—indicates that other areas connected to OFC also contribute. Here, the authors identify specific regions involved in a “correlated” task with a reversal component, applying whole-brain analysis and tFUS to confirm causal involvement. However, given OFC's established causal role in simpler reversal paradigms, the authors should clarify how their new task design diverges from previous ones. For example, do lower reversal frequencies or probabilistic structures (as opposed to deterministic ones) modulate OFC engagement? The authors should compare their methodology with earlier protocols and discuss whether the identified circuit is uniquely tied to probabilistic or latent state-based reversals, whereas OFC might be more pivotal in simpler or more discrete reversal tasks.

We had attempted to do this at the very beginning of the Discussion. Given the reviewer's comments, we have revised this part of the Discussion further as shown below and we would be happy to revise it even further if it is not clear. Perhaps a key point to emphasize is that one of the areas that we discuss, area 47/12o, is actually a region that may be carrying out a key function linked to the OFC. Moreover, it is actually immediately adjacent to the OFC and is sometimes even treated as a part of the OFC. We hope that by making the following revisions to Discussion we are able to address the reviewer's concerns.

Previous studies have recorded activity in OFC and in adjacent regions, such as area 47/12o on the border between the OFC proper and the ventrolateral prefrontal cortex, when animals are learning how likely choices are to lead to rewards¹⁻³. OFC activity also reflects choice reward probability and other aspects of choice value and choice comparison during decision making⁴⁻⁷. While lesions or manipulations of OFC lead to alterations in value-guided decision making^{6,8,9}, disruption of 47/12o causes failures of credit assignment – a diminished ability to learn which choices caused which outcomes^{9,10}. The deficit makes it especially difficult to establish which one of several choices leads to a reward when multiple choices are interleaved with one another because the credit for any reward is assigned not just to the choice that actually led to it but to other choices made at adjacent points in time^{11,12}. Area 47/12o disruption, therefore, makes it difficult for animals to perform reversal tasks because at the point of reversal animals should switch from one choice to another as the choice-reward associations change. It is at precisely such points, when choices of different types are interleaved with one another, that 47/12o disruptions leads animals to struggle to assign rewards to the choice that actually caused them as opposed to other choices that occurred close in time but before or after. This, in turn, hampers the animal's ability to switch effectively as the reward contingencies change. However, such impairments are especially apparent in stochastic and constantly changing environments in which there are many possible choices.

Other mechanisms, however, may be in play during reversal tasks that have sometimes been thought to be simpler because the animals reverse between a limited number of discrete states¹³⁻¹⁵, usually two, in each of which the better choice is different.

Minor Comments

By addressing these points, the authors could provide a more cohesive and comprehensive account.

(minor-1) Additional Behavioral Indices (RT, Gaze, etc.)

Beyond basic choice data, measures like reaction time (RT) or eye tracking can provide additional insights into how animals track and update latent states or reversal probabilities. For instance, RT may reflect decision difficulty or uncertainty, while gaze patterns might indicate exploratory behavior regarding alternative options. Such richer behavioral metrics (e.g., exploration-exploitation balance, confidence levels, hesitation times) could reveal how tFUS affects learning strategies and further link model parameters to observed behavior. Including these metrics would greatly enrich interpretations of the tFUS results and model-based analyses.

It is a very good idea to consider reaction time (RT) measures as suggested by the reviewer and so we have done that by conducting analyses of RT that are analogous to those of the choice data that

revealed the effects of our variables of interest on switching. None of the variables exerted anything even close to a significant impact on RT. It is, however, perhaps not surprising that this is the case as the task was not really designed to optimize RT measurements and the animals were not encouraged to respond quickly. Performing a task of this type is not as straight forward for a macaque as it would be for a human. The suggestion of examining gaze data is an interesting one but we were not able to obtain reliable gaze data in our experiment. This partly reflects the difficult environment in which any eye tracking would have to be conducted, inside the MRI scanner, and partly reflects the difficulty of adapting MRI-safe methods for human eye tracking for macaques given the fact that they typically have dark and large irises and the visible “white of the eye” – the sclera is very limited.

(minor-2) Reproducibility and Individual Differences

How consistent are the findings across individual macaques? To what extent do variations in behavior or anatomy among subjects influence the main conclusions?

As suggested, we make clear in the revised manuscript that the group mean results are indicated by bars but that the data for individual sessions are also included and shown as circles. Each vertical column of circles shows the sessions from one individual. For example, the revised manuscript results now states:

Although the correlation between the reward probabilities associated with the two options was not exactly $r = -1$ in the correlated task but averaged approximately -0.798 across all schedules used, this correlation was consistently more negative than in the uncorrelated sessions (Fig.1C; **the bars indicate group mean performance but the data for each individual session are plotted as circles and the circles in each column in this figure and elsewhere are the data for each of the four individual macaques**).

The revised figure legend for figure 1 now states:

(C) In the correlated sessions, the Pearson correlation coefficient between the two options' reward probabilities for the four macaques (M1-M4) averaged at around -0.8 . It was not exactly -1 , but choice probabilities were still significantly more negatively correlated than in the uncorrelated sessions. Here, as elsewhere **the group mean results are indicated by bars but the data for individual sessions are also included and shown as circles. Each vertical column of circles shows the sessions from one individual.**

All the other figures include similar columns of circles corresponding to each individual animal's data (for example, figure 2F, figure 3, figure 4D, E, F, figure 5A, B, figure 6C, D, figure 7).

(minor-3) Local Impact of tFUS

When tFUS is applied to deeper regions such as dmPFC, hippocampus, or thalamus, how confined is the stimulation to the intended target? Could nearby white matter tracts or adjacent regions be inadvertently affected? Further, how did the author confirm the selective and focal modulation of the tFUS? Clarifying this would help ensure an accurate interpretation of any observed behavioral

and neural effects.

The reviewer raises an important point concerning TUS's ability to affect white matter. We used the acoustic simulation toolbox (k-Plan, Brainbox, UK) to simulate the depth and acoustic intensity of the TUS focus in the dmPFC, hippocampus, and thalamus. The simulations used both the T1-weighted scan and a pseudo computed tomography scan, which was generated based on the T1-weighted scan. On re-reading our manuscript we realised that we omitted to make clear that the insets in Figure 7 shows our estimate of the ultrasound relative field intensities and in every case these are maximal in the grey matter rather than the white matter. We have made this clearer in the revised manuscript as shown below:

7. TUS diminished the influence of reversal estimate on switching. Logistic regression analyses applied to each session (illustrated by grey dots) from each animal (illustrated in three columns corresponding to each of the three animals) were used to illustrate whether the reversal estimate (highlighted in yellow column) changed between sham (A) and after disruption of dmPFC (B), hippocampus (C), or thalamus (D). **Insets in each panel illustrate estimates of the ultrasound relative field intensities obtained using the acoustic simulation toolbox (k-Plan, Brainbox, UK) and in every case these are maximal in the grey matter region targeted rather than the white matter.**

However, we understand that the reviewer is also asking if the TUS does reach white matter then can it exert its effects via those white matter regions. The offline methods we used in the current study entail application of stimulation to the brain several minutes before task performance even begins and therefore depend on the induction of a short term plastic change. The ultrasound waves produced by TUS interact with neuron and astrocyte cell membranes and this, in turn, affects adjacent synapses through an N-methyl-d-aspartate (NMDA)-dependent plasticity mechanism. Thus changes can occur in white matter structure such as astrocytes but we think that the plastic changes are exerted only in circumscribed regions of grey matter in which the synapses are found (they are not found in the white matter). This leads to the stimulated grey matter regions, but not adjacent or other regions, becoming insensitive to active changes in interconnected areas. We have added consideration of these points to the revised Discussion as follows:

The offline TUS method we used in the current study entails application of TUS several minutes before task performance even begins and therefore depends on the induction of a short-term plastic changes at synapses^{10,16-21}. The ultrasound waves produced by TUS interact with neuron and astrocyte cell membranes and this, in turn, affects adjacent synapses in grey matter through an N-methyl-d-aspartate (NMDA)-dependent plasticity mechanism. This leads to the stimulated grey matter regions, but not adjacent or other regions, becoming insensitive to active changes in interconnected areas.

Reviewer #2 (Remarks to the Author):

The manuscript reports a study investigating the networks and computations underlying performance in reversal learning tasks in non-human primates. Specifically, a model emphasizing identification of transitions between latent states is tested. An fMRI experiment shows that macaques track latent state transitions in addition to choice values in dorsomedial frontal cortex and anterior and dorsomedial thalamus. The hippocampus tracks the probability of a reversal between latent states. In a follow-up experiment, focused transcranial ultrasound (TUS) is applied to these regions impaired reversal learning, mainly the use of the reversal estimate parameter.

This is an interesting manuscript addressing a long-debated topic. It has a potentially high impact on the way reversal learning (or, more generally, learning latent task structures) is implemented in the primate brain. The approach is innovative in several ways. Recording fMRI in the awake and behaving macaque is technically challenging and TUS is an innovative method of non-invasive brain stimulation.

However, at this point, a number of questions and unclarities remain, such that it is difficult to evaluate whether the inferences drawn are firm enough to warrant publication. The following list of comments may be helpful for revising and clarifying the manuscript:

1. How were the animals trained to learn the different task structures (correlated vs. uncorrelated)? Were they cued before each session which task variant will be performed? If not, how did the animals find out on which version they currently work? Could there be a spill-over from one variant to the other? E.g., how can we be sure that the animals did not try to search for latent state transitions in the uncorrelated version or actually tracked two sets of states (one for each stimulus) that could change between high or low reward probability?

The reviewer raises two important points. First, how were the animals trained and tested on the two different tasks and, second, might animals seek latent state transitions or stimulus-specific transitions. Turning, first, to the first point, in the revised manuscript we make clear that the correlated task data were collected prior to the uncorrelated task data. We explain that we needed to do this because to interleave uncorrelated and correlated trials would be very demanding for a macaque (it has not something that, to the best of our knowledge, has been previously attempted). Critically, it would, effectively, require the macaques to learn an even higher order latent state – are they in the state in which the options' values are negatively correlated (the correlated task) or are they in the state where the options' values are uncorrelated (the uncorrelated task). This would produce a problematic artefact in the experimental design because the macaques would be tracking this latent state even in the uncorrelated task so that we might expect the comparison of brain activity in the correlated and uncorrelated task states to be less useful. We have explained this in the revised manuscript.

Some features of our task design are worth noting. First, in experiment 1, we tested animals on the correlated task prior to the uncorrelated task. One important reason for doing this, however, is to avoid introducing a potential confound; if macaques alternate over days, between the correlated and the uncorrelated tasks then they would effectively learn and track a higher order latent state variable: is the current state the one in which the rules of the correlated task prevail or the one in which the

rules of the uncorrelated state prevail. Neural mechanisms for tracking latent task states would, thus, be employed even in the uncorrelated task and it would no longer be possible to identify their operation by comparing the correlated and uncorrelated tasks.

Turning now to the second point, about whether macaques might seek latent states in the uncorrelated task. This is possible and something that we have attempted to consider. Note that the switch estimate is a continuous variable corresponding to the animal's estimate that switching from one choice to the other is now the correct thing to do. It is derived by fitting model2 to the recordings made of each animal's behaviour and rewards in a given session in order to estimate what belief the macaque might hold on each trial given the experiences it has had. The model can, therefore, be applied to the uncorrelated task just as it can be applied to the correlated task. Even in the uncorrelated task, an animal might hold beliefs about a reversal that we can estimate. The animal's estimates, however, are unlikely to be accurate ones in the uncorrelated task because the ground truth is that reversals do not occur. However, we think that these estimates are precisely what the reviewer is interested in. The red bar in the fourth pair of bars in figure 3 tell us that in aggregate estimates of latent state switching did not appear to influence switching behaviour to a significant degree in the uncorrelated sessions and that they influenced switching behaviour significantly less than in the correlated sessions. Nevertheless, there might be sessions, or times in sessions, when animals may have sought such latent states. This means that if neural activity were to be found that was correlated with the reversal estimate in the uncorrelated task, then that activity would be related to an estimate of the reversal probability that the animal had formed even though that estimate might not be appropriate for that task. When we tested for the existence of neural activity correlated with reversal estimates in the uncorrelated task we were unable to find it when we searched across the whole brain and, in fact, we did not find it in either the dmPFC or the hippocampus ROIs (red bars on the right hand side of figures 4D, E, and F), nor did we find it in thalamus (figure 6).

We also looked at activity related to the change in the reversal estimate that occurs at the time of reward/error feedback at the end of each trial which we refer to as reversal estimate D_{KL} or simply as reversal D_{KL} . We found activity related to reversal D_{KL} in one part of dmPFC but not in hippocampus (red bars on the left hand sides of figures 4D, E, and F). In aggregate with the right-hand patterns of results discussed in the previous paragraph, this suggests that when animals receive feedback they may transiently attempt to update a reversal estimate even in the uncorrelated task but that it does not result in a sustained pattern of activity that for a reversal estimate at the beginning of the next trial (right hand sides of figures 4D, E, and F). Moreover figure 5 suggests that this might be because, in the uncorrelated task, but not in the correlated task, macaques update an estimate of something akin to an option specific update (ie not an estimate about the probability of a reversal in latent state but an estimate of the change in one specific choice), which we think is akin to what the reviewer is thinking about in suggesting a stimulus-specific state. The effects did not quite reach significance in dmPFC but, because we agree with the reviewer that they are potentially interesting, they are illustrated in figure 5. In the revised manuscript we have attempted to address these issues by making a series of changes as follows:

One possibility is that macaques may attempt to infer latent task states even in the uncorrelated task even if none are present. We can estimate what estimates animals themselves might have about such states and the probability of switching between them – the reversal estimate – in the uncorrelated

task by applying model 2 to the uncorrelated task data. By doing this we are estimating what beliefs the animals might hold at any point in time given the choices that they have made and the feedback – rewards and errors – they have observed even if those beliefs might be erroneous given the nature of the reversal task. Because the reversal estimate is a continuous rather than binary variable it may capture a weak or partial belief that the animal has about a latent state reversal, although if such beliefs are held in the uncorrelated task the reversal estimate may often be high. We can then examine whether these estimates are predictive of actual switching behaviour or of neural activity even in the uncorrelated task. In this way we can compare behaviour and neural activity in the two tasks more directly.

...The reversal estimate is, however, only furnished by model 2 and so we employed the reversal estimate from model 2. As explained above, this means that we are able to compare the strength of influence the reversal estimate might have on both behaviour and brain activity....

... Importantly, the animals were more likely to switch when the reversal estimate – model 2's estimate of the probability of reversal between one latent state and another – was high (Fig.3, right; $\chi^2_1=32.1149$; $p<0.001$) and this effect was greater in correlated sessions ($\chi^2_1=24.7756$; $p<0.001$). Even if animals might seek to identify latent states in the uncorrelated task and transitions between them, our estimates of when they might be doing so, derived from application of model 2 to the uncorrelated task suggests that they have significantly less influence in the uncorrelated task.

Analogous analyses can also be performed with the full set of five factors (chosen probability, unchosen probability, chosen uncertainty, unchosen uncertainty, reversal estimate) using LMEM2; (Supplementary Fig.S2B). These analyses confirmed that the animals' reversal estimates drove their switching behaviour more in correlated sessions even when each option's values and uncertainties were considered separately ($\chi^2_1=28.8443$; $p<0.001$) again confirming that such beliefs had a bigger impact on behaviour in the correlated task.

If the animals search for latent task states and reversals between them in the uncorrelated task then by applying model 2 to the uncorrelated task it is possible to estimate what estimates the animals themselves might have about reversal even if these are weak and partial. We were, however, unable to find significant evidence for activity linked to reversal estimates in the uncorrelated task when we examined activity across the whole brain. This was not simply a consequence of the conservative nature of the whole brain analysis approach, because ROI-based analyses similarly revealed no evidence for reversal effects in the uncorrelated task in dmPFC or hippocampus (all $r<0.406$; all $p>0.3$); even though the hippocampus and dmPFC carry reversal estimate-related activity in the correlated task, they do not do so in the uncorrelated task.

While any reversal D_{KL} -related activity in hippocampus was restricted to the correlated task, intriguingly, D_{KL} activity in anterior dmPFC was statistically greater than zero even in uncorrelated sessions (Fig.4E; $t_{34}=2.315$; $p<0.013$). It is possible that this may reflect animals attempting to identify latent task states and transitions between them even in the uncorrelated task. We return to the question of D_{KL} -related activity in dmPFC below to show that, in the uncorrelated task, the animals may possibly do this at the time of feedback, but any attempt to do so does not result in sustained neural activity at the onset of the next trial. The activity may actually reflect tracking features of individual options as well as latent state reversal (discussed further below in Fig.5).

Activity in dmPFC and hippocampus reflected updates in reversal estimates – reversal D_{KL} . It is possible that that this may reflect the animals trying to use feedback to infer a latent task or the possibility of a transition between latent task states. If this is the case, then it does not seem to result in sustained neural activity related to the reversal estimate by the beginning of the next trial (note the absence of positive red bars on the right hand side of figure 5D, E, and F). However, it is possible that this might simply be because dmPFC and hippocampus were actually tracking some other update-related variable that resembled reversal D_{KL} . We therefore employed new versions of models 1 and 2 that yielded separate update estimates for each option. For models 1 and 2 this was under the assumption that options' reward probabilities were uncorrelated or negatively correlated, respectively (Methods, *Model Estimates*). In other words, the models yielded estimates of how much an individual option's associated reward probability was to be updated when the outcome arrived after it was chosen as opposed to an estimate of whether there was likely to be a transition between latent states. We reran the fMRI analyses in the anterior dmPFC and hippocampus ROIs shown in figure 4 but now included both reversal D_{KL} and individual option-specific update term (as appropriate for correlated and uncorrelated sessions) and allowed them to compete to explain activity. Two important results emerged. First, reversal D_{KL} continued to explain significant variance in outcome-locked activity in correlated sessions in both dmPFC and hippocampus (Fig.5, left, dmPFC: $t_{34}=2.003$, $p=0.027$; right, hippocampus: $t_{34}=1.988$, $p=0.027$). Second, the picture was very different in uncorrelated sessions in hippocampus (Fig5B, right); hippocampus showed no evidence of tracking option-specific updates in the same way as reversal D_{KL} . In anterior dmPFC in the uncorrelated task, reversal D_{KL} and option-specific update both explained less variance than they had in correlated sessions and in neither case did the effects reach significance although they were marginal in one case (reversal D_{KL} : $t_{34}=1.687$, $p=0.051$). In summary, dmPFC tracked reversal D_{KL} updates but did so most clearly when the task contained latent structure as was the case in the correlated task. Anterior dmPFC may possibly have estimated, or attempted to estimate, reversal D_{KL} even in uncorrelated sessions, perhaps as a consequence of experience of its utility in the correlated sessions which the same individuals performed. **Instead in the uncorrelated task, dmPFC activity, if anything, reflected the updating of values of the individual options.** However, by contrast, it was very clear that hippocampus was only concerned with tracking of reversal estimates for latent task states, and not option-specific updates, and that it did so only in the correlated task where there was latent structure.

2. It seems counterintuitive that the switch rate is reduced after three consecutive non-rewarding trials compared to the switch rate after two non-rewarding trials. It is astonishing that the macaques would persevere further in 2/3 to 3/4 of the trials when they already received three negative feedbacks. What was the frequency of these trials overall? Wouldn't the models and intuition on the task structure predict a monotonic increase of switch probability as a function of accumulating, consecutive non-reward trials?

The reviewer's intuition is correct that there are relatively few occasions on which the animals' behaviour can be assessed after they have received three non-rewards for repeating the same action because usually the animals have switched already after just one or two errors. This is especially true in the correlated task. We had sought to make this clear by using the lower part of figure 2D to illustrate precisely this point (see below for a reminder). We have now revised the legend so that the meaning of the data shown in the lower part of the panel are clearer as shown below.

D) Across all trials, animals were unlikely to switch from one choice to the other. However, when animals did not receive a reward for a choice, they were likely to switch to choosing the alternative option on the next trial. While switching rates (top) in the correlated (blue) and uncorrelated sessions (red) were similar after one non-reward, switching was significantly more likely in correlated sessions after either two consecutive non-rewards or three consecutive non-rewards. Such a pattern of behaviour is adaptive. In the correlated sessions, the repeated failure to receive a reward for one option indicates that the other option is now likely to be rewarded. In the uncorrelated sessions the repeated failure to receive a reward reveals that that option has a low probability of reward, but it is not possible to make any inference about how good the alternative option might be. The number of observations (#obs) underlying the estimation of the switching rate are also shown (bottom). **As can be seen, there were comparatively few occasions when animals had received three consecutive errors for making the same choice repeatedly in the correlated task.**

Nevertheless, it is true that there are trials on which animals persist even after three errors. It is still important to note, however, that both the relative numbers of occasions in correlated (blue) and uncorrelated (red) sessions on which animals persist repeatedly in making an unrewarded choice and the relative frequencies of switching after these trials are reached in correlated and uncorrelated sessions are in line with expectations; there are fewer occasions on which animals repeatedly choose an unrewarded option in correlated sessions and they tend to switch if there is a further non-reward. The higher frequency with which animals make repeatedly unrewarded choices in the uncorrelated sessions might be adaptive; in uncorrelated environments there will be periods when both options have a low probability of reward and even the better option, which the animal should choose, might be followed by non-reward on three successive occasions. We have noted this in the revised manuscript as shown below.

Finally, we note that in a reasonably large data set such as this one, there may well be some periods when performance lapses as animals become disengaged from the task for short periods. We have referred to this point in the revised manuscript.

On average, animals switched choice from one trial to the next on approximately 13.84% and 16.09% trials in the correlated and uncorrelated task, respectively but, after one reward absence, animals switched to the alternative option at higher rates of, on average, 46.80 and 43.45% of trials in the correlated and uncorrelated task respectively (Fig.1D). While these switching rates in the two tasks were approximately similar after one error, macaques were significantly more likely to switch after

either two and three reward absences in the correlated than the uncorrelated sessions (on average, 73.01 and 52.85% respectively, after two reward absences; 39.46 and 23.65% respectively, after three reward absences). Indeed, an analysis of variance (ANOVA) with factors of session type (correlated versus uncorrelated) and number of consecutive errors (1, 2, or 3) showed an interaction between session type and error number ($F_{2,204}=7.0002$, $p=0.0012$). Such strategies are adaptive in the two task environments. If no reward has been received over two or three successive trials for taking the same option in the correlated task, it is likely that the animal has entered the other latent task state in which it is the other option will be rewarded. In other words, knowledge of the negative correlation structure and the presence of two states latent in the task allows for fast behavioural updates. This is not necessarily the case in the uncorrelated task. While switching away from an option that is repeatedly unrewarded is still adaptive, the outcomes received for one choice option do not provide any information about latent task states and the reward probability of the alternative, thus encouraging more persistence in animal's choices. **In the uncorrelated task, it can sometimes be adaptive for an animal to continue making the same choice for longer despite repeated non-reward. This might happen because there are periods when both options have low probabilities of reward and even the better option, which the animal should choose, might be followed by non-reward on three successive occasions. Finally, it is also worth noting that in a reasonably large data set such as this one, there are likely to some periods when performance lapses as animals become disengaged from the task for short periods before re-engaging²².**

3. What was the reasoning behind the change of response procedure in Experiment 2? Did the fact that the cursor always moved from the left to the right induce any response bias? Reaction times for left responses must have been substantially shorter. Could this have had any effects on how the task was performed and perceived by the animals?

Our aim behind the change in response procedure in experiment 2 was an unrealised one. We had hoped to test additional hypotheses by making further changes to the task but we abandoned this goal. The experiments described were difficult and took place over a protracted period of time that included the COVID lockdown when we were unable to test. We therefore revised and simplified our aims in Experiment 2. However, despite the differences between the superficial features of the tasks, the key behaviours and influences on behaviours were similar across the two tasks. This can be seen by comparing the blue bars in figure 3 and figure 6a and by comparing the blue bars in figure S2 and S5A. We have made this clearer in the revised manuscript as follows. Although as noted, RT was not a very sensitive measure in this study, we did compare the ratios of correct responses on trials where the right hand and the left hand response were correct and we did not find evidence of significant differences. We also found no evidence that the ratios of correct left and right hand responses changed after TUS to any of the areas ($F_{3,6}=1.433$; $p>0.2$).

Our aim behind the change in response procedure in experiment 2 had initially been to test additional hypotheses by making further changes to the task but we abandoned this goal after disruption to testing during the COVID lockdown. Despite differences between superficial features of the tasks, the key behaviours and influences on behaviours were similar across the two tasks. This can be seen by comparing the blue bars in figure 3 and figure 6A and by comparing the blue bars in figure S2 and S5A.

4. For the two experiments, two different groups of macaques were trained. Was there any specific reason to do so? Could the fact that the animals in Exp. 2 knew only the correlated version of the task have influenced their learning of the task structure and their performance? At the very least, assuming that there was no task cue in Exp. 1, uncertainty about the structure was conceivably much higher in the beginning of each session in Exp. 1 compared to Exp. 2.

As noted in the previous response, the experiments were difficult and undertaken over a protracted period of time. We are, however, required to work with animals for a limited period by the regulations governing our research and it was not possible to conduct further research using TUS with the first group of animals and this necessitated the training of a second group of animals.

As far as we can tell, however, the animals' uncertainty was probably similar in the two tasks. As noted in the previous response, the behavioural evidence for this claim is that the blue bars in figure 3 and the bars in figure 6a resemble one another and the blue bars in figure S2 and the bars in figure S5A one another. The reason why behaviour is similar is probably because of the feature of testing that was explained in response to the reviewer's first point: the correlated task data were collected prior to the uncorrelated task data. As explained above, not only would it be very demanding for a macaque to alternate between two versions of the task, but attempting to train them to perform in this way, if successful, might require the macaques to learn an even higher order latent state – are they in the state in which the options' values are negatively correlated (the correlated task) or are they in the state where the options' values are uncorrelated (the uncorrelated task). Therefore, in both experiments 1 and 2, data were collected from the correlated task under similar conditions; in experiment 1, correlated task data were collected before the animals performed the uncorrelated task. In experiment 2, correlated task data were also collected before any uncorrelated task was performed (in fact an uncorrelated task was not performed at all). Hopefully, one of the paragraphs that we inserted in response to the reviewer's first point makes clearer the conditions under which experiment 1 was performed:

Some features of our task design are worth noting. First, in experiment 1, we tested animals on the correlated task prior to the uncorrelated task. One important reason for doing this, however, is to avoid introducing a potential confound; if macaques alternate over days, between the correlated and the uncorrelated tasks then they would effectively learn and track a higher order latent state variable: is the current state the one in which the rules of the correlated task prevail or the one in which the rules of the uncorrelated state prevail. Neural mechanisms for tracking latent tasks states would, thus, be employed even in the uncorrelated task and it would no longer be possible to identify their operation by comparing the correlated and uncorrelated tasks.

5. Regarding the modeling, how was the hypothesis space for the reversal probability (H) determined? Was the 0.3 based on the maximal hazard rate that an animal could encounter during testing?

The choice of the value range used for the reversal probability H (between 0-0.3 as shown in equation 1) was based on behavioural simulations with the types of schedules actually used during the experiments. These simulations showed that even at true reversal time points, H would not take values large than 0.3 and that this was a reasonable parameter range to assume (see Figure 2e blue

dotted line for the changes in H seen for a typical session). We have made the following change to the manuscript.

The choice of the value range used for the reversal probability H (between 0-0.3) was based on behavioural simulations with the types of schedules actually used during the experiments. These simulations showed that even at true reversal time points, H would not take values large than 0.3 and that this was a reasonable parameter range to assume (blue dotted line in Figure 2E illustrates the range of values of H in a typical session).

Related to this, why did the number of reversals vary between 1-7 in the correlated environment? Was reversal occurrence depending on the performance in the uncorrelated task and how many reversals did occur in the uncorrelated environment? Did the mean reversal rates differ between the correlated and uncorrelated variants, and, if so, could that have confounded the results?

The number of reversals in the correlated environments varied because of the stochastic nature of the task design. The reversal was not always programmed to occur after a fixed number of trials or after a fixed number of correct responses. This stochasticity ensured that animals monitored the outcomes of their choices in order to decide when to switch rather than switching regardless of choice feedback. It also had the secondary advantage of helping to minimize collinearity between the reversal estimate and choice values. We have explained this in the revised manuscript as follows:

In the correlated task, reversals occurred randomly regardless of animal performance, and each session contained several such reversals (typically more than 3 reversals but varying between 1 and 7). Typically, an error occurred; the animal received no reward when reversals occurred. Animals became more likely to switch to the alternative option after an error occurred and, as multiple errors accrued, this tendency increased (Fig.1D). **The number of reversals in the correlated environments varied because of the stochastic nature of the task design. The reversal was not always programmed to occur after a fixed number of trials or after a fixed number of correct responses. This stochasticity ensured that animals monitored the outcomes of their choices in order to decide when to switch rather than switching after a fixed number of trials regardless of choice feedback. It also had the secondary advantage of helping to minimize collinearity between the reversal estimate and choice values.**

The control task, the uncorrelated task, by construction, does not have the same reversal events when the two options values switch dramatically. Nevertheless, were an animal to attempt to infer a reversal in the same way in the uncorrelated task as it did in the correlated task then our estimate of the reversal should approximate this process. This is explained in greater detail in response to the reviewer's next point.

6. Lines 229-232: It remained unclear how the reversal estimate could be applied to uncorrelated sessions. Further above it is stated that model 1 showed a better fit to the uncorrelated task data.

The reviewer is correct that reversal estimate explains little variation in switching behaviour on average in the uncorrelated task (figure 3 – right hand side) and that model 1, a model lacking the reversal estimate, is a better model for the uncorrelated task. Nevertheless, as the reviewer notes in their first point “can we be sure that the animals did not try to search for latent state transitions in the uncorrelated version or actually tracked two sets of states (one for each stimulus) that could

change between high or low reward probability?”. To address questions such as this, that seem to have arisen in the reviewer’s mind, we have tried to test whether macaques might seek latent states in the uncorrelated task by applying model 2. In figure 3, the pattern of results on the right-hand side confirm that it has limited explanatory power. Moreover, the neural analyses demonstrate that the reversal estimate from model 2 explains little variance in neural activity.

As pointed out above, the switch estimate is a continuous variable corresponding to the animal’s estimate that switching from one choice to the other is now the correct thing to do and so by examining what happens when we use it in the context of the uncorrelated task, arguably we are testing the possibility that even in the uncorrelated task, an animal might hold beliefs about a reversal even if they are weak and inaccurate beliefs. It is partly with a view to addressing this point 6 that we made some of the changes in response to point 2 and summarized again here for convenience:

One possibility is that macaques may attempt to infer latent task states even in the uncorrelated task even if none are present. We can estimate what estimates animals themselves might have about such states and the probability of switching between them – the reversal estimate – in the uncorrelated task by applying model 2 to the uncorrelated task data. By doing this we are estimating what beliefs the animals might hold at any point in time given the choices that they have made and the feedback – rewards and errors – they have observed even if those beliefs might be erroneous given the nature of the reversal task. Because the reversal estimate is a continuous rather than binary variable it may capture a weak or partial belief that the animal has about a latent state reversal, although if such beliefs are held in the uncorrelated task the reversal estimate may often be high. We can then examine whether these estimates are predictive of actual switching behaviour or of neural activity even in the uncorrelated task. In this way we can compare behaviour and neural activity in the two tasks more directly.

...The reversal estimate is, however, only furnished by model 2 and so we employed the reversal estimate from model 2. As explained above, this means that we are able to compare the strength of influence the reversal estimate might have on both behaviour and brain activity...

... Importantly, the animals were more likely to switch when the reversal estimate – model 2’s estimate of the probability of reversal between one latent state and another – was high (Fig.3, right; $\chi^2_1=32.1149$; $p<0.001$) and this effect was greater in correlated sessions ($\chi^2_1=24.7756$; $p<0.001$). Even if animals might seek to identify latent states in the uncorrelated task and transitions between them, our estimates of when they might be doing so, derived from application of model 2 to the uncorrelated task suggests that they have significantly less influence in the uncorrelated task.

Analogous analyses can also be performed with the full set of five factors (chosen probability, unchosen probability, chosen uncertainty, unchosen uncertainty, reversal estimate) using LMEM2; (Supplementary Fig.S2B). These analyses confirmed that the animals’ reversal estimates drove their switching behaviour more in correlated sessions even when each option’s values and uncertainties were considered separately ($\chi^2_1=28.8443$; $p<0.001$) again confirming that such beliefs had a bigger impact on behaviour in the correlated task.

If the animals search for latent task states and reversals between them in the uncorrelated task then by applying model 2 to the uncorrelated task it is possible to estimate what estimates the animals

themselves might have about reversal even if these are weak and partial. We were, however, unable to find significant evidence for activity linked to reversal estimates in the uncorrelated task when we examined activity across the whole brain. This was not simply a consequence of the conservative nature of the whole brain analysis approach, because ROI-based analyses similarly revealed no evidence for reversal effects in the uncorrelated task in dmPFC or hippocampus (all $r < 0.406$; all $p > 0.3$); even though the hippocampus and dmPFC carry reversal estimate-related activity in the correlated task, they do not do so in the uncorrelated task.

While any reversal D_{KL} -related activity in hippocampus was restricted to the correlated task, intriguingly, D_{KL} activity in anterior dmPFC was statistically greater than zero even in uncorrelated sessions (Fig.4E; $t_{34}=2.315$; $p < 0.013$). It is possible that this may reflect animals attempting to identify latent task states and transitions between them even in the uncorrelated task. We return to the question of D_{KL} -related activity in dmPFC below to show that, in the uncorrelated task, the animals may possibly do this at the time of feedback, but any attempt to do so does not result in sustained neural activity at the onset of the next trial. The activity may actually reflect tracking features of individual options as well as latent state reversal (discussed further below in Fig.5).

Activity in dmPFC and hippocampus reflected updates in reversal estimates – reversal D_{KL} . It is possible that that this may reflect the animals trying to use feedback to infer a latent task or the possibility of a transition between latent task states. If this is the case, then it does not seem to result in sustained neural activity related to the reversal estimate by the beginning of the next trial (note the absence of positive red bars on the right hand side of figure 5D, E, and F). However, it is possible that this might simply be because dmPFC and hippocampus were actually tracking some other update-related variable that resembled reversal D_{KL} . We therefore employed new versions of models 1 and 2 that yielded separate update estimates for each option. For models 1 and 2 this was under the assumption that options' reward probabilities were uncorrelated or negatively correlated, respectively (Methods, *Model Estimates*). In other words, the models yielded estimates of how much an individual option's associated reward probability was to be updated when the outcome arrived after it was chosen as opposed to an estimate of whether there was likely to be a transition between latent states. We reran the fMRI analyses in the anterior dmPFC and hippocampus ROIs shown in figure 4 but now included both reversal D_{KL} and individual option-specific update term (as appropriate for correlated and uncorrelated sessions) and allowed them to compete to explain activity. Two important results emerged. First, reversal D_{KL} continued to explain significant variance in outcome-locked activity in correlated sessions in both dmPFC and hippocampus (Fig.5, left, dmPFC: $t_{34}=2.003$, $p=0.027$; right, hippocampus: $t_{34}=1.988$, $p=0.027$). Second, the picture was very different in uncorrelated sessions in hippocampus (Fig5B, right); hippocampus showed no evidence of tracking option-specific updates in the same way as reversal D_{KL} . In anterior dmPFC in the uncorrelated task, reversal D_{KL} and option-specific update both explained less variance than they had in correlated sessions and in neither case did the effects reach significance although they were marginal in one case (reversal D_{KL} : $t_{34}=1.687$, $p=0.051$). In summary, dmPFC tracked reversal D_{KL} updates but did so most clearly when the task contained latent structure as was the case in the correlated task. Anterior dmPFC may possibly have estimated, or attempted to estimate, reversal D_{KL} even in uncorrelated sessions, perhaps as a consequence of experience of its utility in the correlated sessions which the same individuals performed. Instead in the uncorrelated task, dmPFC activity, if anything, reflected the updating of values of the individual options. However, by contrast, it was very clear that hippocampus was only concerned with tracking of reversal estimates for latent task states, and not option-specific updates, and that it did so only in the correlated task where there was latent structure.

7. Did the model compute inferences separately for the two response options? From the

description of the model, it is not clear how the expected value of reversal (or expected value of the unchosen option) is derived.

Our central interest in this study was to look at whether animals might infer a latent state and, if so, how they did this. The same piece of information – the feedback on each trial – informs both the animal's estimate of the value of the choice it is taking and the reversal estimate. Given that they infer a reversal in latent state, there are two ways that they might then establish a value for the unchosen option. To see how why this is the case, consider what happens before and during the reversal. Prior to the reversal, the animals would learn from experience that the value of choice 1 was high and the value of choice 2 was low. They would therefore take choice 1 and refrain from choice 2. When the reversal occurred, they would observe now that few rewards followed choice 1. They should therefore stop making choice 1 and infer that a reversal was occurring. Two possible things might happen next. First, having inferred that the state has changed, they might move into an exploratory mode and try out the other response and learn by observation if the other choice, choice 2, is now rewarded. Alternatively, they might make a second type of inference; in addition to inferring the change in latent state they might infer a precise value about the value of the unchosen option. They might do this if they exploited the knowledge that the values of the two choices were negatively correlated with one another; if they observed that the chosen value decreased, might they infer that the value of the unchosen option increased. We did not want our determination of latent state inference to be confounded with the second type of inference. Model 2, therefore, did not incorporate the assumption that this second type of inference about the unchosen option might be made. Despite this, model 2 still fit the data from the correlated sessions better than model 1.

Nevertheless, the idea that animals might make inferences about the value of the unchosen option prior to actually choosing it would mean that we should estimate the choice values in a slightly different manner and so we can also investigate a different version of model 2 in which the value of the unchosen option is inferred prior to taking the choice. The difference between models is a small one that particularly affects the value of the unchosen option on a small number of trials when the reversal estimate is high. Nevertheless, we also tried this alternative approach to analysing the neural data from dmPFC and anterior medial thalamus in Supplementary Figure S4. However, the results are very similar (compare supplementary figure S4A and S4B with figures 6C and 6D respectively).

One final note, when considering figures 6C-D and S4 is that under the second model, the values of the two choices are negatively correlated with one another and so this makes their value estimates collinear. Under this assumption, we cannot look for activity separately related to the value of the chosen option and the value of the unchosen option but we can look for activity related to the decision variable – the difference between the value of the chosen and unchosen options. This is what we do in Supplementary Figure S4. The blue bars in Supplementary Figure S4 should therefore approximately correspond to the difference between the values of each pair blue bars in Figures 6C-D where we can look separately at the value of the chosen option and the value of the unchosen option.

We have attempted to make this clearer in the revised manuscript. In the Results section we have added the following section:

Our central interest in this study was to look at whether animals might infer a latent state and, if so, how they did this. The same piece of information – the feedback on each trial – informs the animal about both its estimate of the value of the choice it is taking and the reversal estimate. Given that they infer a reversal in latent state, there are two ways that they might then establish a value for the unchosen option. First, having inferred that the state has changed, they might move into an exploratory mode and try out the other response and learn by observation if the other choice, choice 2, is now rewarded. Alternatively, they might make a second type of inference; in addition to inferring the change in latent state, they might infer a precise value about the value of the unchosen option on the basis that the values of the two choices were negatively correlated with one another; if they observed that the chosen value decreased, they might infer that the value of the unchosen option increased. We therefore used a second version of Model 2 that incorporated the assumption that this second type of inference about the unchosen option might be made in the analyses shown in Supplementary Figure S4. Anterior and medial thalamus and dmPFC activity reflect information about choice values in the correlated sessions even under this alternative assumption.

To address this point, we have made the following changes to Supplementary Figure S4.

S4. Decision-related activity estimated under the assumption that monkeys make inferences about the value of unchosen options from their experience with chosen options. Given that animals infer a reversal in latent state, there are two ways that they might then establish a value for the unchosen option. They might try out the unchosen response and learn by observation what its value now is. Activity related to the chosen and unchosen values under this assumption is illustrated in figure 6. Alternatively, animals might make a second type of inference; in addition to inferring the change in

latent state, they might infer a precise value about the value of the unchosen option on the basis that the values of the two choices were negatively correlated with one another; if they observed that the chosen value decreased, they might infer that the value of the unchosen option increased. Here, we therefore used a second version of Model 2 that incorporated the assumption that this second type of inference about the unchosen option might be made. In other words, If the animals learn from experience that one option that they have chosen recently has a probability, p , of being rewarded then, because the options' reward probabilities are negatively correlated, they should infer that the unchosen option's probability of reward is $1-p$. Decision-related activity in dmPFC (A) and anterior and medial thalamus (B) is estimated under the assumption that monkeys make such inferences in correlated sessions. Because the regressors for the two options are now perfectly negatively correlated with one another it is not possible to look at neural activity that is specifically related to one option or the other (because activity that appears positively related to the value of one option is equally negatively related to the value of the other option) but only to look at activity related to the difference in the options' reward probabilities (chosen option's reward probability – unchosen option's reward probability) and so this is what is illustrated here. The blue bars in this figure should, therefore, approximately correspond to the difference between the values of each pair of blue bars in Figures 6C-D where we can look separately at the value of the chosen option and the value of the unchosen option. This is, indeed, the case.

8. How was the model fit reported in Figure 2F and on page 5 line 175 determined. Was the model set fit to the choice behaviour of the animals or is this the fit of the logistic regression based on parameters extracted either from model 1 or 2?

Apologies, we realise now that the figure should have said "AIC difference model 2 – model 1" instead of simply "AIC score". Both Model 1 and Model 2 were separately fitted to the choice behaviour of the animals and the AIC values are compared here. Thus, we are not plotting the raw AIC values for either model but the difference between them. Because smaller AIC values are preferred, if the M2-M1 difference is negative, this means M2 is superior which is the case for correlated sessions (and vice versa for uncorrelated sessions). We have now made this clear both in the legend and the main text:

Even without inclusion of this extra feature, however, Model 1 provided a better account of the majority of uncorrelated sessions, while model 2 provided a better description of the majority of correlated sessions; the difference in Akaike's information criterion (AIC) scores for Model 2 minus Model 1 are shown for a comparison of the two models for each session (Fig.2F).

Legend Figure 2: Individual grey dots indicate the AIC difference between Model 2 and Model 1 derived from fitting the choice data of each session and sessions from the four animals are shown in four columns as in figure 1c.

9. In the description of Exp. 2 the reader may get the impression that fMRI was recorded as well and be surprised to see no results from that (see line 591). Careful rereading then made clear that the animals were only tested behaviourally after TUS. It might be helpful to clarify this more explicitly to avoid sending readers on a wrong garden path.

We did actually collect fMRI data after TUS application. We did this, however, an upgrade of the MRI scanner that was interrupted by the early stages of COVID lockdown. We developed new scanning sequences for the new coil during the lockdown period that subsequently proved very difficult to reconstruct and analyse. We do not anticipate being able to present the data. We have tried to make this clearer in the revised manuscript.

Seven adult male rhesus monkeys (*Macaca mulatta*) were tested; four performed the correlated and uncorrelated reversal tasks in experiment 1 while functional magnetic resonance imaging (fMRI) data were collected. Three performed the correlated task in experiment 2 in which the impact of transcranial ultrasound stimulation (TUS) on behaviour was measured **while animals were in the same MRI scanner environment**. They were aged between 9-12 years and weighed between 11-17 kg at the time of the experiments. They lived on a 12-hour light-dark cycle, were fed once per day after testing, and had *ad-lib* water access for an average of 15 hours a day (and a minimum of 3 hours per day).

10. Some details on the TUS methodology seem to be missing.

- How long was the interval between stimulation and task (approx. in min)?

The animals were moved from an adjacent room to the MRI scanner room for behavioural testing as fast as possible. On average, we estimate that this took about 5-10 minutes. We have added this information the manuscript:

After stimulation, monkeys were immediately moved to the testing room for behavioural data collection. **We estimate that the delay between the stimulation and the start of the behavioural task took about 5-10 minutes on average.**

- Can you specify how the pseudo-CTs were created? Depending on the parameter value, the skull estimation can be more or less conservative, leading to different values on the thermal and acoustic simulations.

*We realise that this study started quite a while ago now and that the standards in the field are developing fast. Unfortunately, at the time, we did not acquire **individual** Petra/ZTE scans to estimate each individual's pseudo-CT scans. Instead, as done similarly in previous macaque work, the skull was estimated from pseudo-CT images obtained from a typical monkey using a Black Bone MRI sequence²³. We followed a pipeline used in human research (<https://github.com/ucl-bug/petra-to-ct>) to create a pseudo-CT (pCT). We debiased the Black Bone MR image using Slicer's N4ITK MRI bias correction^{24,25}, performed a segmentation to obtain head and skull masks, and then created a pCT by setting the background to values of -1000, setting voxels classified as soft tissue to +42, and linearly mapping the intensities of voxels classified as skull into a range of 42 to 3274. We checked that the distribution of pseudo-Hounsfield units inside the skull of the pCT was similar to the distribution of Hounsfield units inside the skull of a real CT acquired for another animal (mode around 1500, roll-off to 2500). We then used a mapping of Hounsfield units onto units of mass density (kg/m³) based on the pCT scan of a phantom with known mass densities on a comparable 3T Scanner in Oxford. We have explained this as follows in the revised manuscript:*

Acoustic simulations were performed using the pseudo-CT from a representative macaque brain. The pseudo-CT scan was estimated from pseudo-CT images obtained from a typical monkey using a Black Bone MRI sequence²³ (<https://github.com/ucl-bug/petra-to-ct>) . We debiased the Black Bone MR image using Slicer's N4ITK MRI bias correction^{24,25}, performed a segmentation to obtain head and skull masks, and then created the pseudo-CT by setting the background to values of -1000, setting voxels classified as soft tissue to +42, and linearly mapping the intensities of voxels classified as skull into a range of 42 to 3274. We checked that the distribution of pseudo-Hounsfield units inside the skull of the pseudo-CT was similar to the distribution of Hounsfield units inside the skull of a real CT acquired for another animal (mode around 1500, roll-off to 2500). We then used a mapping of Hounsfield units onto units of mass density (kg/m^3) based on the pseudo-CT scan of a phantom with known mass densities on a comparable 3T scanner.

- The **Isppa (80W/cm2)** is quite high compared to previous studies (**60W/cm2**) such as Mahmoodi et al. (Neuron, 2023) . Please specify the results of the acoustic and thermal simulations.

The pulse generator was set to stimulate at an intensity of 80W/cm2 and this is what we reported in the first draft of our manuscript. However, our own recordings of the intensity revealed that the actual intensity in the free-field at focus is approximately 60W/cm2 which corresponds to the intensity reported in the study reported by Mahmoodi et al. (Neuron, 2023). In the revised manuscript, in order to avoid any confusion, we have reported what the measured free-field spatial-peak pulse average intensity was (60W/cm2) rather the pulse generator setting.

The free-field spatial-peak pulse-average intensity (Isppa) corresponded to 60W/cm² in free field at the focus (based on our own calibration of the device and in line with the intensity recorded in previous studies¹⁹).

*We now also report the results of the acoustic simulations. The mean **Isppa** at the focus in hippocampus, thalamus, and dmPFC was 18.5W/cm2, 17W/cm2 and 16W/cm2, respectively, and thus very comparable across regions. The **MI** was 1.49, 1.42 and 1.39. Note that this would make the MI acceptable even in a human study given ITRUSST recommends an MI of less than 1.9. However, the thermal dose exceeded the ITRUSST guidelines of <0.25 CEM43 for all three regions: hippocampus 0.69; thalamus 9.25; dmPFC>100 (but difficult to estimate precisely given the nonlinear nature of the thermal dose). However, we note that the maximum thermal dose was estimated in the skull rather than in dmPFC or elsewhere in the brain. We have now added these values to the manuscript as follows:*

The simulations indicated values of 18.5W/cm2, 17W/cm2 and 16W/cm2, for the Isppa achieved *in situ* around the focus in hippocampus, anterior medial thalamus and dmPFC, respectively. The MI was 1.49, 1.42 and 1.39 and the thermal dose 0.69, 9.25 and >100, respectively, in the three targets. Although the highest thermal dose was in the skull rather than in the brain itself, this shows that for human studies, the chosen pulse protocol is not recommended and can exceed ITRUSST-recommended safety limits.

- Line 1099: "Each stimulation condition was repeated 5-8 times for each animal and each test took

place on a different day.” Why not a fixed number?

We cycled through the various conditions but added one additional day of sham testing at the beginning and end of the testing sessions. This resulted in 6 days of testing for each stimulation condition (dmPFC, thalamus, hippocampus) and 8 days of testing in the sham condition. The one exception to this was that one animal did not complete the final round of testing and so there were only 5 days of testing ifor each target region in this animal. This was break in testing for this final animal was due to a moratorium on testing that occurred because of wider colony management issues. We have clarified this in the revised manuscript as follows:

After stimulation, monkeys were immediately moved to a testing room for behavioural data collection. **We estimate that the delay between the stimulation and the start of the behavioural task took about 5-10 minutes on average.** The sham condition completely matched a typical stimulation session (setting, stimulation procedure, neuro-navigation, targeting, transducer preparation and timing of its bilateral application to the shaved skin on the head of the animal) except that sonication was not triggered **although the TUS sound was simulated.** During the sham session the montage was pseudo-randomly positioned to target dmPFC, hippocampus, thalamus. Each stimulation condition was repeated 5-8 times for each animal and each test took place on a different day. The order of TUS targets was pseudo-randomized for each animal **in order to cycle through the different conditions but with the addition of a sham day at the beginning and end of the series. This resulted in 8 sham testing days and 6 days of testing at each other site with the exception of one animal that only underwent 5 sessions of testing at each target site.** The stimulation **and testing session were** always performed at the same time of the day and there was always a 24-hour gap between each session, regardless of it being a real or sham stimulation session.

11. In the introduction (line 107), it would be more informative to justify the use of TUS some lines above, instead of jumping directly to the findings. It might also be mentioned that TUS is not in all cases disruptive but that certain protocols could be facilitating.

We have followed the reviewer’s suggestions and reordered the text and we have, as suggested, avoided suggesting that TUS only disrupts behaviour.

Here we attempt to re-examine how macaques perform reversal tasks, first, by recording across the whole brain and looking not just for neural activity tracking the values of choices but also testing whether activity reflects an estimate of the probability of a transition from one state to another: a reversal estimate. We employ a Bayesian model to estimate the choice values that monkeys might learn from feedback after taking choices, their uncertainty about those values, but also the probability of reversal from one task state to another. Importantly, we compare neural activity recorded during performance of a reversal task with neural activity recorded when animals perform a task with a similar pair of options but where the values of each option are independent of one another and change independently of one another. Because the options’ values are negatively correlated with one another in our first task, which most closely resembles a classic probabilistic reversal learning task, we refer to it as the “correlated” task. Because the values of the two choices in the control task are unrelated to one another, we refer to this task as the “uncorrelated” task.

Functional magnetic resonance imaging (fMRI) was used to look across the whole brain for neural activity linked to choice value, choice uncertainty, and reversal estimates from the Bayesian model.

Then, in the regions that had been identified, we next examined the impact of transient alteration of neural activity using offline transcranial ultrasound stimulation (TUS), a technique that can be used to manipulate neural activity even deep in the brain without affecting overlying tissue^{10,16–21,26,27}. We considered the possibility of finding activity beyond the OFC in other parts of prefrontal and cingulate cortex which are likely to be deafferented by white matter disconnection lesions are likely to result from the lesions that cause reversal deficits. Areas such as dorsomedial prefrontal cortex (dmPFC) have been associated with task control albeit in different contexts. We also considered the hippocampus because of evidence that its activity also tracks changes in context. Finally, we also considered the possibility of activity in the dorsomedial and anterior thalamic regions that are associated with both dorsomedial frontal cortex and hippocampus. By comparing activity between the two tasks, we identify three regions, one in dmPFC (extending from the dorsal bank of the cingulate sulcus to the superior frontal gyrus), medial dorsal thalamus, and mid hippocampus, where activity reflects the probability of reversal between task states but in distinct and complementary ways, either in isolation or in addition to other task variables. We then showed that transiently **manipulating** each of the three regions with TUS caused changed the way animals performed the task; animals performing the correlated task **were** no longer guided by an estimate of the probability of reversal between states and instead explored options as a function of their uncertainty about them.

12. There was a significant difference between sham and stimulated regions. But was there really a significant difference between the regions in any of the factors?

More generally, did TUS actually impair overall performance in the task (reflected, e.g., in lower accumulated reward)?

In the revised manuscript we have provided evidence that TUS targeted at different areas had different effects as summarized below:

Finally, we examined whether the impact of TUS in each area did not just differ from sham but that the pattern of TUS effects for each area differed from those seen after TUS to each other area. We used an ANOVA to examine the pattern of influences on switching of the predictors relating to the value of the chosen option, the value of alternative option, the uncertainty of the chosen option, and uncertainty of the unchosen option, and the reversal (Supplementary Figure S5) in pairs of areas. We found a significant interaction between TUS target and predictors influencing switching when we compared dmPFC and thalamus ($F_{4,8}=9.325$; $p=0.004$) and when we compared dmPFC and hippocampus ($F_{4,8}=4.341$; $p=0.037$). While we did not find evidence for the same interaction when we compared hippocampus and thalamus TUS ($F_{4,8}=1.71$; $p=0.240$) we did find evidence that the influence of the reversal estimate on switching differed significantly between thalamus TUS and hippocampus TUS ($z=2.322$; $p=0.020$) as indeed it also did between thalamus TUS and dmPFC TUS ($z=2.684$; $p=0.007$).

In the revised manuscript we have also compared the effects of the different types of TUS on the number of rewards received.

As a consequence of the changes to the way that the task was performed, TUS led to a decrease in the rate at which rewards were received (main effect of TUS: $F_{3,6}=9.222$; $p=0.012$; sham versus dmPFC: $F_{1,2}=36.241$; $p=0.027$; sham versus hippocampus: $F_{1,2}=42.912$; $p=0.023$; sham versus thalamus: $F_{1,2}=24.904$; $p=0.038$).

13. Line 1050. “The upsampled data was then epoched to 6s time windows spanning 1s before to 5s after...” – this contrasts with the figures showing 10s epochs, spanning 5s before to 5s after the event of interest.

Thank you for spotting this. This has been corrected as suggested as follows:

For BOLD time course analyses, we extracted the filtered BOLD time series from each ROI (dmPFC, anterior medial thalamus, hippocampus). The extracted signals were then averaged, normalised and up-sampled by a factor of 15^{3,19–21}. The upsampled data was then epoched to 10s time windows spanning 5s before to 5s after the appearance of the reward-opportunity stimulus on each trial, or the outcome on each trial. We then examined the relationship between behaviour and brain activity with ordinary least squares (OLS) GLMs performed at each timepoint in each epoch.

14. Line 553: after discussing the TUS results, the results of the models were included in the discussion. I would suggest to move it right after the fMRI discussion. This way you would have a discussion relating the correlational results and ending with the causal relation of the regions and behavioural outcomes.

As suggested, we have reversed the order of the black and red paragraphs in the main text as shown below:

As noted, in addition to tracking reversal estimates, activity in dmPFC and hippocampus also tracked updates in the reversal estimate, reversal D_{KL} , at the time that macaques obtained feedback, reward or no-reward, for their choices. While it was clear that reversal estimate-related activity was only found in hippocampus in the correlated task, there was some evidence that dmPFC was modulated, albeit weakly, by reversal D_{KL} , in the uncorrelated task. This pattern of activity was, at first, difficult to interpret; it is difficult to know what such an activity pattern might mean when reversal estimates did not influence behaviour in the same way in the uncorrelated task (Fig.3). To interrogate the meaning of the activity patterns further an alternative Bayesian model was constructed that tracked the values of individual choices in ways that were appropriate to either the correlated or uncorrelated task (Fig.4). These models yielded individual choice option value updates – individual option D_{KL} estimates – onto which brain activity could be regressed. When this was done, it was clear that both dmPFC and hippocampus tracked reversal D_{KL} and only reversal D_{KL} in the correlated task rather than individual option D_{KL} . Moreover, it was clear that hippocampus contained no update-related activity in the uncorrelated task when no latent task states were present. However, the dmPFC activity pattern was less clear in the uncorrelated task. The marginally significant activity patterns suggest dmPFC may track option-specific D_{KL} or even attempt to track reversal D_{KL} in the uncorrelated task, perhaps because this had been adaptive in the other tasks that the animals had performed. However, these estimates did not propagate to the hippocampus. While dmPFC may have a more general role in updating, hippocampus only tracks latent states.

This picture of a distributed, multi-component circuit for translating latent state reversal estimates into choice evaluation was also supported by examination of the impact of disrupting each of the three areas with TUS (Fig.7). TUS produces regionally specific changes in neural activity and behaviour that may be mediated by neuronal mechanosensitive ion channels^{10,16–21,26–29}. Application of TUS to any of the three areas led macaques to perform the correlated task in a manner that was more similar to the

uncorrelated task in that switching was no longer guided by reversal estimates. Moreover, just as switching in the uncorrelated task was significantly more influenced by uncertainty about the choice values than was the case in the correlated task (Fig.3, Supplementary Fig.S2), switching also became more influenced by uncertainty about choice values in the correlated task after TUS of hippocampus or anterior medial thalamus (Fig.7, Supplementary Fig.S5). Consistent with the notion that it is within dmPFC that latent state tracking and choice evaluation converge, we found that dmPFC TUS led to the loss of influence of almost all information concerning latent states, choice values, and uncertainties about choice values on switching. Now only the value of the choice that animals had just been taking influenced whether they would take it again.

15. Moreover, a schematic figure or a radar plot for visualising the regions which are connected to each other and what they represent would be helpful for summarizing the findings.

As suggested, and as shown below, we have made a summary figure, figure 8, in the revised manuscript that incorporates radar plots.

Figure 8. Summary of the results. Each of the principal three regions, hippocampus, dmPFC, and anterior medial thalamus are shown at the top. Beneath each brain section, the radar plots indicate the behavioural variables with which activity was most strongly related in the correlated (blue) and uncorrelated (red) tasks (reversal estimate and reversal D_{KL} in hippocampus, reversal estimate, reversal D_{KL} , and the difference between the reward probabilities of the two choices in dmPFC, and the difference between the reward probabilities of the two choices in anterior medial thalamus). Interactions between hippocampal and dmPFC activity occurred as a function of the reversal estimate (top left) and between dmPFC and anterior medial thalamus as a function of the reversal estimate in interaction with chosen option's reward probability.

16. fMRI figures should contain color bars.

Thank you for pointing out this omission. This has been rectified in the revision.

17. Fig. S1 is not correctly reproduced in the MS.

We have revised figure S1 as suggested. We think that, previously, the figure was correctly displayed individually but not in the concatenated file so we will double check that it looks ok in the final version that we submit.

Reviewer #2 (Remarks on code availability):

The code was not available. The link to OSF did not reveal the code, because it is protected.

Thank you for pointing this inadvertent error. We have now hopefully arrange the OSF setting so that it is possible to access the code.

Reviewer #3 (Remarks to the Author):

This study investigates two reversal tasks in macaque monkeys. In both tasks, the animals make a choice between two identical-looking stimuli at each trial. In the correlated task, reward probability of the two stimuli is negatively related (when the reward probability of stimulus A is high, that of stimulus B is low and vice-versa). By contrast, in the uncorrelated task, the reward probabilities of the two stimuli are not correlated and fluctuate independently of each other. The authors then used fMRI to measure brain activity and built a Bayesian model to estimate the underlying latent belief states (note that the correlated but not the uncorrelated task contains distinct states). The Bayesian estimate of reversal probability correlated with activity in multiple regions, including dmPFC and hippocampus. Moreover, dmPFC and thalamus differentially encoded chosen and unchosen reward probability. The authors then used transcranial ultrasound stimulation (TUS) to disrupt dmPFC, hippocampus, or thalamus activity and report that TUS at all three regions reduces the impact of the reversal estimate on behavior.

The work presented in this paper is relevant for the reversal task literature. By using Bayesian estimation methods to track the latent state structure as estimated by the animals, the authors could show which areas are important to track latent states, as opposed to the simpler interpretation in terms of inhibition by frontal areas. However, multiple findings are brittle and currently not borne out by appropriate statistics. Moreover, several methodological issues limit the impact of the work.

We thank the reviewer for their interest in the paper and for the positive comments that they have made. We note the care with which the reviewer has conducted their review which has made us aware of what might initially appear to be limitations of our study at some points. We have revised our manuscript in response to every comment that the reviewer has made and we hope that we address all the reviewer's concerns and pre-empt the possibility of similar concerns arising in the minds of other potential readers.

Major Comments

#1 It is unfortunate that, and unclear why, all the monkeys performed the correlated task first. As the authors allude to, this could have affected beliefs in the uncorrelated task, e.g., that the reversal of one option meant the other option was now more rewarding. Why not randomize task order? More detailed explanation of this design choice is needed. Moreover, it is not clear whether the 8 sessions were consecutive for each type of task or intermingled. It would be important to state this clearly.

In the revised manuscript we make clear that the correlated task data were collected prior to the uncorrelated task data. We explain that we needed to do this because to interleave uncorrelated and correlated trials would be very demanding for a macaque (it has not something that, to the best of our knowledge, has been previously attempted). Critically, it would, effectively, require the macaques to learn an even higher order latent state – are they in the state in which the options' values are negatively correlated (the correlated task) or are they in the state where the options' values are uncorrelated (the uncorrelated task). This would produce a problematic artefact in the experimental design because the macaques would be tracking this latent state even in the uncorrelated task so

that it we might expect the comparison of brain activity in the correlated and uncorrelated task states to be less useful. We have explained this in the revised manuscript.

Some features of our task design are worth noting. First, in experiment 1, we tested animals on the correlated task prior to the uncorrelated task. One important reason for doing this, however, is to avoid introducing a potential confound; if macaques alternate over days, between the correlated and the uncorrelated tasks then they would effectively learn and track a higher order latent state variable: is the current state the one in which the rules of the correlated task prevail or the one in which the rules of the uncorrelated state prevail. Neural mechanisms for tracking latent tasks states would, thus, be employed even in the uncorrelated task and it would no longer be possible to identify their operation by comparing the correlated and uncorrelated tasks.

#2 (Methods). For the fMRI analyses, the authors used cluster-level correction with a cluster-inducing threshold of $p < 0.01$ ($z = 2.3$). However, as has been shown before (Eklund et al., 2016, PNAS), $p < 0.001$ ($z = 3.1$) should be used as cluster-inducing threshold. Should this not work and a completely ROI-based be necessary, then the next point becomes even more important.

We thank the reviewer for raising this important point which has led us to make changes to our manuscript which we hope will make the manuscript much clearer to other readers. We are aware of the paper by Eklund and our colleagues in Oxford. However, we are also aware that the Eklund results pertain to the human brain and their translation to the macaque brain is fraught because of its much smaller size: approximately $85,000\text{mm}^3$ (Malkova et al., European Journal of Neuroscience, 2006) in macaques versus approximately $850,000\text{mm}^3$ in humans (Good et al., NeuroImage, 2001). This means that the entire macaque brain would approximately fit inside the human temporal lobe. Maclaren and colleagues (Methods, 2010), who developed one of the most commonly used macaque atlases, estimate the median region volume of major cortical and subcortical regions identified from the digital Paxinos Atlas as 120.12mm^3 . In brief, this means that it is difficult for activation in the macaque to cover a very extended area. This means that cluster correction techniques, which require activity to be spread over a large area, are not straightforward.

Nevertheless, we note that all three areas that we investigated in the current study, dmPFC, hippocampus, and anterior medial thalamus also survived a cluster-inducing threshold of 3.1. We have explained this in our revised manuscript but we have done so in a cautious way given some of the potential difficulties associated with this approach if it is taken by other investigators in future studies.

Activity in all three areas – hippocampus, dmPFC, and anterior medial thalamus – survived more stringent thresholds for establishing activity in the whole brain analyses ($Z > 3.1$ as well as $Z > 2.3$). This was apparent in whole brain contrasts for reversal estimate in correlated versus uncorrelated sessions in hippocampus and in contrasts for correlated versus uncorrelated sessions and contrasts for correlated sessions for the reward probability difference associated with the two choices during decision making in anterior medial thalamus and dmPFC respectively. We note, however, that the smaller size of the macaque brain may mean that techniques using whole brain cluster correction need to be used with caution in macaque studies.

#3 ROIs (lines 85-87): The authors mention that there was a recent shift away from the OFC in the

reversal literature. This alone however does not sufficiently specify the choice of ROIs. There should be a more explicit link between which areas will be of interest for the current study, by showing literature that clearly shows the role of the ROIs (dmPFC, medial dorsal thalamus, and mid hippocampus) specifically for reversal (estimates or other relevant model parameters).

We have followed the reviewer's suggestion:

Functional magnetic resonance imaging (fMRI) was used to look across the whole brain for neural activity linked to choice value, choice uncertainty, and reversal estimates from the Bayesian model. We considered the possibility of finding activity beyond the OFC in other parts of prefrontal and cingulate cortex which are likely to be deafferented by white matter disconnection lesions are likely to result from the lesions that cause reversal deficits. Areas such as dorsomedial prefrontal cortex (dmPFC) have been associated with task control albeit in different contexts. We also considered the hippocampus because of evidence that its activity also tracks changes in context. Finally, we also considered the possibility of activity in the dorsomedial and anterior thalamic regions that are associated with both dorsomedial frontal cortex and hippocampus.

#4 (Results). The authors (paragraph starting at line 440) suggest that TUS had specific effects for different brain regions. To bear this out statistically, a region-by-effect interaction analysis is required. If this is not significant, the interpretation should be toned down.

In the revised manuscript we have provided evidence that TUS targeted at different areas had different effects as summarized below:

Finally, we examined whether the impact of TUS in each area did not just differ from sham but that the pattern of TUS effects for each area differed from those seen after TUS to each other area. We used an ANOVA to examine the pattern of influences on switching of the predictors relating to the value of the chosen option, the value of alternative option, the uncertainty of the chosen option, and uncertainty of the unchosen option, and the reversal (Supplementary Figure S5) in pairs of areas. We found a significant interaction between TUS target and predictors influencing switching when we compared dmPFC and thalamus ($F_{4, 8}=9.325$; $p=0.004$) and when we compared dmPFC and hippocampus ($F_{4, 8}=4.341$; $p=0.037$). While we did not find evidence for the same interaction when we compared hippocampus and thalamus TUS ($F_{4, 8}=1.71$; $p=0.240$) we did find evidence that the influence of the reversal estimate on switching differed significantly between thalamus TUS and hippocampus TUS ($z=2.322$; $p=0.020$) as indeed it also did between thalamus TUS and dmPFC TUS ($z=2.684$; $p=0.007$).

#5 (Methods). Given the task design with two stimuli, the variables characterizing reversal apply also to the unchosen option (in the correlated task). Accordingly, the question arises whether the findings portrayed as related to reversal indeed concern reversal or apply to unchosen options more generally. This point should be addressed more explicitly than in currently the case.

The reviewer's comment makes us believe that there is something about the way that we have explained our approach that was not clear in the previous version of our manuscript. We apologise for any confusion and if we have misjudged which section of the manuscript needs revision. We think, however, that perhaps there is something unclear in text before and after equation 8 in the Methods that may have led to confusion. We have made the following changes

In summary, model 2 is a Bayesian model appropriate for correlated sessions. It is appropriate for the correlated sessions because it tracks the latent state structure of the environment to inform selection between options. The model makes it possible to define the expected value of reversal (referred to as the reversal estimate throughout the Results) as the expected value of the unchosen option:

$$\text{ReversalEV} = \sum p(V_2\text{UnCh}_t) * V_2\text{UnCh}_t \quad (8)$$

Note that while equation 8 summarizes the estimated reward value to the animal of switching and taking the alternative action, our focus in both the behavioural and neural analyses is on how animals tracking the probability of reversal, which we refer to as the reversal estimate, and then separate variables track the reward values of the chosen option and the unchosen option (which are the estimates of the probabilities of reward for the chosen and unchosen option respectively) and the uncertainties of those estimates. The reversal estimate and the choice value variables used in our analyses were uncorrelated (Supplementary Fig. S1).

Each time an outcome is observed after a choice is made, the estimate of the reversal probability H is updated...

We hope that this revision to the text ensures that it is now consistent with the rest of the manuscript and that it is clear that separate variables in the model track the probability of reversal and the reward value of the chosen and unchosen options and that these variables are uncorrelated with one another.

#6 (Methods). Given the auditory (e.g., Braun et al., 2020, Brain Stimul.) and potentially other confounds of active transcranial ultrasound stimulation, it seems insufficient for the control (sham) condition to use no stimulation at all. This is relevant also in the present offline protocol, because like the stimulation effects, the auditory effects may gradually subside with time.

We thank the reviewer for raising the important issue. As a result, we have revised our manuscript in two ways. First, As the reviewer notes, TUS delivery is associated with a sound. Human participants can detect this sound (Braun et al., Brain stimulation, 2020) and it induces a startle response in mice (Sato et al., Neuron, 2018). These auditory effects are accompanied by a change in neural activity that could be measured with calcium imaging in mice and EEG in humans (Braun et al., Brain Stimulation, 2020; Sato et al., Neuron, 2018) but impact lasts for a approximately 2 s or less. The behavioural testing that we performed, was conducted tens of minutes later. There is no evidence that the train of TUS, if applied to another brain area, has any long-term effect on auditory cortex (Verhagen et al., eLife, 2019; Folloni et al., Neuron, 2019; We have made this clear in the revised manuscript.

Second, we make clear in our revised manuscript that the sham stimulation involved a simulation of the TUS sound and so hopefully this will allay the reviewer's concerns. However, in our opinion, there are other features of the sham stimulation that are likely to be of greater importance for any researcher using TUS in animal models. We, therefore, thank the reviewer for raising the issue in the way in which they did because it has reminded us of the importance of noting these other features. The sham stimulation that we employed involved taking the animal to the same testing room in which TUS was normally administered and registering the animal's head and brain to an MRI image using brainsight and preparing the stimulation site in the same manner that was normally the case for actual stimulation. Therefore, any anxiogenic effects of the TUS procedure were carefully controlled for. We have made this clear in the revised manuscript as follows in the Discussion:

...we note that the TUS procedure we employed was an offline one; the TUS was applied for 40 s and animals only began performing the task several minutes later and continued to do so for several tens of minutes. In this way, we hoped to reduce the impact of any auditory sensation associated with TUS application; the auditory impact of the TUS produces auditory-associated changes in neural activity that last several hundreds of milliseconds or possibly even 2 s but they have not reported to produce neural activity changes beyond that time when we collected behavioural data in experiment 2. The 40 s train induces a change in activity in the stimulated area that exerts a measurable impact on neural activity for approximately 3 hours and on behaviour for approximately 1 hour. Because the impact of the stimulation outlasts the auditory impact of the stimulation, the behavioural effects cannot be explained as the consequence of auditory stimulation. In line with this interpretation, while this TUS protocol consistently produces neural changes that outlast the TUS application period in the targeted area, it does not induce any measurable change in auditory brain structures. We note that other neurostimulation techniques such as theta burst transcranial magnetic stimulation similarly induce neural changes that outlast the stimulation period but no auditory changes that outlast the TUS application. Moreover, it is important to emphasize that the sham TUS procedure that we employed entailed bringing the animal to the laboratory and making all the normal preparations for TUS administration such as registering the animal's head and brain to a MRI of the head and brain using neuronavigation tools and preparing the skin for TUS. The sham stopped short of actual TUS delivery and ended simply with simulation of the TUS sound. Through very careful and gradual training of the animals, we ensured that the TUS procedure caused as little as anxiety as possible but any residual anxiety that might have been associated with the TUS procedure was controlled for in sham and TUS conditions.

In the revised Methods we note:

After stimulation, monkeys were immediately moved to a testing room for behavioural data collection. The sham condition completely matched a typical stimulation session (setting, stimulation procedure, neuro-navigation, targeting, transducer preparation and timing of its bilateral application to the shaved skin on the head of the animal) except that sonication was not triggered **although the TUS sound was simulated**. During the sham session the montage was pseudo-randomly positioned to target dmPFC, hippocampus, thalamus. Each stimulation condition was repeated 5-8 times for each animal and each test took place on a different day. The order of TUS targets was pseudo-randomized for each animal. The stimulation was always performed at the same time of the day and there was always a 24-hour gap between each session, regardless of it being a real or sham stimulation session.

We used the acoustic simulation toolbox (k-Plan, Brainbox, UK) to simulate the depth and acoustic intensity of the TUS focus in the dmPFC, hippocampus, and thalamus. The simulations used both the T1-weighted scan and a pseudo computed tomography scan, which was generated based on the T1-weighted scan.

In the introduction we note that TUS applied is offline TUS.

Functional magnetic resonance imaging (fMRI) was used to look across the whole brain for neural activity linked to choice value, choice uncertainty, and reversal estimates from the Bayesian model. Then, in the regions that had been identified, we next examined the impact of transient alteration of neural activity using offline transcranial ultrasound stimulation (TUS), a technique that can be used to manipulate neural activity even deep in the brain without affecting overlying tissue^{10,16–21,26,27}. We considered the possibility of finding activity beyond the OFC in other parts of prefrontal and cingulate cortex which are likely to be deafferented by white matter disconnection lesions are likely to result from the lesions that cause reversal deficits. Areas such as dorsomedial prefrontal cortex (dmPFC) have been associated with task control albeit in different contexts. We also considered the hippocampus because of evidence that its activity also tracks changes in context. Finally, we also considered the possibility of activity in the dorsomedial and anterior thalamic regions that are associated with both dorsomedial frontal cortex and hippocampus. By comparing activity between the two tasks, we identify three regions, one in dmPFC (extending from the dorsal bank of the cingulate sulcus to the superior frontal gyrus), medial dorsal thalamus, and mid hippocampus, where activity reflects the probability of reversal between task states but in distinct and complementary ways, either in isolation or in addition to other task variables. We then showed that transiently manipulating each of the three regions with TUS caused changed the way animals performed the task; animals performing the correlated task were no longer guided by an estimate of the probability of reversal between states and instead explored options as a function of their uncertainty about them.

We have clarified the mode of stimulation and the fact that it obviates any potential for a confounding influence of an auditory artefact in the revised Results:

To test this possibility, in experiment 2, we trained three new macaques to perform the correlated task and examined the impact of disrupting activity in dmPFC, hippocampus, and anterior and dorsomedial thalamus with transcranial ultrasound stimulation (TUS). We used a similar procedure to that used in the past where a short 40-second train of TUS disrupts subsequent neural activity and behaviour for a sustained period; effects gradually decline over a period of hours and are not discernible on the following day^{10,16–21,26,27}. The approach of delivering TUS prior to task performance and then examining its subsequent effects precludes the possibility of the TUS simply distracting animals via a peripheral effect or auditory confound that occurs only at the time of stimulation. Previous studies using this approach have found no evidence of any change in auditory brain areas that outlast the stimulation period (Folloni et al 2021, Folloni et al 2019, Fouragnan et al 2019, Khalighinejad et al 2020, Verhagen et al 2019).

Minor Comments

#1 (Methods/Intro). Some details of the methods, which are necessary to understand the results, are found only in the Methods section. The authors might consider providing some more

information, e.g. an intuition on models 1 and 2, so that the results section can be read without having to continuously jump to the methods section.

As suggested, we have added additional information to guide intuitions about the models. The revised manuscript includes the following section.

Ultimately the task that the animals are performing is a decision-making task in which they must select one choice rather than another. The important decision variable is how much better is one choice than the other. We therefore considered the difference between the probability of reward associated with the chosen option and the unchosen option – on neural activity and the difference between the uncertainties associated with the estimates of reward probability that the animals might make. Finally, and a key focus for the study, is whether an additional third variable – reversal estimate – also explains variation in behaviour. We developed two Bayesian models to track the key features of the correlated and uncorrelated tasks (Methods: Bayesian modelling for correlated environments; Fig.2A). Both models tracked belief estimates of reward probabilities, and the associated uncertainties of those estimates, and thus belief estimates that macaques might optimally hold given the outcomes (reward/non-reward) observed following their choices (Fig.2B). The beliefs about the values of each option were expressed as a probability distribution and updated according to Bayes' theorem, by multiplication of the prior belief at the beginning of each trial and likelihood of the outcome observed at the end of the trial to produce a posterior belief. While these were the only task parameters tracked by model 1, model 2 also tracked the probability of a task state reversal (Fig.2C, D). Beliefs were quantified by a joint probability density function (pdf) over a pre-defined hypothesis space (Figure 2C). When it estimated that the animal was transitioning to a different latent task state, it quickly revised the probability and uncertainty estimates of the choices that animal took each time feedback (reward or non-reward) arrived after making a choice. Thus, the reward probability estimates for the choices and the uncertainties on those estimates in model 2 were informed by the reversal estimate. In the correlated environment, options' contingencies are partially negatively correlated with each other such that if one option has a high reward contingency, the other option is likely to have a low reward contingency (Figure 1A, Figure 2A, B) so that a value of an option – its probability of reward – tends to change in sudden jumps (from high to low or vice versa). In model 2 the subjective estimate of the probability of a reversal influences the estimates of choice value and helps to ensure the sudden changes in choice values that accord with the ground truth of the situation in correlated sessions. In other respects, however, the reward probability and uncertainty estimates furnished by the models were similar in nature (Fig.2E). A separate but related feature that might be incorporated into Model 2 is a capacity to infer an unchosen option's value from feedback obtained from the other, chosen option. We also consider such a capacity for inference below (Fig.S4). Even without inclusion of this extra feature, however, Model 1 provided a better account of the majority of uncorrelated sessions, while model 2 provided a better description of the majority of correlated sessions; Akaike's information criterion (AIC) scores are shown for a comparison of the two models for each session (Fig.2F).

#2 (Results). In the correlated task, a reversal means that the option which was previously most rewarding becomes the least rewarding and vice-versa. In the uncorrelated task, however, it means that either one of the options which was previously rewarding, is now not rewarding (and the opposite is also true)? How is the reversal specifically defined? Presumably “when the probability of reward goes from below (above) 0.5 to above (below) 0.5”? The authors should better specify this definition, which can only be found in the Methods section. Also, what are the possible latent states in the uncorrelated task?

The reviewer raises an important issue that, by definition, reversals do not occur in the same manner in the uncorrelated task as they do in the correlated task so how can we compare the tasks? The first eschews attempting to determine whether or not a “reversal” is occurring and focusses instead on objectively definable events: whether or not a choice led to an error and whether the animal switched (i.e. reversed) its behaviour on the next trial. This is what we do in some of the initial analyses we report (for example, in figure 1D).

The second approach is to apply model 2 which estimates the macaques’ subjective beliefs about the choice values and also their subjective estimates about the probability of a switch. We refer to this belief as the reversal estimate. This reversal estimate is a continuous variable rather than just a binary error outcome as in figure 1D; the animal might believe there is a tiny chance of a reversal, a moderate chance of a reversal, or a high chance of a reversal or anything in between. This means that we can use model 2 to estimate what belief the macaque might hold given the experience it has had. The model can even be applied to the uncorrelated task because the animal might hold beliefs about a reversal that we can estimate although any such estimates that the animals would have a limited accuracy given that, as the reviewer points, there is not, objectively speaking a reversal. Using this approach, however, we can estimate the animals’ own subjective estimates about reversal and we can examine whether neural activity is present that is related to these estimates. Figure 2F suggests that occasionally, but not often, animal might estimate a reversal is possible in the uncorrelated task when each day’s testing session is considered. To the degree to which macaques do attempt to estimate a probability of reversal, even if it has limited utility in the uncorrelated task then we can test whether neural activity is correlated with the estimates that the animal might form. The red bars on the right-hand sides of figures 4D, E, and F suggest that, in line with reversal estimates’ limited utility for behaviour, there is limited evidence that neural activity tracks reversal estimates in the uncorrelated task.

The left hand side of figures 4E, however, suggests that in one part of dmPFC, but not in hippocampus, there is activity that might be expected at the point when estimates of reversal might be updated in the uncorrelated sessions. Figure 5, however, suggests that, even if this might happen to some degree, activity in dmPFC at the time of decision outcomes (reward/non-reward) is probably because the macaques in the uncorrelated sessions are updating their estimates of the values of individual options.

We have made the following changes to the manuscript to address the reviewer’s concerns. In the results we have added the following text:

One possibility is that macaques may attempt to infer latent task states even in the uncorrelated task even if none are present. We can estimate what estimates animals themselves might have about such states and the probability of switching between them – the reversal estimate – in the uncorrelated task by applying model 2 to the uncorrelated task data. By doing this we are estimating what beliefs the animals might hold at any point in time given the choices that they have made and the feedback – rewards and errors – they have observed even if those beliefs might be erroneous given the nature of the reversal task. Because the reversal estimate is a continuous rather than binary variable it may capture a weak or partial belief that the animal has about a latent state reversal, although if such beliefs are held in the uncorrelated task the reversal estimate may often be high. We can then examine whether these estimates are predictive of actual switching behaviour or of neural activity even in the

uncorrelated task. In this way we can compare behaviour and neural activity in the two tasks more directly.

...The reversal estimate is, however, only furnished by model 2 and so we employed the reversal estimate from model 2. As explained above, this means that we are able to compare the strength of influence the reversal estimate might have on both behaviour and brain activity....

... Importantly, the animals were more likely to switch when the reversal estimate – model 2's estimate of the probability of reversal between one latent state and another – was high (Fig.3, right; $\chi^2_1=32.1149$; $p<0.001$) and this effect was greater in correlated sessions ($\chi^2_1=24.7756$; $p<0.001$). Even if animals might seek to identify latent states in the uncorrelated task and transitions between them, our estimates of when they might be doing so, derived from application of model 2 to the uncorrelated task suggests that they have significantly less influence in the uncorrelated task.

Analogous analyses can also be performed with the full set of five factors (chosen probability, unchosen probability, chosen uncertainty, unchosen uncertainty, reversal estimate) using LMEM2; (Supplementary Fig.S2B). These analyses confirmed that the animals' reversal estimates drove their switching behaviour more in correlated sessions even when each option's values and uncertainties were considered separately ($\chi^2_1=28.8443$; $p<0.001$) again confirming that such beliefs had a bigger impact on behaviour in the correlated task.

If the animals search for latent task states and reversals between them in the uncorrelated task then by applying model 2 to the uncorrelated task it is possible to estimate what estimates the animals themselves might have about reversal even if these are weak and partial. We were, however, unable to find significant evidence for activity linked to reversal estimates in the uncorrelated task when we examined activity across the whole brain. This was not simply a consequence of the conservative nature of the whole brain analysis approach, because ROI-based analyses similarly revealed no evidence for reversal effects in the uncorrelated task in dmPFC or hippocampus (all $r<0.406$; all $p>0.3$); even though the hippocampus and dmPFC carry reversal estimate-related activity in the correlated task, they do not do so in the uncorrelated task.

While any reversal D_{KL} -related activity in hippocampus was restricted to the correlated task, intriguingly, D_{KL} activity in anterior dmPFC was statistically greater than zero even in uncorrelated sessions (Fig.4E; $t_{34}=2.315$; $p<0.013$). It is possible that this may reflect animals attempting to identify latent task states and transitions between them even in the uncorrelated task. We return to the question of D_{KL} -related activity in dmPFC below to show that, in the uncorrelated task, the animals may possibly do this at the time of feedback, but any attempt to do so does not result in sustained neural activity at the onset of the next trial. The activity may actually reflect tracking features of individual options as well as latent state reversal (discussed further below in Fig.5).

Activity in dmPFC and hippocampus reflected updates in reversal estimates – reversal D_{KL} . It is possible that that this may reflect the animals trying to use feedback to infer a latent task or the possibility of a transition between latent task states. If this is the case, then it does not seem to result in sustained neural activity related to the reversal estimate by the beginning of the next trial (note the absence of positive red bars on the right hand side of figure 5D, E, and F). However, it is possible that this might simply be because dmPFC and hippocampus were actually tracking some other update-related variable that resembled reversal D_{KL} . We therefore employed new versions of models 1 and 2 that yielded separate update estimates for each option. For models 1 and 2 this was under the assumption that

options' reward probabilities were uncorrelated or negatively correlated, respectively (Methods, *Model Estimates*). In other words, the models yielded estimates of how much an individual option's associated reward probability was to be updated when the outcome arrived after it was chosen as opposed to an estimate of whether there was likely to be a transition between latent states. We reran the fMRI analyses in the anterior dmPFC and hippocampus ROIs shown in figure 4 but now included both reversal D_{KL} and individual option-specific update term (as appropriate for correlated and uncorrelated sessions) and allowed them to compete to explain activity. Two important results emerged. First, reversal D_{KL} continued to explain significant variance in outcome-locked activity in correlated sessions in both dmPFC and hippocampus (Fig.5, left, dmPFC: $t_{34}=2.003$, $p=0.027$; right, hippocampus: $t_{34}=1.988$, $p=0.027$). Second, the picture was very different in uncorrelated sessions in hippocampus (Fig5B, right); hippocampus showed no evidence of tracking option-specific updates in the same way as reversal D_{KL} . In anterior dmPFC in the uncorrelated task, reversal D_{KL} and option-specific update both explained less variance than they had in correlated sessions and in neither case did the effects reach significance although they were marginal in one case (reversal D_{KL} : $t_{34}=1.687$, $p=0.051$). In summary, dmPFC tracked reversal D_{KL} updates but did so most clearly when the task contained latent structure as was the case in the correlated task. Anterior dmPFC may possibly have estimated, or attempted to estimate, reversal D_{KL} even in uncorrelated sessions, perhaps as a consequence of experience of its utility in the correlated sessions which the same individuals performed. **Instead in the uncorrelated task, dmPFC activity, if anything, reflected the updating of values of the individual options.** However, by contrast, it was very clear that hippocampus was only concerned with tracking of reversal estimates for latent task states, and not option-specific updates, and that it did so only in the correlated task where there was latent structure.

#3 (Results). In the text, it is mentioned that figure 2F shows the AIC for both models for both sessions. It is not clear which is which, as model1 and model 2 are not mentioned in the figure (line 176).

We apologise for this oversight. The y-axis of figure 2F has now been updated so that it runs from "model 2-based" at the bottom to "model 1-based" at the top. We have also clarified the main text relating to this figure and the figure legend as follows:

Even without inclusion of this extra feature, however, Model 1 provided a better account of the majority of uncorrelated sessions, while model 2 provided a better description of the majority of correlated sessions; **the difference in Akaike's information criterion (AIC) scores for Model 2 minus Model 1** are shown for a comparison of the two models for each session (Fig.2F).

(F) In general, the animals' behaviour in uncorrelated and correlated tasks was explained better by models 1 and 2 respectively. Individual grey dots indicate **the AIC difference between Model 2 and Model 1 derived from fitting the choice data of** each session and sessions from the four animals are shown in four columns as in figure 1c.

#4 (Results, Methods). Based on Fig. 1d, bottom, it appears that overall, the animals switched more often in the uncorrelated than in the negatively correlated task even though session numbers were similar. Was the total number of trials the same in the two tasks? If not, why not? If yes, was the difference in overall switching frequency adaptive and how can it be reconciled with the lower switch rates in the uncorrelated experiment as shown in Fig. 1d, top? Also, how many

trials entered the imaging analyses? Finally, Lines 648-649 state that switching became more likely as multiple errors accrued and refers to Fig. 1D to back the statement up. However, Fig. 1D suggests that if anything, the statement only holds for two consecutive errors, not for three.

First, in response to the first lines of the reviewer's comment, it is important to clarify that figure 1d does not show that animals switched more often in the uncorrelated than the negatively correlated task. Instead, the top part of figure 1d shows that, after an error, animals switched more frequently in the correlated task than in the uncorrelated. Or, more precisely, animals switch more often after two or three errors in the correlated task than in the uncorrelated task. The bottom part of the figure shows the number of events that the analysis is based on but note that these events are errors (not switches). In other words, the bottom part of the figure shows how many errors were observed, pairs of errors were observed, and triplets of consecutive errors were observed, and then the top part shows the frequency of switching from one choice to the other on the next trial. We think, therefore, that what the reviewer is referring to is that the lower panel suggests that slightly more errors occurred in the uncorrelated session but hopefully it is now clear that this is not inconsistent with an increased rate of switching after errors in the correlated sessions. We have therefore, rewritten the legend to panel D in the hope that a clearer description of what is shown will prevent misunderstanding.

(D) Across all trials, animals were unlikely to switch from one choice to the other. However, when animals did not receive a reward for a choice, they were likely to switch to choosing the alternative option on the next trial. While switching rates (top) in the correlated (blue) and uncorrelated sessions (red) were similar after one non-reward, switching was significantly more likely in correlated sessions after either two consecutive non-rewards or three consecutive non-rewards. Such a pattern of behaviour is adaptive. In the correlated sessions, the repeated failure to receive a reward for one option indicates that the other option is now likely to be rewarded. In the uncorrelated sessions the repeated failure to receive a reward reveals that that option has a low probability of reward, but it is not possible to make any inference about how good the alternative option might be. **The observations at the top of panel d could only be made on the trials that followed errors (or pairs or triplets of errors) and for clarity the lower part of the figure shows the number of observations (#obs; bottom) used when calculating the switching rate. In other words, the number trials following errors, pairs of consecutive errors, and triplets of consecutive errors, and which are included in the calculation of the switching rate in the top part of the panel are also shown in the bottom part of the panel.**

We turning next to the reviewer's comments about switching rates after two or three errors. We hope that, having clarified what is shown in each part of the panel, that it is now not surprising that the switching rate is higher after three errors in the correlated task compared with the uncorrelated task; the blue bar on the right of the top panel is much higher than the adjacent red bar just as the blue bar in the centre of the top panel is much higher than the adjacent red bar.

Turning to the reviewer's next questions, the uncorrelated sessions were slightly longer, approximately 200 trials in length, than the correlated session which were approximately 175 trials in length. There were approximately 6,000 trials correlated trials and 6,800 trials from the uncorrelated task included in the analysis. This meant that regardless of which session were

performed, animals consumed a similar amount of juice. A second benefit of such an arrangement is that the uncorrelated session is well powered so any absence or weakness of activity in the uncorrelated task cannot be attributed to a lack of power in the uncorrelated sessions. We have made this clear in the revised manuscript as follows:

There were approximately 6,000 and 6,800 included in the correlated and uncorrelated task analyses respectively.

#5 (Results). In Fig. 4, the authors analyze multiple timepoints, without providing a clear rationale and without correcting for multiple time points. This needs to be addressed properly.

Some of our time courses are provided for further illustration of the timing of effects the statistical significance of which was established at an earlier stage of analysis and so, to avoid any circularity, we refrain from performing statistical inference on the time course. In other cases, where statistical inference is needed we take care to use a leave-one-out approach to establish a single point in the time course at which to conduct our test. For example:

When we identified the peak positive effects in correlated and uncorrelated sessions after choice, using a leave-one-out (LOO) procedure, it was significantly greater in correlated than uncorrelated sessions (Wilcoxon test: $z=2.03$, $p=0.043$). Such a pattern suggests dmPFC and hippocampus interact with one another after each choice but that they continue to do so as the outcomes for the choices are received and as the reversal estimate that will be used on the next trial is established.

In summary, using this approach we focus on a single time and thereby avoid testing multiple time points.

#6 (Methods, Results). Seven animals were used; however, it remains unclear whether the behavior was similar across animals within experiment.

As suggested, we make clear in the revised manuscript that the group mean results are indicated by bars but that the data for individual sessions are also included and shown as circles. Each vertical column of circles shows the sessions from one individual. For example, the revised manuscript results now states:

Although the correlation between the reward probabilities associated with the two options was not exactly $r = -1$ in the correlated task but averaged approximately -0.798 across all schedules used, this correlation was consistently more negative than in the uncorrelated sessions (Fig.1C; **the bars indicate group mean performance but the data for each individual session are plotted as circles and the circles in each column in this figure and elsewhere are the data for each of the four individual macaques**).

The revised figure legend for figure 1 now states:

(C) In the correlated sessions, the Pearson correlation coefficient between the two options' reward probabilities for the four macaques (M1-M4) averaged at around -0.8. It was not exactly -1, but choice probabilities were still significantly more negatively correlated than in the uncorrelated sessions. Here, as elsewhere **the group mean results are indicated by bars but the data for individual sessions are also included and shown as circles. Each vertical column of circles shows the sessions from one individual.**

All the other figures include similar columns of circles corresponding to each individual animal's data (for example, figure 2F, figure 3, figure 4D, E, F, figure 5A, B, figure 6C, D, figure 7).

We next turn more specifically, to the reviewer's question about the similarity in behaviour across experiments. Behaviour in the fMRI experiment (experiment 1) and the TUS experiment (experiment 2) can be compared very directly by looking at the blue bars in figure 2 with the bars in figure 7a. They are shown side-by-side below to facilitate the comparison and to underline their similarity.

#7 (Discussion). Line 518-519 highlights the “striking” nature of differential reversal estimate encoding between correlated and uncorrelated tasks. However, only the hippocampus seems to have shown such a difference at the statistical thresholds used, and when testing the correlated task alone, the hippocampus did not seem to significantly encode reversal estimate at the same timepoint. If so, the statement should be qualified and toned down.

First, we note that in the revised manuscript we have changed “striking” to “notable”. However, perhaps more needs to be said regarding this issue because we believe that the reviewer raises an important point concerning the specificity of reversal estimate effects to the correlated task and it is one that we would also like to address clearly. We thought that we had done this but the reviewer's

comments, and our own re-reading our own manuscript, suggests that perhaps we did not do as good a job as we had hoped.

The reviewer is correct to say that it is only in the hippocampus that, after whole-brain correction for multiple comparisons, there is a significant difference between reversal estimate encoding in the correlated versus the uncorrelated task. There is not a significant difference between the correlated and uncorrelated tasks in the dmPFC after whole brain correction. However, as we realise the reviewer is aware, first, there is a risk of a type II error – a failure to report an effect that might exist – when correcting for multiple comparisons across the whole brain. Second, reversal estimate effects are significant in dmPFC in the correlated task but not in the uncorrelated task. This lack of significance in the uncorrelated task means that we cannot imply that a reversal estimate effect exists in the dmPFC in the uncorrelated task.

One further step that we can take in an attempt to understand the dmPFC activity better is we can move away from a whole brain analysis. Instead, we can focus on the dmPFC activity in an ROI and compare the reversal effects we see there in the two conditions and we can compare the dmPFC ROI with a hippocampus ROI. We could use the same LMM that we have used elsewhere (LMM3) to tests statistically whether there is a main effect of task [correlated versus uncorrelated] across all the areas and that is indeed the case: $\chi^2_{23} = 11.8686$; $p < 0.001$). Such a statistic might be taken to suggest that actually, in general, across both areas, in dmPFC and hippocampus, the pattern is basically the same: reversal estimate is encoded more strongly in the correlated than the uncorrelated task.

The catch, which we imagine that the reviewer might have spotted, is that it might be argued that there is some circularity in undertaking such an analysis; when we started looking at the dmPFC ROI we knew that it had a significant reversal effect in the correlated task and that it had no significant reversal effect in the uncorrelated task so is it correct to go ahead and test whether the reversal effect is bigger in one condition than the other in just this ROI?

If we believe that we cannot do this (as we do) and if we believe that there is a risk of a type II error if we rely only on whole brain comparisons, then there is just one test that can be performed. We can use an ANOVA approach to test whether there is any interaction between a factor of task (correlated versus uncorrelated) and a factor of brain region (dmPFC areas or hippocampus). If there is no interaction between task and brain area then it means that essentially similar information, about the correlated as opposed to the uncorrelated task, is present across dmPFC and hippocampus. On reviewing the manuscript, we felt that we had already carefully considered the issues that we are discussing here and that we had performed the correct analysis. However, our re-reading of the manuscript made us realise that perhaps we had failed to spell out the approach that we were taking carefully enough. Moreover we wondered if the implications of the result that we found were clear. We have, therefore, in the light of the reviewer's comments revised the critical paragraph as follows:

The whole brain cluster-based approach that identified the reversal estimate-related activity in dmPFC (Fig.4A) did so in data from correlated sessions. By contrast, the reversal estimate result in hippocampus (Fig4C) reflected comparison of correlated and uncorrelated sessions. One potential interpretation of such a pattern is that there is some fundamental difference in terms of how selective the **dmPFC and hippocampus** are for correlated as opposed to uncorrelated tasks. However, an

alternative interpretation is that activity in all the areas is broadly similar and, in general, it is most prominent in correlated sessions. We can arbitrate between these two accounts by testing whether activity related to the two tasks differed from one another across the three areas (Fig. 4A-C) using an ANOVA (Methods) with factors of brain area and condition as well as session. There was, however, no interaction between task and brain area ($F_{2, 156}=0.329$; $p>0.05$). This suggests that overall, the areas – **dmPFC and hippocampus** – carried similar information about the potential for a reversal of task states **in the correlated task but not in the uncorrelated task**.

#8 (Discussion). dmPFC and hippocampus apparently tracked latent states in the uncorrelated task. However, one should not see such a correlation if reversal estimates are not supposed to exist in the task. The authors could discuss these points in more detail, as well as add suggestions on how future research could go about to interrogate this unexpected activity.

We thank the reviewer for making this comment because it makes us realise that our writing was not as clear as we had hoped. As noted above in the reviewer's minor point 2, the switch estimate is a continuous variable corresponding to the animal's estimate that switching from one choice to the other is now the correct thing to do. It is derived by fitting model2 to the recordings made of each animal's behaviour and rewards in a given session in order to estimate what belief the macaque might hold on each trial given the experiences it has had. The model can even be applied to the uncorrelated task just as it can be applied to the correlated task. Even in the uncorrelated task, an animal might hold beliefs about a reversal that we can estimate. The animal's estimates, however, are unlikely to be accurate ones in the uncorrelated task because the ground truth is that reversals do not occur. This means that if neural activity were to be found that was correlated with the reversal estimate in the uncorrelated task, then that activity would be related to an estimate of the reversal probability that the animal had formed even though that estimate is not appropriate for that task. When we tested for the existence of neural activity correlated with reversal estimates in the uncorrelated task we were unable to find it when we searched across the whole brain and, in fact, we did not find it in either the dmPFC or the hippocampus ROIs (red bars on the right hand side of figures 4D, E, and F).

We also looked at activity related to the change in the reversal estimate that occurs at the time of reward/error feedback at the end of each trial which we refer to as reversal estimate D_{KL} or simply as reversal D_{KL} . We found activity related to reversal D_{KL} in one part of dmPFC but not in hippocampus (red bars on the left hand sides of figures 4D, E, and F). In aggregate with the right hand patterns of results discussed in the previous paragraph, this suggests that when animals receive feedback they may transiently attempt to update a reversal estimate even in the uncorrelated task but that it does not result in a sustained pattern of activity that for a reversal estimate at the beginning of the next trial (right hand sides of figures 4D, E, and F). Moreover figure 5 suggests that this might be because, in the uncorrelated task, but not in the correlated task, macaques update an estimate of something akin to an option specific update (ie not an estimate about the probability of a reversal in latent state but an estimate of the change in one specific choice), but the effects did not reach significance.

We have made the following changes to the text in the Results to address this point and the reviewer's related minor point 2.

One possibility is that macaques may attempt to infer latent task states even in the uncorrelated task even if none are present. We can estimate what estimates animals themselves might have about such states and the probability of switching between them – the reversal estimate – in the uncorrelated task by applying model 2 to the uncorrelated task data. By doing this we are estimating what beliefs the animals might hold at any point in time given the choices that they have made and the feedback – rewards and errors – they have observed even if those beliefs might be erroneous given the nature of the reversal task. Because the reversal estimate is a continuous rather than binary variable it may capture a weak or partial belief that the animal has about a latent state reversal, although if such beliefs are held in the uncorrelated task the reversal estimate may often be high. We can then examine whether these estimates are predictive of actual switching behaviour or of neural activity even in the uncorrelated task. In this way we can compare behaviour and neural activity in the two tasks more directly.

...The reversal estimate is, however, only furnished by model 2 and so we employed the reversal estimate from model 2. As explained above, this means that we are able to compare the strength of influence the reversal estimate might have on both behaviour and brain activity...

... Importantly, the animals were more likely to switch when the reversal estimate – model 2's estimate of the probability of reversal between one latent state and another – was high (Fig.3, right; $\chi^2_1=32.1149$; $p<0.001$) and this effect was greater in correlated sessions ($\chi^2_1=24.7756$; $p<0.001$). Even if animals might seek to identify latent states in the uncorrelated task and transitions between them, our estimates of when they might be doing so, derived from application of model 2 to the uncorrelated task suggests that they have significantly less influence in the uncorrelated task.

Analogous analyses can also be performed with the full set of five factors (chosen probability, unchosen probability, chosen uncertainty, unchosen uncertainty, reversal estimate) using LMEM2; (Supplementary Fig.S2B). These analyses confirmed that the animals' reversal estimates drove their switching behaviour more in correlated sessions even when each option's values and uncertainties were considered separately ($\chi^2_1=28.8443$; $p<0.001$) again confirming that such beliefs had a bigger impact on behaviour in the correlated task.

If the animals search for latent task states and reversals between them in the uncorrelated task then by applying model 2 to the uncorrelated task it is possible to estimate what estimates the animals themselves might have about reversal even if these are weak and partial. We were, however, unable to find significant evidence for activity linked to reversal estimates in the uncorrelated task when we examined activity across the whole brain. This was not simply a consequence of the conservative nature of the whole brain analysis approach, because ROI-based analyses similarly revealed no evidence for reversal effects in the uncorrelated task in dmPFC or hippocampus (all $r<0.406$; all $p>0.3$); even though the hippocampus and dmPFC carry reversal estimate-related activity in the correlated task, they do not do so in the uncorrelated task.

While any reversal D_{KL} -related activity in hippocampus was restricted to the correlated task, intriguingly, D_{KL} activity in anterior dmPFC was statistically greater than zero even in uncorrelated sessions (Fig.4E; $t_{34}=2.315$; $p<0.013$). It is possible that this may reflect animals attempting to identify latent task states and transitions between them even in the uncorrelated task. We return to the question of D_{KL} -related activity in dmPFC below to show that, in the uncorrelated task, the animals may possibly do this at the time of feedback, but any attempt to do so does not result in sustained

neural activity at the onset of the next trial. The activity may actually reflect tracking features of individual options as well as latent state reversal (discussed further below in Fig.5).

Activity in dmPFC and hippocampus reflected updates in reversal estimates – reversal D_{KL} . It is possible that that this may reflect the animals trying to use feedback to infer a latent task or the possibility of a transition between latent task states. If this is the case, then it does not seem to result in sustained neural activity related to the reversal estimate by the beginning of the next trial (note the absence of positive red bars on the right hand side of figure 5D, E, and F). However, it is possible that this might simply be because dmPFC and hippocampus were actually tracking some other update-related variable that resembled reversal D_{KL} . We therefore employed new versions of models 1 and 2 that yielded separate update estimates for each option. For models 1 and 2 this was under the assumption that options' reward probabilities were uncorrelated or negatively correlated, respectively (Methods, *Model Estimates*). In other words, the models yielded estimates of how much an individual option's associated reward probability was to be updated when the outcome arrived after it was chosen as opposed to an estimate of whether there was likely to be a transition between latent states. We reran the fMRI analyses in the anterior dmPFC and hippocampus ROIs shown in figure 4 but now included both reversal D_{KL} and individual option-specific update term (as appropriate for correlated and uncorrelated sessions) and allowed them to compete to explain activity. Two important results emerged. First, reversal D_{KL} continued to explain significant variance in outcome-locked activity in correlated sessions in both dmPFC and hippocampus (Fig.5, left, dmPFC: $t_{34}=2.003$, $p=0.027$; right, hippocampus: $t_{34}=1.988$, $p=0.027$). Second, the picture was very different in uncorrelated sessions in hippocampus (Fig5B, right); hippocampus showed no evidence of tracking option-specific updates in the same way as reversal D_{KL} . In anterior dmPFC in the uncorrelated task, reversal D_{KL} and option-specific update both explained less variance than they had in correlated sessions and in neither case did the effects reach significance although they were marginal in one case (reversal D_{KL} : $t_{34}=1.687$, $p=0.051$). In summary, dmPFC tracked reversal D_{KL} updates but did so most clearly when the task contained latent structure as was the case in the correlated task. Anterior dmPFC may possibly have estimated, or attempted to estimate, reversal D_{KL} even in uncorrelated sessions, perhaps as a consequence of experience of its utility in the correlated sessions which the same individuals performed. Instead in the uncorrelated task, dmPFC activity, if anything, reflected the updating of values of the individual options. However, by contrast, it was very clear that hippocampus was only concerned with tracking of reversal estimates for latent task states, and not option-specific updates, and that it did so only in the correlated task where there was latent structure.

#9 (Introduction). When using Brodmann area notation (Area 47/12o, line 76), it might be useful to mention which area this refers to as a clarification but also for readers who are less familiar with Brodmann area notations.

We have followed the reviewer's suggestions as follows:

Area 47/12o is situated just lateral to the lateral orbitofrontal sulcus in a region that has sometimes been treated as either the most lateral part of the OFC or as the most ventral part of ventrolateral prefrontal cortex; it is critical for accurate credit assignment of outcomes to choices but this process is most taxed by probabilistic reversal tasks with multiple options rather than simpler reversal tasks^{9,30}.

Previous studies have recorded activity in OFC and in adjacent regions, such as area 47/12o on the border between the OFC proper and the ventrolateral prefrontal cortex, when animals are learning

how likely choices are to lead to rewards¹⁻³. OFC activity also reflects choice reward probability and other aspects of choice value and **choice** comparison during decision making⁴⁻⁷. While lesions or manipulations of OFC lead to alterations in value-guided decision making^{6,8,9}, disruption of 47/12o causes failures of credit assignment – a diminished ability to learn which choices caused which outcomes^{9,10}. The deficit makes it especially difficult to establish which one of several choices leads to a reward when multiple choices are interleaved with one another because the credit for any reward is assigned not just to the choice that actually led to it but to other choices made at adjacent points in time^{11,12}. Area 47/12o disruption, therefore, makes it difficult for animals **to perform reversal tasks because at the point of reversal animals should** switch from one choice to another **as the** choice-reward associations change. **It is at precisely such points, when choices of different types are interleaved with one another, that 47/12o disruptions leads animals to struggle to assign rewards to the choice that actually caused them as opposed to other choices that occurred close in time but before or after. This, in turn, hampers the animal's ability to switch effectively as the reward contingencies change. However,** such impairments are **especially** apparent in stochastic and constantly changing environments in which there are many possible choices.

#10 (Methods). Function 5) in the bottom line, V2 instead of V2. Same in lines 739-740. Line 777: Liebler instead of Leibler.

Thank you for pointing out these typographic errors. They have now been corrected.

Reviewer #4 (Remarks to the Author):

Reviewer #4 (Remarks on code availability):

The code was not available

Thank you for pointing out this oversight. We have attempted to ensure that it is now accessible.

References

- Folloni D, Fouragnan E, Wittmann MK, Roumazeilles L, Tankelevitch L, et al. 2021. Ultrasound modulation of macaque prefrontal cortex selectively alters credit assignment-related activity and behavior. *Sci Adv* 7: eabg7700
- Folloni D, Verhagen L, Mars RB, Fouragnan E, Constans C, et al. 2019. Manipulation of Subcortical and Deep Cortical Activity in the Primate Brain Using Transcranial Focused Ultrasound Stimulation. *Neuron* 101: 1109-16 e5
- Fouragnan EF, Chau BKH, Folloni D, Kolling N, Verhagen L, et al. 2019. The macaque anterior cingulate cortex translates counterfactual choice value into actual behavioral change. *Nat Neurosci* 22: 797-808
- Khalighinejad N, Bongioanni A, Verhagen L, Folloni D, Attali D, et al. 2020. A Basal Forebrain-Cingulate Circuit in Macaques Decides It Is Time to Act. *Neuron* 105: 370-84 e8
- Verhagen L, Gallea C, Folloni D, Constans C, Jensen DE, et al. 2019. Offline impact of transcranial focused ultrasound on cortical activation in primates. *eLife* 8

Reviewer #1 (Remarks to the Author):

All of the concerns I previously raised have been adequately addressed and satisfactorily resolved by the authors.

Reviewer #1 (Remarks on code availability):

N/A

We are very pleased that the reviewer is satisfied with the changes made. We thank them for their helpful and constructive comments.

Reviewer #2 (Remarks to the Author):

The authors revised the manuscript in a remarkably thorough and comprehensive fashion. All points were addressed well and there are no issues left.

We are very pleased that the reviewer is satisfied with the changes made. We thank them for their helpful and constructive comments.

I have a very minor question and I would like to leave it to the authors whether they want to include a response into the manuscript: The delay between sonication and the task was quite short. Recent TUS work suggests that first TUS effects are rather local, whereas later effects can be found in the entire network. Are there any hints in the data that the TUS effects changed over the course of the experiment?

We understand that it would be interesting to know how protracted the TUS effect is. To examine this, we have re-run the LMEM analyses examining the differences in the reversal effect between sham and each of the three TUS targets but now with an additional factor time in the session (indexed by trial number). We found no differences between sham and each of the three TUS conditions when we looked at the interaction between the reversal estimate and time (all $z > -1.821$; all $p > 0.069$ – the area with a possible trend effect was the anterior dorsomedial thalamus) and, in the same analyses, the reversal estimate continued to differ between each of the three TUS conditions and sham (all $z < -3.919$; all $p < 0.001$). We have added text to the revised manuscript summarizing these results.

In order to understand how if the impact of TUS changed over the course of the session, we re-ran the analyses examining the differences in the reversal effect between sham and each of the three TUS targets but now with an additional factor time in the session (indexed by trial number). We found no differences between sham and each of the three TUS conditions when we looked at the interaction between the reversal estimate and time (all $z > -1.821$; all $p > 0.069$ – the area with a possible trend effect was the anterior dorsomedial thalamus) and, in the same analyses, the reversal estimate continued to differ between each of the three TUS conditions and sham (all $z < -3.919$; all $p < 0.001$).

Reviewer #2 (Remarks on code availability):

While OSF says that files are accessible, I can't find any. Perhaps this has to do with the recent service disruption event at OSF.

We think that this might have been due to the OSF disruption but we will check that the code is accessible when we upload the revised manuscript.

Reviewer #3 (Remarks to the Author):

Thank you for a responsive and thorough revision. My previous major point #3 needs a bit more care: There are in principle any number of regions other than hippocampus, thalamus and dmPFC that could play a role in reversal outside OFC. There should be a clear, literature-based (including references) argument for why exactly these three and not other regions are the most likely candidates. This is not provided at the moment. Alternatively, if all regions do survive whole-brain correction (as the authors argue), such a motivation would not be necessary, the identification of regions outside OFC would be data driven rather than based on ROIs and the procedure thus could be clarified more explicitly rather than in the current half-way manner.

We are very pleased that the reviewer is satisfied with most of the changes made. We thank them for their helpful and constructive comments. In line with the reviewer's suggestion regarding their remaining concern, given that we have looked across the whole-brain correction, we have removed the section in the first paragraph on page 4 that mentioned the potential areas in which activity might be found. In the revised version of the manuscript we simply mention the whole-brain imaging approach that was taken.

Reviewer #4 (Remarks to the Author):

We are very pleased that the reviewer is satisfied with the changes made. We thank them for their helpful and constructive comments.